# Counterfactual Structural Causal Bandits

**Min Woo Park**
Seoul National University
Seoul, Republic of Korea
alsdn0110@snu.ac.kr

**Sanghack Lee**[*]
Seoul National University
Seoul, Republic of Korea
sanghack@snu.ac.kr

## Abstract

Causal reasoning lies at the heart of robust and generalizable decision-making, and the *Pearl Causal Hierarchy* provides a formal language for distinguishing between observational ($\mathcal{L}_1$), interventional ($\mathcal{L}_2$), and counterfactual ($\mathcal{L}_3$) levels of reasoning. Existing bandit algorithms that leverage causal knowledge have primarily operated within the $\mathcal{L}_1$ and $\mathcal{L}_2$ regimes, treating each realizable and physical intervention as a distinct arm. That is, they have largely excluded counterfactual quantities due to their perceived inaccessibility. In this paper, we introduce a *counterfactual structural causal bandit* (CTF-SCB) framework which expands the agent's feasible action space beyond conventional observational and interventional arms to include a class of realizable counterfactual actions. Our framework offers a principled extension of structural causal bandits and paves the way for integrating counterfactual reasoning into sequential decision-making.

## 1 Introduction

The *Pearl Causal Hierarchy* (PCH) (Pearl and Mackenzie, 2018; Bareinboim et al., 2022) is a crucial milestone in our understanding of causality. The three layers of the PCH correspond to distinct regimes of reasoning about an environment: *seeing*, *doing*, and *imagining*. The first layer $\mathcal{L}_1$ represents *observational* distributions, such as $P(Y \mid x)$, the second layer $\mathcal{L}_2$ represents *interventional* distributions, e.g., $P(Y \mid do(x))$, using the *do*-operator (Pearl, 1995). The last highest layer, $\mathcal{L}_3$ represents *counterfactual* distributions addressing conflicting realities, such as effect of the treatment on the treated (ETT), $P(Y_x \mid x')$ (Heckman and Robb Jr, 1985; 1986): the distribution of $Y$ had $X$ been fixed as $x$, given that $X$ was observed to be $x'$. It is understood that higher layers subsume lower ones, but are unanswerable by them (Ibeling and Icard, 2020; Yang and Bareinboim, 2025).

It has long been believed that only $\mathcal{L}_1$ and $\mathcal{L}_2$ distributions are feasible to sample (known as *realizability* of the distributions (Raghavan and Bareinboim, 2025)) in practice; the former through passive observation of the system's natural behavior, and the latter via *Fisherian randomization* (Fisher, 1935). In contrast, $\mathcal{L}_3$ distributions (e.g., ETT) are typically considered non-realizable, since once a unit naturally adopts the decision $X = x'$, the hypothetical outcome $Y_x$ under the counterfactual intervention $do(x)$ cannot be simultaneously observed for the unit. However, Bareinboim et al. (2015), Forney et al. (2017) and Forney and Bareinboim (2019) have shown that it is feasible to draw samples from $P(Y_x, x')$ through a specific procedure called *counterfactual randomization*, in which one randomizes a unit's actual decision while also recording the natural decision that the unit would have normally taken according to its intention. More generally, Raghavan and Bareinboim (2025) characterized realizable distributions and provided guidance on how to draw such samples.

A parallel line of research has explored how causal models can be used to structure and optimize decision-making (Kumor et al., 2021; Zhang and Bareinboim, 2022; Bareinboim et al., 2024). Specifically, Bareinboim et al. (2015), Lattimore et al. (2016), and subsequent works (Lu et al., 2020; Bilodeau et al., 2022; Feng and Chen, 2023; Varici et al., 2023) have viewed multi-armed bandit (MAB) (Robbins, 1952; Lai and Robbins, 1985; Lattimore and Szepesvári, 2020) through a causal lens, where each action corresponds to an intervention on a variable, thereby grounding the bandit problem in a *structural causal model* (SCM) (Pearl, 2000). Building on this view, Lee and Bareinboim (2018; 2019) formalized a *structural causal bandit* (SCB) framework. They showed that

---

[*]Corresponding author

naively applying standard bandit algorithms to a full set of interventions can unnecessarily incur large regret, and proposed refining the action space prior to applying standard learning methods without relying on parametric assumptions. Extending this line of work, we aim to investigate SCB at a higher layer of PCH, encompassing hypothetical yet realizable actions.

**Example (AI-assisted clinical decision support system).**
Consider a hospital aiming to improve the quality of patient care. Let $Y$ denote the final *treatment outcome* (e.g., whether patients are discharged without complications), $X$ a patient's *health status*, $W$ an *expert physician's assessment*, and $Z$ the score from an *AI-assisted clinical tool*. Fig. 1a illustrates the scenario, where bidirected edges represent unobserved confounders such as history bias (Peiffer-Smadja et al., 2020). In the natural regime, a patient describes their

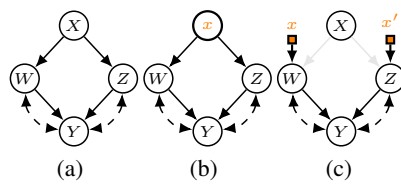

Figure 1: Counterfactual regime.

health condition $x$ in their own words, and the subsequent evaluation yields the expected outcome $\mathbb{E}Y_x$ (Fig. 1b). Alternatively, a standardized symptom-reporting interface can be introduced to translate the patient's input into a structured form $\tilde{X}$[1]. With this interface, the patient's natural description $x$ remains available for the expert assessment, while the AI tool instead perceives a reformulated and potentially different report $x'$, e.g., a version that removes exaggeration and supplements prospective information, such as the anticipated health benefits of beginning regular exercise (Fig. 1c). This setup enables the evaluation of the hypothetical intervention $\mathbb{E}Y_{W_x, Z_{x'}}$, allowing one to ask counterfactual-level questions—*"what if the AI assistant had perceived the patient's prospective health status $x'$ rather than the current one $x$?"*—and to design decision strategies that improve patient outcomes.

**Contributions.** We open the door to extending the structural causal bandit (SCB) framework to accommodate a broader class of decision-making scenarios. Our main contributions are as follows.

- We first formulate a counterfactual structural causal bandit (CTF-SCB) which enables an agent to have a wider range of considerations beyond $\mathcal{L}_2$-level interventions.
- We investigate *equivalence* (Sec. 3.1) and *partial-order* (Sec. 3.2) relations among distinct actions, along with their complete graphical characterizations.
- Building on these, we present an efficient algorithm for computing a refined action space in which redundant or verifiably suboptimal interventions are removed (Sec. 3.2.2).

Simulations in Sec. 4 corroborate our findings. All omitted proofs are provided in Appendix F.

## 2 PRELIMINARIES

Following conventions, we use a capital letter, such as $X$, to represent a variable, with its corresponding lowercase letter, $x$, denoting a realization of the variable. Boldface is employed to represent a set of variables or values, denoted by $\mathbf{X}$ or $\mathbf{x}$. The domain of $X$ is indicated by $\mathcal{D}_X$ and $\mathcal{D}_\mathbf{X} = \times_{X \in \mathbf{X}} \mathcal{D}_X$. We consistently use $P(\mathbf{x})$ as an abbreviation for $P(\mathbf{X} = \mathbf{x})$. We denote by $\mathbb{I}\{\mathbf{X} = \mathbf{x}\}$, the indicator function. Two values $\mathbf{x}$ and $\mathbf{z}$ are *consistent* if they share common values for $\mathbf{X} \cap \mathbf{Z}$. We denote $\mathbf{x} \setminus \mathbf{Z}$ the value of $\mathbf{X} \setminus \mathbf{Z}$ consistent with $\mathbf{x}$ and by $\mathbf{x} \cap \mathbf{Z}$ the subset of $\mathbf{x}$ corresponding to variables in $\mathbf{Z}$.

We use structural causal model (SCM) (Pearl, 2000) as the semantic framework to represent the underlying environment in which a decision-maker (agent) is deployed. An SCM $\mathcal{M}$ is a quadruple $\langle \mathbf{U}, \mathbf{V}, \mathcal{F}, P(\mathbf{U}) \rangle$ where $\mathbf{U}$ is a set of exogenous variables determined by factors outside the model following a joint distribution $P(\mathbf{U})$, and $\mathbf{V}$ is a set of endogenous variables whose values are determined following a collection of functions $\mathcal{F} = \{f_V\}_{V \in \mathbf{V}}$ such that $v \leftarrow f_V(\mathbf{pa}_V, \mathbf{u}_V)$ where $\mathbf{Pa}_V \subseteq \mathbf{V} \setminus \{V\}$ and $\mathbf{U}_V \subseteq \mathbf{U}$. We focus on recursive SCMs (corresponding to causal diagrams that are acyclic) over $\mathbf{V}$. The observational probability $P(\mathbf{v})$ is defined as $\sum_\mathbf{u} \prod_{V \in \mathbf{V}} \mathbb{I}\{f_V(\mathbf{pa}_V, \mathbf{u}_V) = v\}P(\mathbf{u})$. Intervention $do(\mathbf{x})$ in an SCM $\mathcal{M}$ creates a submodel $\mathcal{M}_\mathbf{x}$, where functions generating $\mathbf{X}$ are replaced with constant values $\mathbf{x}$. The functions in $\mathcal{M}_\mathbf{x}$ are denoted as $\mathcal{F}_\mathbf{x}$. Given a variable $X \in \mathbf{V}$, the solution for $X$ in $\mathcal{M}_\mathbf{w}$ defines a *potential response* for a unit $\mathbf{u}$, denoted as $X_\mathbf{w}(\mathbf{u})$. Averaging over the space of $\mathbf{U}$, a potential response $X_\mathbf{w}(\mathbf{u})$ induces a counterfactual variable $X_\mathbf{w}$. We use bracketed subscripts when the variables already have index subscripts (e.g. $X_{1,\mathbf{w}_1}$ to $X_{1[\mathbf{w}_1]}$).

---

[1]This *counterfactual mediator* $\tilde{X}$ (Raghavan and Bareinboim, 2025) fully encodes information about the variable $X$, and mediates how $Z$ perceives the value of $X$.

We denote by $\mathbf{X}_{\mathbf{w}} = \{X_{i[\mathbf{w}]}\}_{i=1}$ the set of counterfactual variables that share the same subscript $\mathbf{w}$. Moreover, $\mathbf{X}_* = \{X_{1[\mathbf{w}_1]}, X_{2[\mathbf{w}_2]}, \cdots\}$ represents an arbitrary counterfactual event (a set of counterfactual variables). We denote $\mathbf{V}(\mathbf{X}_*) = \{X \in \mathbf{V} \mid X_{\mathbf{w}} \in \mathbf{X}_*\}$. When it causes no confusion, we abbreviate $\mathbf{V}(\mathbf{X}_*)$ as $\mathbf{X}$. Every SCM $\mathcal{M}$ is associated with a *causal diagram* (also called a semi-Markovian graph) $\mathcal{G} = \langle \mathbf{V}, \mathbf{E} \rangle$ where a directed edge $V_i \to V_j \in \mathbf{E}$ if $V_i \in \mathbf{Pa}_{V_j}$, and a bidirected edge between $V_i$ and $V_j$ if $\mathbf{U}_{V_i}$ and $\mathbf{U}_{V_j}$ are not independent. $\mathcal{G}[\mathbf{X}]$ denotes an induced graph over $\mathbf{X}$. We use kinship notation for graphical relationships: parents (Pa), children (Ch), descendants (De), and ancestors (An). Note that $\mathrm{Pa}(V)_{\mathcal{G}}$ corresponds to $\mathbf{Pa}_V$. We denote the set of variables and edges in $\mathcal{G}$ by $\mathbf{V}(\mathcal{G})$ and $\mathbf{E}(\mathcal{G})$, respectively.

An SCM induces all quantities within PCH; for any $\mathbf{Y}, \mathbf{Z}, \cdots, \mathbf{X}, \mathbf{W} \subseteq \mathbf{V}$, the three layers of distributions are given by: (Observational; $\mathcal{L}_1$): $P(\mathbf{y}) = \sum_{\mathbf{u}} \mathbb{I}\{\mathbf{Y}(\mathbf{u}) = \mathbf{y}\} P(\mathbf{u})$; (Interventional; $\mathcal{L}_2$): $P(\mathbf{y}_{\mathbf{x}}) = \sum_{\mathbf{u}} \mathbb{I}\{\mathbf{Y}_{\mathbf{x}}(\mathbf{u}) = \mathbf{y}\} P(\mathbf{u})$; and (Counterfactual; $\mathcal{L}_3$): $P(\mathbf{y}_{\mathbf{x}}, \cdots, \mathbf{z}_{\mathbf{w}}) = \sum_{\mathbf{u}} \mathbb{I}\{\mathbf{Y}_{\mathbf{x}}(\mathbf{u}) = \mathbf{y}, \cdots, \mathbf{Z}_{\mathbf{w}}(\mathbf{u}) = \mathbf{z}\} P(\mathbf{u})$. We refer the reader to Appendix B for additional background details.

# 3 COUNTERFACTUAL STRUCTURAL CAUSAL BANDITS

We now formalize the *counterfactual structural causal bandit* (CTF-SCB) problem where an agent interacts with a target system modeled by a structural causal model (SCM) $\mathcal{M} = \langle \mathbf{V}, \mathbf{U}, \mathcal{F}, P(\mathbf{U}) \rangle$ including a reward variable $Y \in \mathbf{V}$. While traditional SCB (Lee and Bareinboim, 2018) optimizes $\mathbb{E}Y_{\mathbf{x}} = \mathbb{E}[Y \mid do(\mathbf{X} = \mathbf{x})]$, we generalize this notion to $\mathbb{E}Y_{\mathbf{X}_*} = \mathbb{E}[Y \mid do(\mathbf{X} = \mathbf{X}_*)]$ where $\mathbf{X}_* = \{X_{1[\mathbf{w}_1]}, X_{2[\mathbf{w}_2]}, \cdots\}$ with $X_i \in \mathbf{V} \setminus \{Y\}$ and $\mathbf{W}_i \subseteq \mathbf{V} \setminus \{Y\}$. This operation implies that each $X_i \in \mathbf{X}$ behaves as another counterfactual variable $X_{i[\mathbf{w}_i]} \in \mathbf{X}_*$ (Correa and Bareinboim, 2025). In other words, the value of $X_{i[\mathbf{w}_i]}$ is computed in a submodel $\mathcal{M}_{\mathbf{w}_i}$ and used to replace the natural mechanism $f_{X_i} \in \mathcal{F}$.

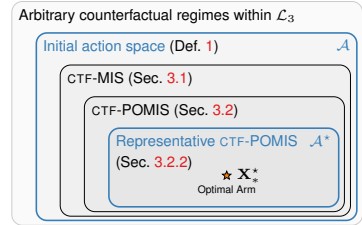

Figure 2: We aim to reduce the total action space $\mathcal{A}$ to a subspace $\mathcal{A}^\star$, while ensuring $\mathbb{E}R_T^{\mathcal{A}^\star} \leq \mathbb{E}R_T^{\mathcal{A}}$.

For concreteness, consider the introductory example in Fig. 1. While $\mathbb{E}Y_x$ represents the expected reward from $\mathcal{M}_x$ where $f_X$ is fixed to a constant $x$, $\mathbb{E}Y_{W_x, Z_{x'}}$ represents the case where the value of $W$ is computed in $\mathcal{M}_x$ while the value of $Z$ is computed in $\mathcal{M}_{x'}$, and then $Y$ is determined within $\mathcal{M}$. Note that $\mathbb{E}Y_x = \mathbb{E}Y_{W_x, Z_x}$ holds since fixing $X = x$ is equivalent to making *all* children of $X$ receive $x$ as their argument instead of $X$; thus, counterfactual regimes are more *fine-grained* than existing $\mathcal{L}_{\leq 2}$ regimes.

We use the terms *arm*, *action*, and *intervention* interchangeably, depending on the context. Throughout this paper, we assume that the causal diagram $\mathcal{G}$ representing $\mathcal{M}$ is fully accessible to the agent, although its parameterization is unknown; that is, an agent plays arms with knowledge of $\mathcal{G}$ and $Y$, but not of $\mathcal{F}$ and $P(\mathbf{U})$. At each time step $t \in \{1, \cdots, T\}$, the agent interacts with a bandit instance by pulling an arm $\mathbf{X}_*$, and subsequently observes the reward $Y$ (i.e., $Y_{\mathbf{X}_*}$)[2].

**Definition 1** (Action space). A total action space of CTF-SCB $\mathcal{A}$ is a set of counterfactual variables $\mathbf{X}_*$, for which the corresponding reward distribution $P(Y_{\mathbf{X}_*})$ is realizable[3]. That is, given a causal diagram $\mathcal{G}$, an agent can interact with any SCM compatible with $\mathcal{G}$ and obtain rewards $Y_{\mathbf{X}_*}$.

**Definition 2** (Counterfactual structural causal bandits). The goal of a counterfactual structural causal bandit agent is to minimize cumulative regret defined as follows:

$$R_T^{\mathcal{A}} = T\mu_{\mathbf{X}_*^\star} - \sum_{t=1}^{T} \mu_{\mathbf{X}_*^{(t)}} = \sum_{\mathbf{X}_* \in \mathcal{A}} \Delta_{\mathbf{X}_*} N_T(\mathbf{X}_*), \qquad (1)$$

where mean reward is denoted by $\mu_{\mathbf{X}_*}$. Moreover, $\mathbf{X}_*^{(t)}$ refers to the chosen arm in each round $t$ following some strategy of the agent, and $\mathbf{X}_*^\star$ is an optimal arm. $\Delta_{\mathbf{X}_*}$ denotes suboptimal gap $\mu_{\mathbf{X}_*^\star} - \mu_{\mathbf{X}_*}$ and $N_T(\mathbf{X}_*)$ denotes the number of times an action $\mathbf{X}_*$ was chosen up to round $T$.

---

[2]One may raise a concern regarding actions based on *nested* counterfactuals (Correa et al., 2021). However, any realizable nested counterfactual regimes such as $\{Z_{i[\mathbf{T}_*^i]}\}_{i=1}$ are dominated by some action $\mathbf{X}_* \in \mathcal{A}^\star$, which is deferred to Appendix E.4 for interested readers.

[3]We say that a distribution is *realizable* if samples can be drawn from it through a sequence of physical operations. A formal definition and background are deferred to Appendix B.3.

**Corollary 1.** *Let* $\mathbf{x}^\star = \arg\max_{\mathbf{x} \in \mathcal{D}_\mathbf{x}, \mathbf{X} \subseteq \mathbf{V} \setminus \{Y\}} \mu_\mathbf{x}$ *be an optimal arm in* $\mathcal{L}_{\leq 2}$. *Then,* $\mu_{\mathbf{x}^\star} \leq \mu_{\mathbf{X}_*^\star}$.

To provide the condition under which a counterfactual event constitutes a valid action, we introduce the notion of ancestral relation among counterfactuals (Correa et al., 2021). The set of ancestors of $X_\mathbf{w}$, denoted by $\mathtt{An}(X_\mathbf{w})$, consists of each counterfactual variable $T_\mathbf{z}$ such that (i) $T \in \mathtt{An}(X)_{\mathcal{G}_{\overline{\mathbf{W}}}} \setminus \mathbf{W}$, and (ii) $\mathbf{z} = \mathbf{w} \cap \mathtt{An}(T)_{\mathcal{G}_{\overline{\mathbf{W}}}}$. For a set of variables $\mathbf{X}_*$, we define $\mathtt{An}(\mathbf{X}_*) = \bigcup_{X_\mathbf{w} \in \mathbf{X}_*} \mathtt{An}(X_\mathbf{w})$. For instance, given $\mathbf{X}_* = \{W_x, Z_{x'}\}$ in Fig. 3a, we have $\mathtt{An}(\mathbf{X}_*) = \{W_x, Z_x\} \cup \{Z_{x'}\} = \{W_x, Z_x, Z_{x'}\}$.

**Proposition 1.** *A counterfactual* $\mathbf{X}_*$ *consists of* CTF-*SCB action space* $\mathcal{A}$ *if and only if* $\mathtt{An}(Y_\mathbf{x}, \mathbf{X}_*)$ *does not contain a pair of* $X_\mathbf{w}, X_\mathbf{t}$ *of the same variable* $X$ *under different regimes where* $\mathbf{w} \neq \mathbf{t}$.

*Proof.* According to *counterfactual unnesting theorem* (CUT) (Correa et al., 2021):
$$P(Y_{\mathbf{T}_*, X_\mathbf{z}} = y) = \sum_x P(Y_{\mathbf{T}_*, x} = y, X_\mathbf{z} = x), \tag{2}$$
where $\mathbf{T}_*$ represents any combination of counterfactuals, we obtain $P(y_{\mathbf{X}_*}) = \sum_\mathbf{x} P(y_\mathbf{x}, \mathbf{X}_* = \mathbf{x})$. Hence, checking for conflicts (i.e., $X_\mathbf{w}, X_\mathbf{t} \in \mathtt{An}(Y_\mathbf{x}, \mathbf{X}_*)$ with $\mathbf{w} \neq \mathbf{t}$) establishes the realizability of $Y_{\mathbf{X}_*}$ by Corollary 3.7 in Raghavan and Bareinboim (2025) (see Lem. 4 in Appendix B.3). $\qquad\square$

For concreteness, consider the causal diagram shown in Fig. 3a. Suppose an agent intends to perform $\mathbf{X}_* = \{W_x, Z_{x'}\}$ with $x \neq x'$. In this setting, the value of $Y_{W_x, Z_{x'}}$ is computed under a system in which $W_x$ is evaluated within the submodel $\mathcal{M}_x$ while $Z_{x'}$ is evaluated within a separate submodel $\mathcal{M}_{x'}$. However, for $Y$ to *listen* to $W_x$, the mechanism $f_Z$ must receive $x$ as input, leading to $Z_x$ (shown in blue). Conversely, for $Y$ to simultaneously *listen* to $Z_{x'}$, the same mechanism $f_Z$ must receive $x'$ (shown in red). These conflicting requirements on $f_Z$ render $\mathbf{X}_*$ non-realizable, making it infeasible to sample rewards through interaction with the system; thus $\mathbf{X}_* = \{W_x, Z_{x'}\} \notin \mathcal{A}$. Leveraging Prop. 1, we have $\mathtt{An}(Y_\mathbf{x}, \mathbf{X}_*) = \{Y_{wz}, W_x, Z_x, Z_{x'}\}$ which contains a *conflict*, i.e., the same variable $Z$ instantiated under different subscripts $x$ and $x'$. Similarly, the regime $\mathbf{X}_* = \{Z_w\}$ in Fig. 3b is also *not* a valid action either, since $\mathtt{An}(Y_\mathbf{x}, \mathbf{X}_*) = \{Y_z, X, W, T, X_w\}$ induces a conflict at $X$.

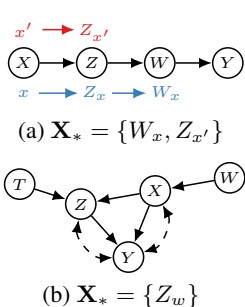

(a) $\mathbf{X}_* = \{W_x, Z_{x'}\}$

(b) $\mathbf{X}_* = \{Z_w\}$

Figure 3: Invalid actions.

In contrast, consider $\mathbf{X}_* = \{W_x, Z_{x'}\}$ in the introductory example (Fig. 1), which corresponds to $Y_{W_x, Z_{x'}}$. To evaluate this regime, we input $x$ and $x'$ into $f_W$ and $f_Z$, respectively, thereby simulating the counterfactual behavior of both variables. Since intervention on $\mathbf{X}_* = \{W_x, Z_{x'}\}$ does not induce any conflict, it constitutes a valid action, identified by $\mathtt{An}(Y_\mathbf{x}, \mathbf{X}_*) = \{Y_{wz}, W_x, Z_{x'}\}$.

Through the remaining parts, we first introduce equivalence relations among arms (Sec. 3.1). Next, we explore a partial order over arms that characterizes "possibly optimal" arms (Sec. 3.2). Using these properties, we shrink $\mathcal{A}$ into a subspace $\mathcal{A}^\star$ allowing agents to efficiently optimize their policy without sacrificing unnecessary exploration. The schema in Fig. 2 shows the overall flow.

## 3.1 COUNTERFACTUAL MINIMAL INTERVENTION SET

Although the semantics of counterfactuals permit considering any regime of the form $\mathbf{X}_* = \{X_{i[\mathbf{w}_i]}\}_{i=1}$—involving exponential combinations of $X_i \in \mathbf{V} \setminus \{Y\}$ and $\mathbf{W}_i \subseteq \mathbf{V} \setminus \{Y\}$—certain counterfactual variables $X_\mathbf{w} \in \mathbf{X}_*$ may be irrelevant to the reward, depending on the topology of causal systems, indicating $\mu_{\mathbf{X}_*} = \mu_{\mathbf{X}_* \setminus \{X_\mathbf{w}\}}$. CTF-calculus (Correa and Bareinboim, 2025) (counterpart of do-calculus for $\mathcal{L}_3$) provides a set of rules for assessing such invariances within the action space. To systematically handle these equivalence relations, we first introduce a key tool, *interventional minimization* (Correa et al., 2021).

**Lemma 1** (Interventional minimization). *Let* $X_\mathbf{w}$ *be a counterfactual variable,* $\mathcal{G}$ *a causal diagram, and* $X_\mathbf{z}$ *such that* $\mathbf{z} = \mathbf{w} \cap \mathtt{An}(X)_{\mathcal{G}_{\overline{\mathbf{W}}}}$. *Then,* $X_\mathbf{w} = X_\mathbf{z}$ *holds for any SCM compatible with* $\mathcal{G}$.

This transformation is denoted as $\|X_\mathbf{w}\| \triangleq X_{\mathbf{w} \cap \mathtt{An}(X)_{\mathcal{G}_{\overline{\mathbf{W}}}}}$. For instance, $\|W_{xz}\| = W_z$ in Fig. 3 by $\{x, z\} \cap \mathtt{An}(W)_{\mathcal{G}_{\overline{\{X,Z\}}}} = \{x, z\} \cap \{W, Z\} = \{z\}$. For a set of counterfactual variables $\mathbf{X}_*$, we define $\|\mathbf{X}_*\| = \bigcup_{X_\mathbf{w} \in \mathbf{X}_*} \|X_\mathbf{w}\|$.

**Definition 3** (Counterfactual minimal intervention set). A set of counterfactual variables $\mathbf{X}_*$ satisfying $\mathbf{X}_* = \|\mathbf{X}_*\|$ is a *counterfactual minimal intervention set* (CTF-MIS) if there is no other counterfactuals $\mathbf{Z}_* \subsetneq \mathbf{X}_*$ such that $\mu_{\mathbf{X}_*} = \mu_{\mathbf{Z}_*}$[4] for every SCM conforming to $\mathcal{G}$.

In words, a CTF-MIS implies that every variable $X_{i[\mathbf{w}_i]} \in \mathbf{X}_*$ affects the reward, and it is sufficient to play only that arm among those in its equivalence class with respect to expected reward.

**Theorem 1** (Graphical characterization of CTF-MIS). *A counterfactual $\mathbf{X}_* = \{X_{i[\mathbf{w}_i]}\}_{i=1} \in \mathcal{A}$ is a CTF-MIS if and only if (i) $\mathbf{X} \subseteq \mathrm{An}(Y)_{\mathcal{G}_{\overline{\mathbf{X}}}}$ and (ii) for any $X_{i[\mathbf{w}_i]} \in \mathbf{X}_*$, $\mathbf{W}_i \cap \mathrm{An}(X_i)_{\mathcal{G}_{\overline{\mathbf{X} \setminus \{X_i\}}}} \neq \emptyset$.*

Verbally speaking, the first condition states that for $\mathbf{X}_*$ to be a CTF-MIS, each $X_i \in \mathbf{X}$ must have its own causal (directed) path to $Y$; otherwise, another minimal equivalent set could be identified.

For example, consider Fig. 3a with $\mathbf{X}_* = \{W_x, Z_x\} \in \mathcal{A}$, which is *not* a CTF-MIS since a subset $\{W_x\} \subset \mathbf{X}_*$ is equivalent to $\mathbf{X}_*$ by the following derivation:

$$\mu_{\mathbf{X}_*} \overset{\text{CUT}}{=\!=} \sum_{ywz} yP(y_{wz}, w_x, z_x) \overset{\text{R3}}{=\!=} \sum_{ywz} yP(y_w, w_x, z_x) = \sum_{yw} yP(y_w, w_x) \overset{\text{CUT}}{=\!=} \mu_{W_x}.$$

Here, the first and last equalities follow from CUT, while the second follows from CTF-Rule 3 of CTF-calculus. Graphically, one observes that the only causal path from $Z$ to $Y$ passes through $W$, implying $\mathbf{X} = \{W, Z\} \not\subseteq \mathrm{An}(Y)_{\mathcal{G}_{\overline{\{W,Z\}}}} = \{W, Y\}$.

To explain the second condition, consider $\mathbf{X}_* = \{X_w, Z_{w'}\}$ in Fig. 3b which satisfies the first condition; namely, $\{X, Z\} \subseteq \mathrm{An}(Y)_{\mathcal{G}_{\overline{\{X,Z\}}}} = \{X, Z\}$. Nevertheless, the only causal path from $W$ to $Z$ passes through $X$. This means that intervening on $X_w$ induces $Z_w$ (i.e., $w = w'$, otherwise it violates Prop. 1), which implies that intervention on $X_w$ alone is sufficient, as shown below:

$$\mu_{\mathbf{X}_*} \overset{\text{CUT}}{=\!=} \sum_{yxz} yP(y_{xz}, z_{w'}, x_w) \overset{\text{Prop. 1}}{=\!=\!=} \sum_{yxz} yP(y_{xz}, z_w, x_w) \overset{\text{R1}}{=\!=} \sum_{yxz} yP(y_{xz}, z_{xw}, x_w)$$

$$\overset{\text{R3}}{=\!=} \sum_{yxz} yP(y_{xz}, z_x, x_w) \overset{\text{R1}}{=\!=} \sum_{yxz} yP(y_x, z_x, x_w) = \sum_{yx} yP(y_x, x_w) \overset{\text{CUT}}{=\!=} \mu_{X_w}$$

where R1 and R3 denote CTF-Rule 1 and CTF-Rule 3, respectively. As a result, $\mathbf{X}_*$ is *not* a CTF-MIS. Remark that $\{W\} \cap \mathrm{An}(Z)_{\mathcal{G}_{\overline{\{X\}}}} = \emptyset$ allows the application of CTF-Rule 3 in the second line and introduces the equivalence $\mu_{\mathbf{X}_*} = \mu_{X_w}$.

**Further equivalences.** Surprisingly, we will show that there is additional room to extract equivalence relationships among counterfactuals. Specifically, although these relationships do not stem from subset inclusion, they nevertheless constitute genuine equivalence relationships.

As an intuition pump, consider the causal diagram shown in Fig. 4a with two CTF-MISs $\mathbf{X}_* = \{W_x, Z_{x'}\}$ and $\mathbf{Z}_* = \{T_x, Z_{x'}\}$; the only difference is that $W$ listens to $x$ in $\mathbf{X}_*$ rather than $T$ in $\mathbf{Z}_*$. We will derive $\mu_{\mathbf{Z}_*} = \mu_{\mathbf{X}_*}$. Applying CUT and marginalization over $W_x$, we have:

$$\mu_{T_x, Z_{x'}} \overset{\text{CUT}}{=\!=} \sum_{y,t,z} yP(y_{tz}, t_x, z_{x'}) = \sum_{y,z,t,w} yP(y_{tz}, t_x, z_{x'}, w_x).$$

We proceed to derive as follows:

$$\overset{\text{R3}}{=} \sum_{y,z,t,w} yP(y_{tzw}, t_x, z_{x'}, w_x) \qquad \{W\} \cap \mathrm{An}(Y)_{\mathcal{G}_{\overline{\{T,Z\}}}} = \emptyset$$

$$\overset{\text{R1}}{=} \sum_{y,z,t,w} yP(y_{tzw}, t_{wx}, z_{x'}, w_x) \qquad w_x \Rightarrow t_x = t_{wx}$$

$$\overset{\text{R3}}{=} \sum_{y,z,t,w} yP(y_{tzw}, t_w, z_{x'}, w_x) \qquad \{X\} \cap \mathrm{An}(T)_{\mathcal{G}_{\overline{\{W\}}}} = \emptyset$$

$$\overset{\text{R3}}{=} \sum_{y,z,t,w} yP(y_{tzw}, t_{zw}, z_{x'}, w_x) \qquad \{Z\} \cap \mathrm{An}(T)_{\mathcal{G}_{\overline{\{W\}}}} = \emptyset$$

$$\overset{\text{R1}}{=} \sum_{y,z,t,w} yP(y_{zw}, t_{zw}, z_{x'}, w_x). \qquad t_{zw} \Rightarrow y_{tzw} = y_{zw}$$

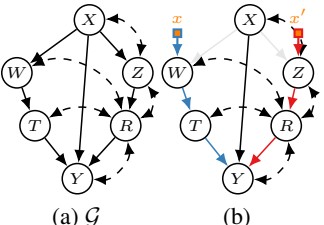

(a) $\mathcal{G}$      (b)

Figure 4: (a) Causal diagram (b) $\mathbf{X}_*$ and $\mathbf{Z}_*$ are equivalent, as they induce the same effect propagation from $x$ and $x'$ to $Y$.

Summation over $T_{zw}$ and then applying CUT again results in:

$$\sum_{y,z,t,w} yP(y_{zw}, t_{zw}, z_{x'}, w_x) = \sum_{y,z,w} yP(y_{zw}, z_{x'}, w_x) \overset{\text{CUT}}{=\!=} \mu_{W_x, Z_{x'}}.$$

---

[4]We refer to $\mathbf{X}_*$ and $\mathbf{Z}_*$ as *equivalent* if the equality holds. For example, $\mathbf{X}_*$ and $\|\mathbf{X}_*\|$ are always equivalent.

Therefore, we find that the two CTF-MISs $\mathbf{X}_* = \{W_x, Z_{x'}\}$ and $\mathbf{Z}_* = \{T_x, Z_{x'}\}$ are equivalent. Graphically, one can observe that whether an agent performs $\mathbf{X}_*$ or $\mathbf{Z}_*$, the subscript $x$ propagates to $Y$ through $W$ and $T$ as in the world $\mathcal{M}_x$ (blue in Fig. 4b), while $x'$ propagates through $Z$ and $R$ as in $\mathcal{M}_{x'}$ (red). In other words, $Y$ cannot distinguish whether the input to the functional mechanism $f_Y$ is governed by $T_x$ or $W_x$. Furthermore, consider a CTF-MIS $\mathbf{T}_* = \{T_x, R_{x'}\}$ which is also equivalent to $\mathbf{X}_*$ and $\mathbf{Z}_*$. The key point is that their propagation can be represented in the same graph (Fig. 4b). This property will serve a useful role in the next section.

**Remark 1.** There are *no* equivalence relations among minimal intervention sets (MISs)[5] in SCB; thus, any MIS can be thought of as a representative set among all possible equivalence classes, but this does *not* hold in CTF-SCB.

## 3.2 COUNTERFACTUAL POSSIBLY-OPTIMAL MINIMAL INTERVENTION SET

We now characterize partial orders among actions within CTF-MISs. Given a causal diagram $\mathcal{G}$, it is possible that counterfactual interventions on certain variables always perform at least as well as interventions on other variables, regardless of the parameterization of the underlying model.

To see this, consider the causal diagram $\mathcal{G}$ in Fig. 5a. Here, we can derive $\mu_{z^*} = \sum_x \mathbb{E}[Y \mid do(z^*), x] P(x \mid do(z^*)) = \sum_x \mu_x P(x \mid do(z^*)) \leq \mu_{x^*}$ and $\mu_\emptyset = \sum_w \mathbb{E}[Y \mid w] P(w) \leq \sum_w \mu_{w^*} P(w) = \mu_{w^*}$ where the superscript $*$ denotes the best arm in the corresponding domain (i.e., $w^* = \arg\max_{w \in \mathcal{D}_w} \mu_w$). This means that both $do(\emptyset)$ and $do(z^*)$ cannot be better than the other.

Beyond partial ordering among such physical interventions over $\mathcal{L}_{\leq 2}$, we now consider comparisons involving counterfactual actions. Let us now select $w^\dagger = \arg\max_{w \in \mathcal{D}_W} \mu_{w^* Z_w}$; this licenses $\mu_{w^*} = \mu_{w^* Z_{w^*}} \leq \mu_{w^* Z_{w^\dagger}}$. For concreteness, consider an SCM with binary variables and the following mechanisms: $f_W = u_W$, $f_Z = u_Z \vee w \oplus u_{ZY}$, $f_X = u_X \oplus z$ and $f_Y = 1 - (u_Y \oplus u_{ZY} \oplus x \oplus (1 - w))$ where all exogenous follow $\texttt{Bern}(0.1)$. In this setting, we obtain $\mu_\emptyset \approx 0.22 \leq 0.24 \approx \mu_{w^*}$ with $w^* = 0$, by fixing its mechanism $f_w$ to a constant function $f'_w = w^*$. Meanwhile the expected reward under the counterfactual arm $\mu_{w^* Z_{w^\dagger}} \approx 0.88$ with $w^\dagger = 1$ dominates $\mu_{w^*} \approx 0.24$.

To build more intuition, we rewrite $\mu_{w^*} = \mu_{W_{w^*} Z_{w^*}}$ and $\mu_{w^* Z_w} = \mu_{W_{w^*} Z_w}$, respectively. This highlights that the two arms differ only in how the variable $Z$ is handled; in the former, $Z$ behaves as it would under the factual intervention $w^*$, whereas in the latter, $Z$ behaves as if $w$ had been applied, as shown in Fig. 5b. If the hypothetical action $Z_w$ yields a higher expected reward than $Z_{w^*}$, the agent should prefer it; otherwise, if their outcomes coincide, $w$ may simply be set to $w^*$. This discrepancy in how $Z$ *listens* to $w$ ultimately propagates to $Y$, resulting in a significantly improved outcome $\mu_{w^*} < \mu_{w^* Z_{w^\dagger}}$ in this SCM instance.

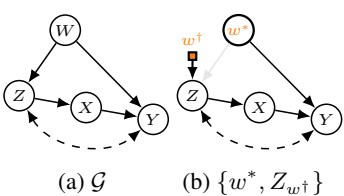

(a) $\mathcal{G}$      (b) $\{w^*, Z_{w^\dagger}\}$

Figure 5: Possibly-optimal action.

**Definition 4** (Counterfactual possibly-optimal intervention set). Let $\mathbf{X}_*$ be a CTF-MIS relative to $\langle \mathcal{G}, Y \rangle$. If there exists an SCM conforming to $\mathcal{G}$ such that $\mu_{\mathbf{X}_*} > \mu_{\mathbf{Z}_*}$ for any non-equivalent CTF-MIS $\mathbf{Z}_*$, then $\mathbf{X}_*$ is a *counterfactual possibly-optimal minimal intervention set* (CTF-POMIS).

We begin by observing a simple setting in which exogenous variables associated with the reward are uncorrelated with exogenous variables of all other endogenous variables (in graphical terms, there is no bidirected edge connecting with $Y$). In this setting, a natural intuition is to directly intervene on the parents of $Y$, assigning them the values that $Y$ most prefers to *listen* to. Fortunately, this leads to a desirable result. Concretely, we observe the values $\mathbf{pa}_Y$ when $\mathbf{X}^*_*$ is intervened upon (best arm), and then *mimic* those values by directly intervening on $\mathrm{Pa}(Y)_\mathcal{G}$. This approach guarantees $\mu_{\mathbf{X}_*} \leq \mu_{\mathbf{pa}^*_Y}$.

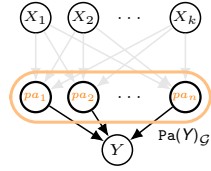

Figure 6: Markovian

To witness, revisit the causal diagram shown in Fig. 3a, equipped with an SCM where mechanisms are defined as $f_V = (\bigwedge \mathbf{pa}_V) \wedge (\bigwedge \mathbf{u}_V)$, and each binary exogenous $U \in \mathbf{U} \setminus \mathbf{U}_Y$ follows $P(U = 1) = \varepsilon \approx 0$, while $U \in \mathbf{U}_Y$ follows $P(U = 1) = 1 - \varepsilon$. In this setting, consider an arm $\mathbf{X}^*_* = \{Z_x\}$ with $x = 1$. We can observe that $\mu_{Z_x} = \varepsilon(1 - \varepsilon)$. Moreover, we find $P(W = 1 \mid Y_{Z_x} = 1) = 1$, indicating that directly fixing $W = 1$ (i.e., $\mathbf{pa}^*_Y$) would result in a better strategy; thus, $\mu_{Z_x} \leq \mu_{\mathbf{pa}^*_Y}$.

---

[5]Formal definitions of the minimal intervention set (MIS) and possibly-optimal minimal intervention set (POMIS) (Lee and Bareinboim, 2018) are provided in Appendix B.1.

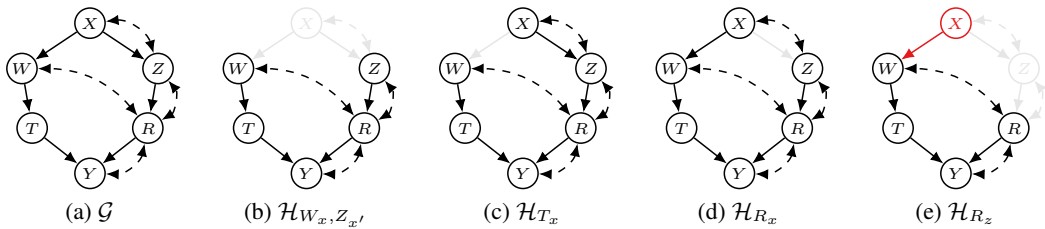

Figure 7: (a) Causal diagram; (b–e) counterfactual regime graphs corresponding to distinct CTF-MISs. The first three CTF-MISs are CTF-POMISs, whereas the last one is *not* by $\mathsf{IB}(\mathcal{H}_{\mathbf{X}_*}, Y) = \{X\} \neq \emptyset$.

**Proposition 2.** *If $Y$ is not confounded with $\mathsf{An}(Y)_\mathcal{G} \setminus \{Y\}$, then $\mathsf{Pa}(Y)_\mathcal{G}$ is the only CTF-POMIS.*

**Corollary 2** (Markovian CTF-POMIS). *if $\mathcal{G}$ is Markovian, then $\mathsf{Pa}(Y)_\mathcal{G}$ is the only CTF-POMIS.*

Therefore, in Markovian settings, agents do not need to consider counterfactual actions. Instead, it suffices to intervene on the parents of $Y$. However, this conclusion may no longer hold if $Y$ is confounded with variables that causally affect it. In the previous example (Fig. 5), we already observe that $\mu_{\mathbf{pa}_Y^*} \approx 0.82 < \mu_{w^* Z_{w^\dagger}}$, indicating that interventions on the parents are suboptimal.

### 3.2.1 GRAPHICAL CHARACTERIZATION OF CTF-POMIS

In this section, we graphically characterize CTF-POMISs. We first introduce unobserved confounders' territory and interventional border, proposed by Lee and Bareinboim (2018). Let $\mathsf{cc}(X)_\mathcal{G}$ be a c-component (Tian and Pearl, 2003) containing $X$ in $\mathcal{G}$, and $\mathsf{cc}(\mathbf{X})_\mathcal{G} = \bigcup_{X \in \mathbf{X}} \mathsf{cc}(X)_\mathcal{G}$.

**Definition 5** (Unobserved-confounders' territory (Lee and Bareinboim, 2018)). Let $\mathcal{H} = \mathcal{G}[\mathsf{An}(Y)_\mathcal{G}]$. A set of variables $\mathbf{T} \subseteq \mathbf{V}(\mathcal{H})$ containing $Y$ is called a *UC-territory* on $\mathcal{G}$ with respect to $Y$ if $\mathsf{De}(\mathbf{T})_\mathcal{H} = \mathbf{T}$ and $\mathsf{cc}(\mathbf{T})_\mathcal{H} = \mathbf{T}$. The UC-territory $\mathbf{T}$ is said to be minimal, denoted $\mathbf{T} = \mathsf{MUCT}(\mathcal{G}, Y)$, if no $\mathbf{T}' \subsetneq \mathbf{T}$ is a UC-territory.

**Definition 6** (Interventional border (Lee and Bareinboim, 2018)). Let $\mathbf{T}$ be a minimal UC-territory on causal diagram $\mathcal{G}$ with respect to $Y$. Then $\mathsf{Pa}(\mathbf{T})_\mathcal{G} \setminus \mathbf{T}$ is called an *interventional border* (IB) for $\mathcal{G}$ with respect to $Y$, denoted as $\mathsf{IB}(\mathcal{G}, Y)$.

Minimal UC-territory represents the smallest closed set that transmits all hidden information from unobserved confounders to the downstream reward, while interventional border comprises the nodes that directly influence this closed mechanism. Lee and Bareinboim (2018) demonstrated that $\mathsf{IB}(\mathcal{G}_{\overline{\mathbf{X}}}, Y) = \mathbf{X}$ is a sound and complete condition for $\mathbf{X}$ to be a POMIS.

**Counterfactual regime graph.** We will show that interventional borders can also fully characterize CTF-POMISs when considered with a graph specially designed to preserve causal relations among counterfactual variables of interest, called a *counterfactual regime graph*. Given a CTF-MIS $\mathbf{X}_*$, we define the counterfactual regime graph for $\mathbf{X}_*$ as $\mathcal{H}_{\mathbf{X}_*} = \langle \mathbf{V}^\dagger, \mathbf{E}^\dagger \rangle$ where $\mathbf{V}^\dagger = \mathbf{V}(\mathsf{An}(Y_\mathbf{x}, \mathbf{X}_*))$ and $\mathbf{E}^\dagger = \mathbf{E}(\mathcal{G}[\mathbf{V}^\dagger])$. For concreteness, consider the causal diagram $\mathcal{G}$ in Fig. 7a with a CTF-MIS $\mathbf{X}_* = \{W_x, Z_{x'}\}$. The construction of $\mathcal{H}_{\mathbf{X}_*}$ adds nodes in $\mathbf{V}^\dagger = \mathbf{V}(\mathsf{An}(Y_\mathbf{x}, \mathbf{X}_*)) = \{W, Z, T, R, Y\}$, and connects edges among $\mathbf{V}^\dagger$ inherited from the causal diagram $\mathcal{G}$ (Fig. 7b).

The key distinction from previous graphical representations (e.g., AMWNs (Correa and Bareinboim, 2024) or counterfactual graphs (Shpitser and Pearl, 2007)) is that $\mathcal{H}_{\mathbf{X}_*}$ connects directed edges from each $X_i \in \mathbf{X}$ to $T \in \mathsf{Ch}(X_i)_\mathcal{G}$, thereby preserving the original causal relations among them, based on *consistency* $\mathbf{X}_* = \mathbf{x} \Rightarrow Y_\mathbf{x} = Y_{\mathbf{X}_*}$ (CTF-Rule 1). For instance, $W \to T$ and $Z \to R$ appear in $\mathcal{H}_{\mathbf{X}_*}$, implying $W_x \to T_{W_x}(= T_w)$ and $Z_{x'} \to R_{Z_{x'}}(= R_z)$, while $W_x \to T_w$ or $Z_{x'} \to R_z$ do *not* appear in those representations.

According to realizability (Prop. 1), whenever $T_\mathbf{z} \in \mathsf{An}(X_{i[\mathbf{w}_i]})$ and $T_\mathbf{s} \in \mathsf{An}(X_{j[\mathbf{w}_j]})$, it follows that $\mathbf{z} = \mathbf{s}$. Hence, the generating process of $Y_{\mathbf{X}_*}$ can be written in the form of a single SCM $\mathcal{M}_{\mathbf{X}_*} = \langle \mathbf{U}, \mathbf{V}, \mathcal{F}_{\mathbf{X}_*}, P(\mathbf{U}) \rangle$ where each mechanism in $\mathcal{F}_{\mathbf{X}_*}$ is modified such that, for every $X_{i[\mathbf{w}_i]} \in \mathbf{X}_*$, the arguments $\mathbf{W}_i$ of its downstream variables in $\mathbf{V}^\dagger$ are fixed to $\mathbf{w}_i$. Therefore, this construction mirrors the way in which the value of $Y_{\mathbf{X}_*}$ is determined.

---

**Algorithm 1:** Algorithm enumerating all representative CTF-POMISs (**simple version**)

**Input:** Causal diagram $\mathcal{G}$; and a reward variable $Y$
**Output:** All representative CTF-POMISs with respect to $\langle \mathcal{G}, Y \rangle$

1 Set $\mathcal{H} = \mathcal{G}[\mathtt{An}(Y)_{\mathcal{G}}]$.
2 **for** each children choice $(\mathbf{X}_W)_{W \in \mathbf{V} \setminus \{Y\}} \in \bigtimes_{W \in \mathbf{V} \setminus \{Y\}} 2^{\mathtt{Ch}(W)_{\mathcal{H}}}$ **do**
3      Generate a map $\mathtt{pa}[X] = \{W \mid X \in \mathbf{X}_W\} (\subseteq \mathtt{Pa}(X)_{\mathcal{H}})$ for all $X \in \mathbf{V}(\mathcal{H})$.
4      **if** $\mathsf{IB}(\mathcal{H}', Y) = \emptyset$ where $\mathcal{H}' = \langle \mathbf{V}(\mathcal{H}), \mathbf{E}(\mathcal{H}) \setminus \{W \to X \mid \forall X \in \mathbf{V}(\mathcal{H}), \forall W \in \mathtt{pa}[X]\} \rangle$
     **then**
5          Let $\mathbf{X}_* := \{X_{\mathbf{z}} \mid X \in \bigcup_{W \in \mathbf{W}} \mathbf{X}_W\}$ where $\mathbf{z} = \bigcup_{T \in \mathtt{An}(X)_{\mathcal{H}_{\overline{\mathtt{pa}[X]}}}} \mathtt{pa}[T]$.
6          Compute $\mathbf{X}'_* = \text{CTF-MISIFY}(\mathcal{G}, \mathbf{X}_*, Y)$.
7          **yield** $\mathbf{X}'_*$ **if** $\mathtt{An}(Y_{\mathbf{x}}, \mathbf{X}'_*)$ satisfies Prop. 1.

8 **function** CTF-MISIFY$(\mathcal{G}, \mathbf{X}_*, Y)$
9      Set $\mathbf{X} = \mathtt{An}(Y)_{\mathcal{G}_{\overline{\mathbf{V}(\mathbf{X}_*)}}}$.
10      **return** $\{X_{j[\mathbf{w}_j]} \in \mathbf{X}_* \mid X_j \in \mathbf{X} \ and \ \mathbf{W}_j \cap \mathtt{An}(X_j)_{\mathcal{G}_{\overline{\mathbf{X} \setminus \{X_j\}}}}\}$.

---

**Corollary 3.** *Given a* CTF-*MIS* $\mathbf{X}_*$*, the counterfactual regime graph* $\mathcal{H}_{\mathbf{X}_*}$ *is a subgraph of the causal diagram compatible with* $\mathcal{M}_{\mathbf{X}_*}$ *over* $\mathbf{V}^\dagger$.

For example, consider a CTF-MIS $\mathbf{X}_* = \{T_x\}$ shown in Fig. 7c. Under an action for $\mathbf{X}_*$, the original mechanism $f_W(X, \mathbf{U}_W) \in \mathcal{F}$ is replaced by $f_W(x, \mathbf{U}_W) = f'_W(\mathbf{U}_W)$ in $\mathcal{M}_{\mathbf{X}_*}$. This implies that it is no longer a function of $X$. Similarly, the CTF-MIS $\mathbf{X}_* = \{R_x\}$ in Fig. 7d operates analogously—the original mechanism $f_Z(X, \mathbf{U}_Z) \in \mathcal{F}$ is replaced by $f_Z(x, \mathbf{U}_Z) = f'_Z(\mathbf{U}_Z)$ in $\mathcal{M}_{\mathbf{X}_*}$. We are now ready to characterize CTF-POMIS using the rich graphical structure.

**Theorem 2** (Graphical characterization of CTF-POMIS). *A* CTF-*MIS* $\mathbf{X}_*$ *with respect to* $\langle \mathcal{G}, Y \rangle$ *is a* CTF-*POMIS if and only if* $\mathsf{IB}(\mathcal{H}_{\mathbf{X}_*}, Y) = \emptyset$ *holds.*

To witness, revisit the CTF-MIS $\mathbf{X}_* = \{W_x, Z_{x'}\}$ and its counterfactual regime graph in Fig. 7b. We begin by constructing a minimal UC-territory $\mathbf{T}$ in $\mathcal{H}_{\mathbf{X}_*}$. By construction, $\mathcal{H}_{\mathbf{X}_*} = \mathcal{H}_{\mathbf{X}_*}[\mathtt{An}(Y)_{\mathcal{H}_{\mathbf{X}_*}}]$ always holds. Starting from $\mathbf{T} = \{Y\}$, we observe $\mathtt{cc}(Y)_{\mathcal{H}_{\mathbf{X}_*}} = \{Y, R, Z, W\}$; thus, we update $\mathbf{T}$ to $\{Y, R, Z, W\}$. Including all their descendants yields $\mathbf{T} = \{Y, R, Z, W, T\}$. Since there are no further unobserved confounders between $\mathbf{T}$ and $\mathtt{An}(Y)_{\mathcal{H}_{\mathbf{X}_*}} \setminus \mathbf{T}$, we obtain $\mathsf{MUCT}(\mathcal{H}_{\mathbf{X}_*}, Y) = \{Y, R, Z, W, T\}$ along with $\mathsf{IB}(\mathcal{H}_{\mathbf{X}_*}, Y) = \emptyset$. Therefore, $\mathbf{X}_*$ qualifies as a CTF-POMIS. Likewise, the CTF-MISs in Figs. 7c and 7d can also be verified as CTF-POMISs. In contrast, consider the CTF-MIS $\mathbf{X}_* = \{R_z\}$ in Fig. 7e, which is not a CTF-POMIS. In this example, $\mathsf{IB}(\mathcal{H}_{\mathbf{X}_*}, Y) = \{X\} \neq \emptyset$ entails that an intervention on $\mathbf{X}_* \cup \{W_x\}$ yields no worse and possibly better outcomes.

Initializing with $\mathbf{T} = \{Y\}$ gives $\mathtt{cc}(Y)_{\mathcal{H}_{\mathbf{X}_*}} = \{Y, R, W\}$, and adding their descendants results in $\mathbf{T} = \{W, T, R, Y\}$. Since there are no additional unobserved confounders between $\mathbf{T}$ and $\mathtt{An}(Y)_{\mathcal{H}_{\mathbf{X}_*}} \setminus \mathbf{T}$, we obtain $\mathsf{IB}(\mathcal{H}_{\mathbf{X}_*}, Y) = \{X\}$, which violates the condition for being a CTF-POMIS. Indeed, $\mathsf{IB}(\mathcal{H}_{\mathbf{X}_*}, Y) = \{X\}$ implies

$$\mathbb{E}Y_{\mathbf{X}_*} = \sum_x \mathbb{E}[Y_{\mathbf{X}_*} \mid x]P(x) = \sum_x \mathbb{E}Y_{\mathbf{X}_*, W_x}P(x) \leq \mathbb{E}Y_{\{\mathbf{X}_*, W_x\}^*},$$

and thus $\mathbf{X}_*$ is *not* a CTF-POMIS with respect to $\langle \mathcal{G}, Y \rangle$.

### 3.2.2 ALGORITHMIC APPROACH: ENUMERATING REPRESENTATIVE CTF-POMISS

Identifying all realizable actions through Prop. 1 over arbitrary counterfactual events, and subsequently verifying whether each constitutes a CTF-MIS and a CTF-POMIS action is computationally prohibitive, which grows super-exponentially with the number of nodes. To circumvent this, we first identify counterfactuals of the form $\{X_{i[\mathtt{Pa}(X_i)_{\mathcal{G}}]}\}_{i=1}$ that satisfy the CTF-POMIS conditions (Thm. 2). Notably, each such counterfactual can represent dozens of equivalent CTF-(PO)MISs, and collectively they cover $\mathcal{A}^\star$, which is verified by the following proposition.

**Proposition 3** (Existence of equivalent action). *For any* CTF-*(PO)MIS* $\mathbf{Z}_*$ *for* $\langle \mathcal{G}, Y \rangle$*, there exists an equivalent action* $\mathbf{X}_* = \{X_{i[\mathbf{w}_i]}\}_{i=1} \subseteq \mathtt{An}(\mathbf{Z}_*)$ *satisfying* $\mathbf{X} \subseteq \mathbf{W} \cup \mathtt{Ch}(\mathbf{W})_{\mathcal{G}}$ *where* $\mathbf{W} \triangleq \bigcup_i \mathbf{W}_i$.

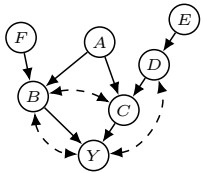

**(a) Task 1**      **(b) Task 2**      **(c) Task 3**

Figure 8: Cumulative regrets for the corresponding KL-UCB (solid) and TS (dashed) under distinct strategies. We plot the average cumulative regrets along with their standard deviations.

To see this, consider again Fig. 4a. We observed that $\mathbf{X}_* = \{W_x, Z_{x'}\} \subseteq \mathrm{An}(\mathbf{Z}_*) = \{T_x, W_x, Z_{x'}\}$ is equivalent to $\mathbf{Z}_*$, which satisfies $\mathbf{X} = \{W, Z\} \subseteq \{X, W, Z\} = \mathbf{W} \cup \mathrm{Ch}(\mathbf{W})_{\mathcal{G}}$. Recall that $\mathcal{H}_{\mathbf{X}_*} \equiv \mathcal{H}_{\mathbf{Z}_*}$, as they share the same effect propagation (Fig. 4b). Thus, it suffices to test whether $\mathbf{X}_*$ is a CTF-POMIS; other equivalent counterfactuals are no longer needed and can be safely discarded. We refer to such $\mathbf{X}_*$ as *representative* CTF-POMIS.

**Theorem 3.** *The Algorithm 1 returns all and only representative* CTF-*POMISs given* $\langle \mathcal{G}, Y \rangle$.

Building on this, Algo. 1 enumerates all representative CTF-POMISs via an edge-selection approach, which runs in $\mathcal{O}(n^2 \cdot 2^{|\mathbf{E}|})$. For concreteness, consider the causal diagram $\mathcal{G}$ in Fig. 3b where $\mathcal{G} = \mathcal{G}[\mathrm{An}(Y)_{\mathcal{G}}]$ (Line 1). With the children choices $\mathrm{pa}[Z] = \{T\}$ and $\mathrm{pa}[X] = \{W\}$ among each power set over children, let $\mathcal{H}'$ be the subgraph obtained by removing the selected edges $T \to Z$ and $W \to X$ from $\mathcal{G}$. In Lines 4–5, since $\mathrm{IB}(\mathcal{H}', Y) = \emptyset$, we construct $\mathbf{X}_* = \{Z_{tw}, X_w\}$ from $\mathrm{pa}[Z] = \{T\}$ and $\mathrm{pa}[X] = \{W\}$ where the subscript $w$ in $Z_{tw}$ is induced by the realizability condition, i.e., $W \in \mathrm{An}(Z)_{\mathcal{G}_{\overline{T}}}$, and $\mathcal{H}' \equiv \mathcal{H}_{\mathbf{X}_*}$ holds (further theoretical explanation provided in Appendices E.2 and E.3).

In Line 6, the algorithm calls a subroutine CTF-MISIFY, which modifies the counterfactual so that it satisfies the two conditions of CTF-MIS (Thm. 1) as follows: given $\langle \mathcal{G}, Y \rangle$ with a counterfactual $\mathbf{X}_* = \{X_{i[\mathbf{w}_i]}\}_{i=1}$, (i) remove $X_{j[\mathbf{w}_j]} \in \mathbf{X}_*$ if $X_j \notin \mathrm{An}(Y)_{\mathcal{G}_{\overline{\mathbf{X}}}}$; and (ii) remove if $\mathbf{W}_j \cap \mathrm{An}(X_j)_{\mathcal{G}_{\overline{\mathbf{X} \setminus \{X_j\}}}} = \emptyset$. Since $\mathbf{X}_* = \{Z_{tw}, X_w\}$ satisfies the two conditions, it follows that $\mathbf{X}'_* = \mathbf{X}_*$. Then, since there is no conflict in $\mathrm{An}(Y_{\mathbf{x}}, \mathbf{X}'_*) = \{Y_{zx}, Z_{tw}, X_w\}$, $\mathbf{X}'_*$ constitutes a valid form by Prop. 1, and is therefore returned. More detail is presented as Algo. 2 in Appendix E.3.

## 4 EXPERIMENTS

We evaluate the cumulative regret (CR) of CTF-SCB under different strategies to assess the effect of employing representative CTF-POMIS. Hereafter, we omit the term representative for brevity. The number of trials is set to 10k for the first two tasks and 100k for the last one, which is sufficient to observe performance differences. Each simulation is repeated 1,000 times to ensure consistency of results. We compare three arm-selection strategies—CTF-POMISs (pink in Fig. 8), CTF-MISs (purple), and POMISs (green)—each combined with two prominent solvers: Thompson Sampling (TS) and KL-UCB[6]. Details of the bandit mechanisms and settings can be found in Appendix C.

**Task 1.** We compared CTF-MIS and CTF-POMIS for the causal diagram in Fig. 5a. The CTF-POMIS based TS and KL-UCB achieve CRs of 66.69 and 148.19, which correspond to $\frac{\text{CR for CTF-POMIS}}{\text{CR for CTF-MIS}} \approx \mathbf{38.92}\%$ and $\mathbf{38.96}\%$, respectively. Since the number of arms for POMIS is smaller than $\mathcal{A}^\star$, POMIS may suffer less from exploration and thus temporarily achieve a smaller CR than CTF-(PO)MIS; nevertheless, CTF-(PO)MIS ultimately prevails, depicted in Fig. 8a.

Figure 9

**Task 2.** We consider the causal diagram in Fig. 3b. Using the three strategies, the CTF-POMIS based TS and KL-UCB achieve CRs of 387.01 and 831.24, which correspond to $\mathbf{57.46}\%$ and $\mathbf{70.17}\%$, respectively, of CR for CTF-MIS. As shown in Fig. 8b, CTF-(PO)MIS consistently outperformed POMIS.

---

[6]We apply standard bandit solvers to $\mathcal{A}^\star$ which enjoy well-established finite-time regret guarantees of order $\mathcal{O}(\sum_{\mathbf{X}_* \in \mathcal{A}^\star : \Delta_{\mathbf{X}_*} > 0} \frac{\log T}{\Delta_{\mathbf{X}_*}})$ (Lattimore and Szepesvári, 2020).

**Task 3.** We evaluate CRs with the causal diagram in Fig. 9 in more involved scenarios where (i) the optimal action is not contained in POMIS, as in the previous tasks (Fig. 8c, left), and (ii) the optimal action lies in $\mathcal{L}_{\leq 2}$ (right). In the first case, the CR for POMIS suffers linear regret due to the strictly positive gap $\Delta_{\mathcal{L}_{\leq 2}} \triangleq \mu_{\mathbf{X}_*^\star} - \mu_{\mathbf{x}^\star} > 0$. When $\mathbf{X}_*^\star$ lies in $\mathcal{L}_{\leq 2}$ (i.e., $\Delta_{\mathcal{L}_{\leq 2}} = 0$)—a special case that does not undermine our theoretical results, since the deployed agent can never be certain prior to interaction whether the optimal arm lies in $\mathcal{L}_{\leq 2}$—the smaller action space allows POMIS to converge faster than the others. All CRs and the numbers of sets and arms are provided in Tables 1 and 2 in Appendix C.

## 5 LIMITATIONS

**Modeling bandit instances in the form of SCMs.** SCMs are a versatile and expressive framework that provides a principled way to represent and reason about causal relationships. Their generality makes them applicable across a wide range of domains. However, SCMs come with certain limitations, such as the assumption of a well-defined set of variables and a fixed causal structure, which may not adequately capture the complexity of dynamic, high-dimensional, or partially observed systems. Nonetheless, our work addresses a fundamental problem within the SCM framework. We believe it provides a solid foundation for future research, such as extending causal bandits to more complex or less structured environments.

**Ability to perform counterfactual actions.** A key limitation of our work lies in the assumption that the deployed agent can execute any action corresponding to realizable counterfactual distributions $P(Y_{\mathbf{X}_*})$. This assumption implicitly requires the identification of suitable counterfactual mediators. While our results provide a rigorous theoretical foundation, the practicality of this assumption is limited, as real-world environments may constrain both the realizability of such operations and the construction of appropriate counterfactual mediators due to ethical, safety, or technical considerations.

**Known causal diagram.** We make the standard assumption that the deployment learner has access to the underlying causal diagram. While knowledge of the causal structure can greatly enhance decision-making, this requirement may limit the broader applicability of the proposed approach. In practice, though several techniques—such as causal discovery methods or the use of ancestral graphs as plausible explanations—can help alleviate this issue, these techniques typically require substantial domain knowledge or precise conditional independence (CI) statements, and thus, the assumption remains a key limitation of our framework. However, some degree of misspecification can be tolerated without invalidating the performance guarantees—particularly when the assumed causal diagram forms a *super-model* of the true environment. Suppose that the true environment is compatible with the causal diagram $\mathcal{G} = \langle \{X, Y\}, \{X \to Y\} \rangle$. If we are unsure about the presence of an unobserved confounder, we can conservatively posit a super-model $\mathcal{G}' = \langle \{X, Y\}, \{X \to Y, X \leftrightarrow Y\} \rangle$. Then, a collection of CTF-POMIS under $\mathcal{G}$ is $\{\{X_x\}\}$, while under $\mathcal{G}'$ it becomes $\{\emptyset, \{X_x\}\}$, which covers the true CTF-POMIS. This leads to less informative but still correct inferences, outperforming methods that ignore structural information altogether. In contrast, misspecifying in the opposite direction can lead to incorrect inferences. This reflects a fundamental asymmetry: being conservative preserves soundness, but missing edges can violate correctness.

## 6 CONCLUSION

In this work, we have extended the structural causal bandit (SCB) framework by incorporating realizable counterfactual regimes into the agent's action space. We introduced the notions of counterfactual minimal intervention sets (CTF-MIS) with its conditions (Thm. 1), and possibly-optimal minimal intervention sets (CTF-POMIS) to systematically prune suboptimal actions. By leveraging counterfactual regime graphs—special graphical representations that preserve consistent causal relations—we developed a characterization of these sets (Thm. 2). Taken together, we developed an efficient algorithm (Algo. 1) which completely enumerates all non-redundant CTF-POMISs relative to the reward in an edge-selection manner (Thm. 3). We believe this work opens a new research avenue at the intersection of causal inference, decision theory, and online learning, inviting further exploration of counterfactual causal bandit algorithms in both theoretical and practical domains.

ACKNOWLEDGMENTS

We thank anonymous reviewers for constructive comments to improve the manuscript. This work was partly supported by the IITP (RS-2022-II220953/30%) and NRF (RS-2023-00211904/70%) grant funded by the Korean government.

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

CONTENTS

# COUNTERFACTUAL STRUCTURAL CAUSAL BANDITS
## (APPENDIX)

## A  RELATED WORKS

**Causal decision making.**    Integrating causal knowledge into a decision-making process enables an agent to model decision problems with abundant dependency structures (Zhang and Bareinboim, 2019; 2020; Zhang et al., 2020; Kumor et al., 2021; Zhang and Bareinboim, 2022; Ruan et al., 2024; Zhang and Bareinboim, 2024; Richens and Everitt, 2024; Richens et al., 2025), where structural causal models (SCMs) (Pearl, 2000) have been employed to represent causal relationships among actions, rewards, and other relevant factors such as context and states. This approach allows agents to make informed decisions by explicitly considering the causal pathways through which each action affects the reward (Mueller and Pearl, 2023; Bareinboim et al., 2024).

The integration of causal inference into *multi-armed bandit* (MAB) framework (Robbins, 1952; Lai and Robbins, 1985; Lattimore and Szepesvári, 2020) has opened new avenues for modeling and solving decision problems with richer dependency structures. Existing studies (Bareinboim et al., 2015; Lattimore et al., 2016; Forney et al., 2017) have shown that causality-aware strategies can significantly outperform MAB algorithms that do not account for such underlying causal relationships. Subsequent work has explored various specialized settings by introducing additional structural assumptions, such as the availability of both observational and experimental distributions, or linear mechanisms (Zhang and Bareinboim, 2017; Lu et al., 2020; Bilodeau et al., 2022; De Kroon et al., 2022; Feng and Chen, 2023; Varici et al., 2023).

Lu et al. (2021) were the first to study causal bandits without assuming access to the full causal diagram. Their approach targets an *atomic* setting in which a reward variable has only a single parent, reducing the problem to identifying that parent for optimal intervention. They further assume that the agent observes instead the skeleton of the true causal diagram. Extending this line of work, Konobeev et al. (2025) eliminated the need for prior knowledge of the graph skeleton. However, their setting remains restricted to the same atomic case. More recently, Feng et al. (2025) considered causal bandits in which each action corresponds to an intervention on a set of variables. Yan and Tajer (2024) considered actions as *soft* interventions on variables, i.e., changing the conditional distribution $P(v \mid \mathbf{pa}_V)$ to $Q(v \mid \mathbf{pa}_V)$. Despite this generalization, all these approaches assumed *causal sufficiency* and thus do not account for the presence of latent variables. Malek et al. (2023) provided some results for settings with unknown graph structures, the authors initially highlight the challenge posed by the exponentially large number of arms in causal bandit problems under unknown graphs, and assumed that no confounding exists between the reward variable and its ancestors.

**Structural causal bandits.**    Lee and Bareinboim (2018) formalized *structural causal bandit* (SCB) framework, in which a bandit instance is structured by an SCM, and each action corresponds to an intervention on a subset of variables. They proposed a sound and complete graphical characterization to identify *minimal intervention sets* (MISs) and *possibly-optimal minimal intervention sets* (POMISs), where the former includes only the variables that affect the reward, and the latter refers to actions that could be part of an optimal strategy among MISs, thereby guiding the agent to avoid unnecessary exploration, without any actual interaction. Lee and Bareinboim (2019) extended this approach to accommodate scenarios involving non-manipulable variables among all the variables in the graph.

Lee and Bareinboim (2020) established the framework under stochastic policies and demonstrated the informativeness of such policies. Everitt et al. (2021) and Carey et al. (2024) further investigated the completeness of the graphical characterization of optimal policy spaces, although the general completeness remains an open problem. Wei et al. (2023) proposed a parameterization-based approach to incorporate shared information among possibly-optimal actions. Elahi et al. (2024) extended the SCB to settings where no causal diagram is assumed to be accessible, requiring their algorithm to

perform causal discovery—i.e., constructing the causal structure—during online interaction. Building on this line of work, causal Bayesian optimization (CBO; Aglietti et al. (2020)) leveraged the systematic characterization of MIS and POMIS for structural pruning in continuous action spaces, and Bhatija et al. (2025) extended it to a multi-outcome variant incorporating Pareto optimality. Park et al. (2025) investigated SCB in settings where the available information does not constitute a full partial ancestral graph (Richardson and Spirtes, 2002; Zhang, 2008) representing the Markov equivalence class of the true causal diagram, and Park and Lee (2025) focused on the transportability (Pearl and Bareinboim, 2011; Bareinboim and Pearl, 2012; 2016) of SCB.

**Counterfactual bandits.** Counterfactual reasoning plays a vital role in decision-making, constructing explanations for decisions in applications (Mueller and Pearl, 2024; Li and Pearl, 2019), analyzing a causal effect into direct and indirect pathways (Pearl, 2001; Rubin, 2004).

Beyond factual actions, Bareinboim et al. (2015) and Forney et al. (2017) proposed MAB strategies aiming to optimize $\mathbb{E}[Y_x \mid x']$, which go beyond interventional studies by incorporating the agent's natural *intention*—the agent's initially intended arm choice at each round prior to the final choice—as part of the decision-making evidence. Building on this idea, Raghavan and Bareinboim (2025) proposed a bandit strategy that leverages additional evidence in the form of $\mathbb{E}[Y_x \mid x', z_{x''}]$. The established studies focused on extracting evidence about agents' intentions and using this information as *context*, which constitutes the most distinctive aspect of our work.

**Path-specific effects.** Path-specific effects (Pearl, 2001) constitute a broad class of causal effects that measure the influence of a treatment on an outcome through specific causal pathways. For example, Total Effect (TE) captures the influence transmitted through *all* causal paths connecting the treatment and the outcome (Pearl, 2000). In the presence of mediators, contemporary research on path-specific analysis (Zhang and Bareinboim, 2018; Chiappa, 2019; Miles et al., 2020; Plecko and Bareinboim, 2022; Farbmacher et al., 2022; Zhang et al., 2025) has broadened its scope to investigate effects transmitted along particular paths using counterfactuals, such as Natural Direct Effect (NDE) and Natural Indirect Effect (NIE) (Pearl, 2001; Imai et al., 2010; 2011; VanderWeele and Knol, 2014).

While our work is related to the original idea of path-specific effects—intervening on *perception* of certain nodes, also known as *edge intervention* (Shpitser and Tchetgen, 2016; Pearl et al., 2016; Robins et al., 2022)—our focus is fundamentally different. Rather than disentangling which specific paths contribute to the rewards and to what extent, we aim to develop optimized strategies for maximizing or minimizing regret in its entirety.

**Graphical representations for counterfactual inference.** Balke and Pearl (1994) introduced a graphical construction that represents two "worlds" within a single graph, known as *twin network*. Avin et al. (2005) later presented *parallel networks*, generalization of the twin networks to multi-world settings. However, d-separation criterion (Pearl, 1995) is sound but *not* complete for parallel networks, as they include more variables with deterministic relationships among them. Shpitser and Pearl (2007) proposed an algorithm for merging nodes in a parallel network under specific variable instantiations, resulting in what is referred to as the *counterfactual graph*, which is conjectured to be complete for determining d-separations between counterfactual events. For *Single World Intervention Graphs* (SWIGs) (Richardson and Robins, 2013; Shpitser et al., 2022), conditional independence among variables can be read using d-separation. However, SCMs imply many cross-world constraints on the counterfactual joint distribution that cannot be captured by a single-intervention representation. Recently, Correa and Bareinboim (2024; 2025) introduced a novel graphical representation, called *Ancestral Multi-World Networks* (AMWNs), which provide sound and complete d-separation statements, and allow for polynomial-time reasoning with respect to the number of different worlds involved in the counterfactual query.

# B    BACKGROUNDS

An SCM induces observational, interventional and counterfactual quantities over the endogenous variables, which form three layers known as Pearl Causal Hierarchy (PCH).

**Definition 7** (Pearl Causal Hierarchy (Bareinboim et al., 2022)). *An SCM $\mathcal{M} = \langle \mathbf{U}, \mathbf{V}, \mathcal{F}, P(\mathbf{U}) \rangle$ induces three layers of probability distributions which form the Pearl Causal Hierarchy. For any $\mathbf{Y}, \mathbf{Z}, \cdots, \mathbf{X}, \mathbf{W} \subseteq \mathbf{V}$, the three layers of distributions are given by:*

- *(Observational):*

$$P^{\mathcal{M}}(\mathbf{y}) = \sum_{\mathbf{u}} \mathbb{I}\{\mathbf{Y}(\mathbf{u}) = \mathbf{y}\} P(\mathbf{u}). \tag{3}$$

- *(Interventional):*

$$P^{\mathcal{M}}(\mathbf{y_x}) = \sum_{\mathbf{u}} \mathbb{I}\{\mathbf{Y_x}(\mathbf{u}) = \mathbf{y}\} P(\mathbf{u}). \tag{4}$$

- *(Counterfactual):*

$$P^{\mathcal{M}}(\mathbf{y_x}, \cdots, \mathbf{z_w}) = \sum_{\mathbf{u}} \mathbb{I}\{\mathbf{Y_x}(\mathbf{u}) = \mathbf{y}, \cdots, \mathbf{Z_w}(\mathbf{u}) = \mathbf{z}\} P(\mathbf{u}). \tag{5}$$

The collection of all $\mathcal{L}_1$ (observational) is denoted as $\mathbf{P}^{\mathcal{L}_1}$, the collection of all $\mathcal{L}_2$ (interventional) is denoted as $\mathbf{P}^{\mathcal{L}_2}$, and the collection of all $\mathcal{L}_3$ (counterfactual) is denoted as $\mathbf{P}^{\mathcal{L}_3}$.

Multiple interventions entail different copies of the mechanisms of the SCM, each for a different world (syntactically represented by a different subscript), but all sharing the same $P(\mathbf{U})$. A counterfactual distribution can be evaluated by passing the set of exogenous variables $\mathbf{U}$ through the different versions of those mechanisms, depending on which hypothetical world one aims to evaluate.

Given an SCM $\mathcal{M}$, a graph can be constructed to capture topological information among endogenous and exogenous variables, called *causal diagram* (Pearl, 1995).

**Definition 8** (Causal diagram; Definition 13 in Bareinboim et al. (2022)). *Consider an SCM $\mathcal{M} = \langle \mathbf{U}, \mathbf{V}, \mathcal{F}, P(\mathbf{U}) \rangle$. Then $\mathcal{G}$ is said to be a causal diagram of $\mathcal{M}$ if constructed as follows:*

*(i) Add a vertex for every endogenous variable in the set $\mathbf{V}$.*

*(ii) Add an edge $(V \rightarrow W)$ for every $V, W \in \mathbf{V}$ if $V$ appears as an argument of $f_W \in \mathcal{F}$.*

*(iii) Add a bidirected edge $(V \leftrightarrow W)$ for every $V, W \in \mathbf{V}$ if the corresponding $\mathbf{U}_V, \mathbf{U}_W \subset \mathbf{U}$ are correlated or the corresponding functions $f_V, f_W$ share some $U \in \mathbf{U}$ as an argument.*

The pairing of a causal diagram with the set of invariance constraints it encodes over a collection of distributions defines a graphical model (Yang and Bareinboim, 2025).

**Definition 9** (Counterfactual Bayesian Network (Correa and Bareinboim, 2024)). *A graph $\mathcal{G}$ is a Counterfactual Bayesian Network (CTFBN) for a collection of counterfactual distributions $\mathbf{P}^{\mathcal{L}_3}$ if:*

*(i) (Independence Restrictions) Let $\mathbf{W}_*$ be a set of counterfactuals of the form $W_{\mathbf{pa}_W}$, and $\mathbf{C}_1, \cdots, \mathbf{C}_l$ the c-components of $\mathcal{G}[\mathbf{V}(\mathbf{W}_*)]$, and $\mathbf{C}_{1*}, \cdots, \mathbf{C}_{l*}$ the corresponding partition over $\mathbf{W}_*$. Then $P(\mathbf{W}_*)$ factorizes as*

$$P(\bigwedge_{W_{\mathbf{pa}_W} \in \mathbf{W}_*} W_{\mathbf{pa}_W}) = \prod_{j=1}^{l} P(\bigwedge_{W_{\mathbf{pa}_W} \in \mathbf{C}_{j*}} W_{\mathbf{pa}_W}) \tag{6}$$

*(ii) (Exclusion Restrictions) For every variable $Y \in \mathbf{V}$ with parents $\mathbf{Pa}_Y$, for every set $\mathbf{Z} \subseteq \mathbf{V} \setminus (\mathbf{Pa}_Y \cup \{Y\})$ and any counterfactual set $\mathbf{W}_*$, we have*

$$P(Y_{\mathbf{pa}_Y, \mathbf{z}}, \mathbf{W}_*) = P(Y_{\mathbf{pa}_Y}, \mathbf{W}_*) \tag{7}$$

*(iii) (Local Consistency) For every variable $Y \in \mathbf{V}$ with parents $\mathbf{Pa}_Y$, let $\mathbf{X} \subseteq \mathbf{Pa}_Y$, then for every set $\mathbf{Z} \subseteq \mathbf{V} \setminus (\mathbf{X} \cup \{Y\})$ and any counterfactual set $\mathbf{W}_*$, we have*

$$P(Y_{\mathbf{z}} = y, \mathbf{X}_{\mathbf{z}} = \mathbf{x}, \mathbf{W}_*) = P(Y_{\mathbf{xz}} = y, \mathbf{X}_{\mathbf{z}} = \mathbf{x}, \mathbf{W}_*) \tag{8}$$

**D-separation.** In a causal diagram $\mathcal{G}$, a path $p$ between vertices $X$ and $Y$ is a *d-connecting* path relative to a set $\mathbf{Z}$ if (i) every non-collider on $p$ is not a member of $\mathbf{Z}$; and (ii) every collider on $p$ is an ancestor of some member of $\mathbf{Z}$. Two variables $X$ and $Y$ are said to be *d-separated* by $\mathbf{Z}$ if there

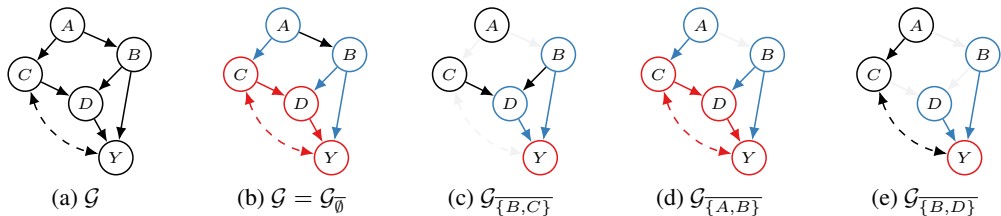

Figure 10: MUCT and IB are shown in red and blue, respectively; (b, c) non-POMISs; (d, e) POMISs.

is no d-connecting path between $X$ and $Y$ relative to $\mathbf{Z}$. Two disjoint sets $\mathbf{X}$ and $\mathbf{Y}$ are said to be d-separated by $\mathbf{Z}$ if every variable in $\mathbf{X}$ is d-separated from every variable in $\mathbf{Y}$ by $\mathbf{Z}$ and denoted as $(\mathbf{X} \perp\!\!\!\perp \mathbf{Y} \mid \mathbf{Z})_{\mathcal{G}}$.

**Do-calculus.** Pearl (1995) devised *do-calculus* which acts as a bridge between observational and interventional distributions from a causal diagram without relying on any parametric assumptions.

**Theorem 4** (Do-calculus (Pearl, 1995)). *Let $\mathcal{G}$ be a causal diagram compatible with a structural causal model $\mathcal{M}$, with endogenous variables $\mathbf{V}$. For any disjoint $\mathbf{X}$, $\mathbf{Y}$, $\mathbf{W}$, $\mathbf{Z} \subseteq \mathbf{V}$, the following rules are valid.*

**Rule 1.** $P(\mathbf{y} \mid do(\mathbf{w}), \mathbf{x}, \mathbf{z}) = P(\mathbf{y} \mid do(\mathbf{w}), \mathbf{z})$      *if $\mathbf{X}$ and $\mathbf{Y}$ are d-separated by $\mathbf{W} \cup \mathbf{Z}$ in $\mathcal{G}_{\overline{\mathbf{W}}}$*

**Rule 2.** $P(\mathbf{y} \mid do(\mathbf{w}), do(\mathbf{x}), \mathbf{z}) = P(\mathbf{y} \mid do(\mathbf{w}), \mathbf{x}, \mathbf{z})$    *if $\mathbf{X}$ and $\mathbf{Y}$ are d-separated by $\mathbf{W} \cup \mathbf{Z}$ in $\mathcal{G}_{\overline{\mathbf{W}}, \underline{\mathbf{X}}}$*

**Rule 3.** $P(\mathbf{y} \mid do(\mathbf{w}), do(\mathbf{x}), \mathbf{z}) = P(\mathbf{y} \mid do(\mathbf{w}), \mathbf{z})$    *if $\mathbf{X}$ and $\mathbf{Y}$ are d-separated by $\mathbf{W} \cup \mathbf{Z}$ in $\mathcal{G}_{\overline{\mathbf{W}}, \overline{\mathbf{X}(\mathbf{Z})}}$*
*where $\mathbf{X}(\mathbf{Z}) \triangleq \mathbf{X} \setminus \mathtt{An}(\mathbf{Z})_{\mathcal{G}[\mathbf{V} \setminus \mathbf{W}]}$.*

## B.1 STRUCTURAL CAUSAL BANDITS

In this section, we provide definitions of minimal intervention set (MIS) and possibly-optimal minimal intervention set (POMIS) (Lee and Bareinboim, 2018), along with their complete graphical characterizations and illustrative examples.

**Definition 10** (Minimal intervention set (MIS)). *Given information $\langle \mathcal{G}, Y \rangle$, a set of variables $\mathbf{X} \subseteq \mathbf{V} \setminus \{Y\}$ is called a minimal intervention set (MIS) if there is no $\mathbf{X}' \subsetneq \mathbf{X}$ such that $\mu_{\mathbf{x}[\mathbf{X}']} = \mu_{\mathbf{x}}$ for every SCM conforming to $\mathcal{G}$.*

MIS leverages Rule 3 of do-calculus (Pearl, 1995) to eliminate variables that are irrelevant to the reward. Intuitively, an MIS can be understood as a set $\mathbf{X}$ in which there exists a directed path from any variable $X \in \mathbf{X}$ to $Y$, ensuring that each $X$ can influence $Y$.

**Proposition 4** (Characterization for MIS; Proposition 1 in Lee and Bareinboim (2018)). *Let $\mathcal{G}$ be a causal diagram over the set of variables $\mathbf{V}$. A set $\mathbf{X} \subseteq \mathbf{V} \setminus \{Y\}$ is an MIS relative to $\langle \mathcal{G}, Y \rangle$ if and only if $\mathbf{X} \subseteq \mathtt{An}(Y)_{\mathcal{G}_{\overline{\mathbf{X}}}}$.*

**Definition 11** (Possibly-optimal minimal intervention set (POMIS)). *Let $\mathbf{X} \subseteq \mathbf{V} \setminus \{Y\}$ be a MIS relative to $\langle \mathcal{G}, Y \rangle$. If there exists an SCM conforming to $\mathcal{G}$ such that $\mu_{\mathbf{x}^*} > \forall_{\mathbf{W} \in \mathrm{MIS}_{\mathcal{G}, Y} \setminus \{\mathbf{X}\}} \mu_{\mathbf{w}^*}$, then $\mathbf{X}$ is a possibly-optimal minimal intervention set (POMIS).*

When given a causal diagram $\mathcal{G}$, minimal UC-territory (MUCT; Def. 5) and interventional border (IB; Def. 6) provide a graphical characterization of POMIS. Intuitively, MUCT is the smallest set of variables that constitute a closed mechanism conveying all hidden information from unobserved confounders to the downstream reward, while the IB consists of the nodes that directly affect this closed mechanism. This implies that intervening on any variable within the territory may disrupt essential information, while intervening on all variables in IB with values that exert a positive effect can be beneficial.

**Theorem 5** (Characterization for POMIS; Theorem 6 in Lee and Bareinboim (2018)). *Given information $\langle \mathcal{G}, Y \rangle$, a set $\mathbf{X} \subseteq \mathbf{V} \setminus \{Y\}$ is a POMIS with respect to $\langle \mathcal{G}, Y \rangle$ if and only if $\mathsf{IB}(\mathcal{G}_{\overline{\mathbf{X}}}, Y) = \mathbf{X}$.*

For example, consider the causal diagram in Fig. 10a. Here, $\mathcal{G} = \mathcal{G}[\text{An}(Y)_{\mathcal{G}}]$ holds. An $\mathcal{L}_1$ action $do(\emptyset)$ is not a POMIS. To see this, we construct MUCT, initializing $\mathbf{T} = \{Y\}$, as follows: Since $Y$ has an unobserved confounder with $C$, we update $\mathbf{T} = \text{cc}(Y)_{\mathcal{G}} = \{C, Y\}$, and thereafter add all the descendants of $C$, obtaining $\mathbf{T} = \{C, D, Y\}$. Since there are no more unobserved confounders between $\mathbf{T}$ and $\text{An}(Y)_{\mathcal{G}} \setminus \mathbf{T}$, MUCT has been found and is given by $\text{MUCT}(\mathcal{G}, Y) = \{C, D, Y\}$ along with $\text{IB}(\mathcal{G}, Y) = \{A, B\}$ (Fig. 10b). According to the graphical characterization, we can conclude that $do(\emptyset)$ is not a POMIS with respect to $\langle \mathcal{G}, Y \rangle$. Similarly, $\{B, C\}$ is also not a POMIS, as $\text{IB}(\mathcal{G}_{\overline{\{B,C\}}}, Y) = \{B, D\}$, as depicted in Fig. 10c. In contrast, the regimes corresponding to Figs. 10d and 10e are POMISs, since they satisfy $\text{IB}(\mathcal{G}_{\overline{\mathbf{X}}}, Y) = \mathbf{X}$.

## B.2 COUNTERFACTUAL CALCULUS

Correa and Bareinboim (2024; 2025) introduced a novel calculus over probability quantities that may be defined at the counterfactual level ($\mathcal{L}_3$), called the CTF-*calculus*.

**Theorem 6** (Counterfactual calculus (CTF-calculus); Theorem 3.1 in Correa and Bareinboim (2024)). *Let $\mathcal{G}$ be a causal diagram, then for $\mathbf{Y}$, $\mathbf{X}$, $\mathbf{Z}$, $\mathbf{W}$, $\mathbf{T}$, $\mathbf{R} \subseteq \mathbf{V}$, the following rules hold for the probability distributions generated by any model compatible with $\mathcal{G}$:*

> ***Rule 1. (Consistency rule - Obs./intervention exchange)***
> $P(\mathbf{y}_{\mathbf{T}_*\mathbf{x}}, \mathbf{x}_{\mathbf{T}_*}, \mathbf{w}_*) = P(\mathbf{y}_{\mathbf{T}_*}, \mathbf{x}_{\mathbf{T}_*}, \mathbf{w}_*)$

> ***Rule 2. (Independence rule - Adding/removing counterfactual observations)***
> $P(\mathbf{y}_{\mathbf{r}} \mid \mathbf{x}_{\mathbf{t}}, \mathbf{w}_*) = P(\mathbf{y}_{\mathbf{r}}, \mathbf{w}_*)$
> *if* $(\mathbf{Y}_{\mathbf{r}} \perp\!\!\!\perp \mathbf{X}_{\mathbf{t}} \mid \mathbf{W}_*)_{\mathcal{G}_A}$

> ***Rule 3. (Exclusion Rule - Adding/removing interventions)***
> $P(\mathbf{y}_{\mathbf{xz}}, \mathbf{w}_*) = P(\mathbf{y}_{\mathbf{z}}, \mathbf{w}_*)$
> *if* $\mathbf{X} \cap \text{An}(\mathbf{Y}) = \emptyset$ *in* $\mathcal{G}_{\overline{\mathbf{Z}}}$

*where $\mathcal{G}_A$ is the AMWN $\mathcal{G}_A(\mathcal{G}, \mathbf{Y}_{\mathbf{r}} \cup \mathbf{X}_{\mathbf{t}} \cup \mathbf{W}_*)$.*

The independence rule requires the construction of another graphical object, known as the *Ancestral Multi-World Network* (AMWN), which serves to identify d-separation (Pearl, 1995) relations among counterfactual variables.

**Nested counterfactuals.** Any quantity of nested counterfactuals can be expressed in terms of unnested ones using CTF-Rule 1 (consistency), enabling their analysis through standard counterfactual semantics.

**Lemma 2** (Counterfactual Unnesting Theorem (CUT); Theorem 4 in Correa et al. (2021)). *Let $Y, X \in \mathbf{V}$, $\mathbf{T}, \mathbf{Z} \subseteq \mathbf{V}$, and let $\mathbf{z}$ be a set of values for $\mathbf{Z}$. Then, the nested counterfactual $P(Y_{\mathbf{T}_*, X_{\mathbf{z}}})$ can be written with one less level of nesting as:*

$$P(Y_{\mathbf{T}_*, X_{\mathbf{z}}} = y) = \sum_{x \in \mathcal{D}_X} P(Y_{\mathbf{T}_*, x} = y, X_{\mathbf{z}} = x). \tag{9}$$

*where $\mathbf{T}_*$ represent any combination of counterfactuals based on $\mathbf{T}$.*

To see this, consider the causal diagram $\mathcal{G}$ in Fig. 11 with three variables, where $X$ represents *level of exercise* ($x'$ for regular exercising, $x$ for none), $Z$ *cholesterol levels*, and $Y$ *cardiovascular disease*. An interesting question is how much exercise prevents the disease through a pathway other than cholesterol regulation, which can be expressed as $\mathbb{E}[Y_{x', Z_x} - Y_x]$. This quantity is also known as Natural Direct Effect (NDE) (Pearl, 2001). Although the second term $\mathbb{E}Y_x$ can be trivially identified, the

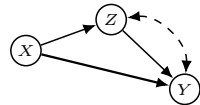

Figure 11: Non-identifiable Natural Direct Effect (NDE).

identifiability of the first term $\mathbb{E}Y_{x', Z_x}$ cannot be tested directly. According to CUT, the corresponding probability term $P(y_{x', Z_x})$ can be written as $P(y_{x', Z_x}) = \sum_z P(y_{x'z}, z_x)$, which implies that the identifiability of $P(y_{x'z}, z_x)$ entails the identifiability of $P(y_{x', Z_x})$.

Note that $P(y_{x'z}, z_x)$ is non-identifiable in this case; thus the NDE quantity cannot be estimated, even with arbitrary experimental data. For further details about CUT and the identification of counterfactual queries, we refer the reader to Correa et al. (2021); Correa and Bareinboim (2024; 2025).

## B.3 REALIZABILITY

Realizability is a property of distributions, indicating that an agent can draw samples from them through physical experimentation (e.g., RCT). We begin by introducing the most granular experimental capabilities, which serve as the fundamental building blocks of experimental procedures, including *counterfactual randomization* (Raghavan and Bareinboim, 2025).

**Definition 12** (Feasible operations). Given a causal diagram $\mathcal{G}$, the physical operations that an agent can perform on any unit $i$ are limited to:

(i) SELECT$^{(i)}$: randomly choosing, without replacement, a unit $i$ from the target population, to observe in the system;

(ii) READ$(V)^{(i)}$: measuring the way in which a causal mechanism $f_V \in \mathcal{F}$ has physically affected unit $i$, by observing its realized feature $V^{(i)}$;

(iii) RAND$(X)^{(i)}$: erasing and replacing $i$'s natural mechanism $f_X$ for a decision variable $X$ with an enforced value drawn from a randomizing device having support over $\mathcal{D}_X$;

(iv) CTF-RAND$(X \to \mathbf{C})^{(i)}$: This *counterfactual randomization* can be performed if there is a special *counterfactual mediator* by which the mechanisms generating $\mathbf{C}$ perceive the value of $X$. This counterfactual mediator then allows the agent to intervene on the value of $X$ as perceived by $\mathbf{C}$, thus *mimicking* an actual intervention on $X$.

For example, RAND$(X)^{(i)}$ denotes the standard Fisherian randomization of a decision variable $X$. By applying RAND$(X)^{(i)}$, the natural decision of unit $i$ is erased (i.e., its natural mechanism is broken). Consequently, performing READ$(Y)^{(i)}$ under this randomized regime allows researchers to estimate the causal effect of $X$ on $Y$.

**Definition 13** (Counterfactual mediator (informal definition); Definition 2.2 in Raghavan and Bareinboim (2025)). We call $\tilde{X}$ a counterfactual mediator of $X$ w.r.t $C \in \text{Ch}(X)_\mathcal{G}$ if the value of $X$ can be retrieved from $\tilde{X}$ by the mechanism generating $C$.

A counterfactual mediator $\tilde{X}$ can receive an exogenous noise, but the inverse from $\tilde{X} \mapsto X$ needs to be deterministic. This means that the domain of $\tilde{X}$ divides into equivalence classes of values that can each be mapped back to a unique $X$. This is quite intuitive and well aligns with real-life situations, for example, a race could be associated with many stereotypical names, but for each stereotypical name, it maps back to a unique race (Bertrand and Mullainathan, 2004). The formal definition and details can be found in Appendix D in Raghavan and Bareinboim (2025).

**Definition 14** (Realizability; Definition 3.4 in Raghavan and Bareinboim (2025)). Given a causal diagram $\mathcal{G}$ and a set of feasible operations $\mathbb{A}$, $\mathcal{L}_3$-distribution $P(\mathbf{W}_*)$ is *realizable* if there exists a sequence of operations from $\mathbb{A}$ by which an agent can draw an i.i.d. for any $\mathcal{M}$ consistent with $\mathcal{G}$.

**Definition 15** (Maximal feasible operation set (Raghavan and Bareinboim, 2025)). Given an SCM $\mathcal{M} = \langle \mathbf{U}, \mathbf{V}, \mathcal{F}, P(\mathbf{U}) \rangle$, the *maximal feasible operation set* $\mathbb{A}^\dagger$ is the set of *all* feasible operations an agent can perform in $\mathcal{M}$ with the most granular interventional capabilities: (i) SELECT$^{(i)}$; (ii) READ$(V)^{(i)}, \forall V \in \mathbf{V}$; (iii) RAND$(X)^{(i)}$; and (iv) $\forall X \in \mathbf{V}$: CTF-RAND$(X \to C)^{(i)}, \forall X \in V, \forall C \in \text{Ch}(X)$.

In short, we say $P(\mathbf{W}_*)$ or $\mathbf{W}_*$ is realizable if it is realizable from *maximal feasible operation set*. We now investigate the realizability of distributions in graphical language through a causal diagram.

**Lemma 3** (Lemma D.6 in Raghavan and Bareinboim (2025)). *Given a causal diagram $\mathcal{G}$, for any SCM $\mathcal{M}$ compatible with $\mathcal{G}$, the jointly necessary and sufficient conditions to measure $X_\mathbf{w}$ are (i) $\mathbf{W}$ is fixed as $\mathbf{w}$ (by intervention) as an input to all children $C \in \text{Ch}(\mathbf{W})_\mathcal{G} \cap \text{An}(X)_\mathcal{G}$; (ii) each $A \in \text{An}(X)_{\mathcal{G}_{\overline{\mathbf{w}}}}$ with $A \notin \mathbf{W} \cup \{X\}$ is received "naturally" (without intervention) by its children $C \in \text{Ch}(A)_\mathcal{G} \cap \text{An}(X)_\mathcal{G}$; and (iii) the mechanism $f_X$ is not erased and overwritten by an intervention.*

| | Total trials | Task 1 (Fig. 5a) 10k | Task 2 (Fig. 3b) 10k | Task 3 (Fig. 9) 100k | |
|---|---|---|---|---|---|
| TS | CTF-POMIS | **66.69** ±40.31 | **387.01** ±71.21 | **1642.1** ±237.4 | 2477.4 ±346.5 |
| | CTF-MIS | 171.37 ±49.0 | 673.5 ±82.0 | 5160.63 ±307.9 | 4825.6 ±201.8 |
| | POMIS | 664.27 ±39.5 | 1171.07 ±102.31 | 3495.4 ±216.1 | **1439.8** ±300.7 |
| KL-UCB | CTF-POMIS | **148.19** ±40.12 | **831.24** ±78.92 | **4226.0** ±279.7 | 4514.8 ±225.7 |
| | CTF-MIS | 380.29 ±47.93 | 1184.68 ±72.99 | 9563.71 ±248.5 | 5358.5 ±165.23 |
| | POMIS | 683.57 ±40.04 | 1397.15 ±70.2 | 5179.9 ±236.7 | **3135.43** ±244.1 |

Table 1: Mean and standard deviation of cumulative regret over 1,000 repeated simulations.

Furthermore, $\mathbf{X}_*$ can be evaluated (realizable) *if and only if* the three conditions (i–iii) in Lem. 3 are met for each $X_\mathbf{w} \in \mathbf{X}_*$ simultaneously.

**Lemma 4** (Corollary 3.7 in Raghavan and Bareinboim (2025)). *Given a causal diagram $\mathcal{G}$, an $\mathcal{L}_3$-distribution $Q = P(\mathbf{W}_*)$ is a realizable distribution induced by any SCM compatible with a given graph $\mathcal{G}$ if and only if the ancestor set $\mathtt{An}(\mathbf{W}_*)$ does not contain a pair of potential responses $W_\mathbf{s}, W_\mathbf{t}$ of the same variable $W$ under different regimes where $\mathbf{s} \neq \mathbf{t}$.*

Note the distinction between *identifiability* (Pearl, 1995; Tian and Pearl, 2003; Lee et al., 2019) and *realizability*. Identifiability refers to whether a distribution (e.g. $P(\mathbf{v_x})$) can be uniquely computed from a collection of available distributions, given a causal diagram $\mathcal{G}$; that is, whether the target quantity can be derivable for any SCM compatible with constraints encoded in $\mathcal{G}$ in a symbolic way. For counterfactual queries, identifiability refers to whether it can be computed from any combination of $\mathcal{L}_{\leq 2}$ distributions (Correa et al., 2021). In contrast, realizability concerns whether it is physically possible for an agent to gather data sampled from the target distribution.

For instance, recall the causal diagram in Fig. 11 where we have observed that $\mathbb{E}Y_{x'Z_x}$ is non-identifiable; i.e., $P(y_{x',Z_x}) = \sum_z P(y_{x',z}, z_x)$ cannot be identifiable from any combination of observational and experimental data. However, $\mathtt{An}(Y_{x'z}, Z_x) = \{Y_{x'z}, Z_x\}$ (to test Lem. 4) does not contain any conflicting subscripts for the same variable. Thus, $P(Y_{x'Z_x})$ is realizable. Therefore, one can directly sample from $P(Y_{x',Z_x})$ via feasible operations as defined in Def. 12. For more details on the realizability of counterfactual distributions, see Raghavan and Bareinboim (2025).

## C EXPERIMENTAL DETAILS

This section provides details on the specific SCMs used in all bandit instances presented in the experiments. Simulations are repeated 1,000 times to obtain consistent results. The simulations were conducted on a Linux server equipped with an Intel® Xeon® Gold 5317 processor running at 3.0 GHz and 64 GB of RAM. No GPUs were used during the simulations.

We consider three strategies for selecting arms: CTF-POMISs, CTF-MISs, and POMISs, combined with two prominent MAB solvers: Thompson Sampling (Thompson, 1933; Chapelle and Li, 2011; Agrawal and Goyal, 2012; Kaufmann et al., 2012) and KL-UCB (Garivier and Cappé, 2011; Cappé et al., 2013). The number of trials is set to 10k for Tasks 1 and 2, and 100k for Task 3, which is sufficient to observe performance differences among action spaces[7]. Table 1 summarizes our simulation results, and Table 2 provides the number of representative CTF-POMISs, CTF-MIS, and POMIS, along with the corresponding number of actions for the three tasks and examples in the main body.

We denote the exclusive-or operation by $\oplus$, and use Bern to represent a Bernoulli distribution. We first construct $\mathcal{M}_1$, corresponding to Fig. 5a, as an illustrative example. In contrast, we *randomly* generate structural functions $\mathcal{F}$ using binary logical operations ($\wedge, \vee, \oplus, \neg$), and the parameters of the exogenous variable distributions are also *randomly* selected for the subsequent SCMs.

---

[7]The number of trials is selected such that the cumulative regret with respect to CTF-POMIS stabilizes across 1000 repeated runs. Our experimental setup closely follows those of Lee and Bareinboim (2018) and Wei et al. (2023).

| | | Task 1 (Fig. 5a) | Task 2 (Fig. 3b) | Task 3 (Fig. 9) | Fig. 4a | Fig. 7a |
|---|---|---|---|---|---|---|
| | CTF-POMIS | 3 | 6 | 5 | 21 | 12 |
| IS | CTF-MIS | 10 | 18 | 40 | 50 | 25 |
| | POMIS | 2 | 4 | 5 | 10 | 10 |
| | CTF-POMIS | 12 | 36 | 68 | 131 | 39 |
| Arms | CTF-MIS | 31 | 69 | 239 | 347 | 89 |
| | POMIS | 6 | 16 | 36 | 43 | 31 |

Table 2: The number of intervention sets (IS; above) and the corresponding number of arms under binary domains (below). The number of CTF-POMIS refers to *representative* CTF-POMIS.

**Task 1.** The bandit instance is associated with an SCM $\mathcal{M}_1$ associated with the causal diagram in Fig. 5a defined as follows:

$$
\mathcal{M}_1 = \begin{cases}
\mathbf{U} & = \{U_W, U_X, U_Z, U_Y, U_{ZY}\} \\
\mathbf{V} & = \{W, X, Z, Y\} \\
\mathcal{F} & = \begin{cases} f_W = u_W, f_X = z \vee u_X, \\ f_Z = u_Z \vee w \oplus u_{ZY}, f_Y = 1 - (u_Y \oplus x \oplus (1-w) \oplus u_{ZY}) \end{cases} \\
P(\mathbf{U}) & = \begin{cases} U_W \sim \texttt{Bern}(0.1), U_X \sim \texttt{Bern}(0.1), U_Z \sim \texttt{Bern}(0.1), \\ U_Y \sim \texttt{Bern}(0.1), U_{ZY} \sim \texttt{Bern}(0.1). \end{cases}
\end{cases}
\tag{10}
$$

In this setting, the expected reward of the optimal action is $\mu_{\mathbf{X}^\star_*} = \mu_{w=0, Z_{w=1}} \approx 0.8848$, and $\Delta_{\mathcal{L}_{\leq 2}} = \mu_{\mathbf{X}^\star_*} - \mu_{\mathbf{x}^\star} \approx 0.0648 > 0$.

**Task 2.** The bandit instance is associated with an SCM $\mathcal{M}_2$ associated with the causal diagram in Fig. 3b defined as follows:

$$
\mathcal{M}_2 = \begin{cases}
\mathbf{U} & = \{U_T, U_W, U_X, U_Y, U_Z, U_{XY}, U_{ZY}\} \\
\mathbf{V} & = \{T, W, X, Z, Y\} \\
\mathcal{F} & = \begin{cases} f_T = u_T, f_W = u_W, f_X = (1-u_{XY}) \oplus (u_X \vee w), \\ f_Z = ((1-(t \oplus (1-u_Z))) \vee x) \oplus u_{ZY}, \\ f_Y = ((1-(u_{ZY} \vee (1-u_{XY}))) \oplus u_Y) \oplus (x \vee z) \end{cases} \\
P(\mathbf{U}) & = \begin{cases} U_T \sim \texttt{Bern}(0.65), U_W \sim \texttt{Bern}(0.65), U_X \sim \texttt{Bern}(0.71), \\ U_Y \sim \texttt{Bern}(0.73), U_Z \sim \texttt{Bern}(0.43), U_{XY} \sim \texttt{Bern}(0.76), \\ U_{ZY} \sim \texttt{Bern}(0.43). \end{cases}
\end{cases}
\tag{11}
$$

In this setting, the expected reward of the optimal action is $\mu_{\mathbf{X}^\star_*} = \mu_{X_{w=1}, Z_{x=0,t}} \approx 0.5912$, which is optimal for both $t = 0$ and $t = 1$. The gap is $\Delta_{\mathcal{L}_{\leq 2}} = \mu_{\mathbf{X}^\star_*} - \mu_{\mathbf{x}^\star} \approx 0.0042 > 0$.

**Task 3.** The first bandit instance is associated with an SCM $\mathcal{M}_3$ (Fig. 8c, left) associated with the causal diagram in Fig. 9 defined as follows:

$$
\mathcal{M}_3 = \begin{cases}
\mathbf{U} & = \{U_A, U_B, U_C, U_D, U_E, U_F, U_{BY}, U_{DY}, U_{BC}\} \\
\mathbf{V} & = \{A, B, C, D, E, F, Y\} \\
\mathcal{F} & = \begin{cases} f_A = u_A, f_B = (1-((1-a) \wedge u_B)) \oplus (u_{BC} \oplus (u_{BY} \oplus f)), \\ f_C = (1-((1-u_C) \oplus (1-a))) \wedge (d \oplus u_{BC}), \\ f_D = (e \oplus u_{DY}) \wedge u_D, f_E = u_E, f_F = u_F, \\ f_Y = (1-((1-b) \wedge u_Y)) \oplus (u_{DY} \oplus (u_{BC} \oplus c)) \end{cases} \\
P(\mathbf{U}) & = \begin{cases} U_A \sim \texttt{Bern}(0.2), U_B \sim \texttt{Bern}(0.3), U_C \sim \texttt{Bern}(0.44), \\ U_D \sim \texttt{Bern}(0.52), U_E \sim \texttt{Bern}(0.55), U_F \sim \texttt{Bern}(0.24), \\ U_Y \sim \texttt{Bern}(0.67), U_{BY} \sim \texttt{Bern}(0.74), U_{DY} \sim \texttt{Bern}(0.35), \\ U_{BC} \sim \texttt{Bern}(0.37). \end{cases}
\end{cases}
\tag{12}
$$

In this setting, the expected reward of the optimal action is $\mu_{\mathbf{X}^\star_*} = \mu_{B_{a=1,b=0},C_{a=0,e=0}} \approx 0.618$. The gap is $\Delta_{\mathcal{L}_{\leq 2}} = \mu_{\mathbf{X}^\star_*} - \mu_{\mathbf{x}^\star} \approx 0.0248 > 0$.

The second SCM $\mathcal{M}'_3$ (Fig. 8c, right) is defined as follows:

$$\mathcal{M}'_3 = \begin{cases} \mathbf{U} & = \{U_A, U_B, U_C, U_D, U_E, U_F, U_{BY}, U_{DY}, U_{BC}\} \\ \mathbf{V} & = \{A, B, C, D, E, F, Y\} \\ \mathcal{F} & = \begin{cases} f_A = u_A, f_B = (1 - u_{BY}) \vee (((1 - (u_{BC} \vee u_B)) \wedge f) \wedge a), \\ f_C = (1 - u_{BC}) \vee ((1 - a) \wedge ((1 - u_C) \wedge d)), \\ f_d = (u_D \vee (1 - u_{DY})) \wedge e, \\ f_E = u_E, f_F = u_F, \\ f_Y = (1 - u_{BY}) \vee (((1 - (u_{DY} \vee u_Y)) \wedge c) \wedge b) \end{cases} \\ P(\mathbf{U}) & = \begin{cases} U_A \sim \mathtt{Bern}(0.34), U_B \sim \mathtt{Bern}(0.52), U_C \sim \mathtt{Bern}(0.75), \\ U_D \sim \mathtt{Bern}(0.47), U_E \sim \mathtt{Bern}(0.46), U_F \sim \mathtt{Bern}(0.76), \\ U_Y \sim \mathtt{Bern}(0.67), U_{BY} \sim \mathtt{Bern}(0.63), U_{DY} \sim \mathtt{Bern}(0.68), \\ U_{BC} \sim \mathtt{Bern}(0.26). \end{cases} \end{cases} \quad (13)$$

The expected reward of the optimal action is $\mu_{\mathbf{X}^\star_*} = \mu_{B_{b=1},C_{c=1}} = \mu_{b=0,c=1} = 1$, which implies $\Delta_{\mathcal{L}_{\leq 2}} = \mu_{\mathbf{X}^\star_*} - \mu_{\mathbf{x}^\star} = 0$.

## D  DISCUSSIONS

**Online optimal strategy.**   One might be concerned that while the algorithm (Algo. 1) effectively uses causal knowledge to eliminate suboptimal actions before learning begins, it then switches to traditional solvers that ignore additional observations available during each round. Indeed, there exists a rich body of research that updates parameters under graphical constraints (Zhang and Bareinboim, 2022; Bellot et al., 2023; Wei et al., 2023; Jalaldoust et al., 2024) in online learning. However, such approaches often rely on optimization-based approaches—such as *canonical SCM* (Zhang et al., 2022) or *neural causal models* (NCMs) (Xia et al., 2021; 2023) that assume full parameterization. In contrast, our approach focuses on leveraging structural knowledge, before any online interaction, and without requiring parameterization or any strong assumptions beyond a given graphical structure.

**Future work.**   Beyond structural causal bandits, we believe that counterfactual decision making will offer substantial practical value when integrated with causal reinforcement learning (Zhang and Bareinboim, 2022; Hwang et al., 2024; Bareinboim et al., 2024), rehearsal learning (Qin et al., 2023; 2025; Du et al., 2024; 2025a;b; Tao et al., 2025; 2026), and sequential planning (Pearl and Robins, 1995; Jung et al., 2024).

## E  TECHNICAL DETAILS

In this section, we provide several technical details to support our main results.

### E.1  COUNTERFACTUAL ACTION

**Lemma 5** (Observational action; Lemma D.8 in Raghavan and Bareinboim (2025)). *An agent can draw a sample from $P(Y)$ associated with an SCM $\mathcal{M}$ by the following operations.*

  *(i)* SELECT$^{(i)}$.

  *(ii)* READ$(Y)^{(i)} = y \sim P(Y)$.

**Lemma 6** (Interventional action; Lemma D.10 in Raghavan and Bareinboim (2025)). *An agent can draw a sample from $P(Y_\mathbf{x})$ associated with an SCM $\mathcal{M}$ by the following operations.*

  *(i)* SELECT$^{(i)}$.

*(ii)* RAND$(\mathbf{X})^{(i)}$.

*(iii) If* RAND$(\mathbf{X})^{(i)} = \mathbf{x}$, *then* READ$(Y)^{(i)} = y \sim P(Y_{\mathbf{x}})$, *else repeat i–iii.*

With the counterfactual randomization operation CTF-RAND, an agent can draw reward samples from counterfactual action $do(\mathbf{X} = \mathbf{X}_*)$ for $\mathbf{X}_*$ satisfying Prop. 1.

**Lemma 7** (Counterfactual action). *Let $\mathbf{X}_*$ be an CTF-SCB action equivalent to a target CTF-MIS satisfying Prop. 3. An agent can draw a sample from $P(Y_{\mathbf{X}_*})$ associated with an SCM $\mathcal{M}$ by the following operations. Repeat (i–ii) for $X_{j[\mathbf{w}_j]} \in \mathbf{X}_*$:*

*(i)* SELECT$^{(i)}$.

*(ii)* CTF-RAND$(W_j \to X_j)^{(i)}$ *for $W_j \in \mathbf{W}_j \cap \mathtt{Pa}(X_j)_{\mathcal{G}}$: fixing the value of counterfactual mediator $\tilde{W}_j$ using a randomizing device for unit $i$. If $\tilde{w}_j$ is inconsistent with $\mathbf{w}_j$, return to (i).*

*(iii)* READ$(Y)^{(i)} = y \sim P(Y_{\mathbf{X}_*})$.

*This procedure is written as if it were based on access to $\mathbb{A}^\dagger$. Further,* CTF-RAND *must always be applied with respect to a graphical children; it is not possible to bypass a child and directly alter the perception of a descendant.*

*Proof.* This follows from the correctness of CTF-REALIZABLE (Theorem 3.5 in Raghavan and Bareinboim (2025)). $\square$

**Remark 2.** To enact an intervention such as $do(\mathbf{x})$, we draw a random value and reject it if the draw is not equal to $\mathbf{x}$. This procedure is adopted for clarity of presentation and to provide general intuition linking a counterfactual action to its practical implementation; in environments where $do(\mathbf{x})$ or $do(\tilde{\mathbf{x}})$ is feasible, this procedure can thus be readily replaced with WRITE$(\mathbf{X} : \mathbf{x})$ and CTF-WRITE$(\mathbf{w}_j \to X_j)$, where CTF-WRITE is simply the deterministic counterpart of CTF-RAND.

### E.2 GRAPHICAL CONSTRAINT INDUCED BY REALIZABILITY

We investigate which graphical constraint is induced by the realizability condition Prop. 1.

**Proposition 5.** *Let $\mathbf{X}_* = \{X_{i[\mathbf{w}_i]}\}_{i=1}$ be a CTF-MIS with respect to $\langle \mathcal{G}, Y \rangle$. For any $X_{i[\mathbf{w}_i]}, X_{j[\mathbf{w}_j]}$, if $X_i \in \mathtt{An}(X_j)_{\mathcal{G}_{\overline{\mathbf{W}_j}}}$, then $\mathbf{w}_i \subseteq \mathbf{w}_j$ holds.*

*Proof.* Since $X_i \in \mathtt{An}(X_j)_{\mathcal{G}_{\overline{\mathbf{W}_j}}}$, we know that $X_{i[\mathbf{w}_j \cap \mathtt{An}(X_j)_{\mathcal{G}_{\overline{\mathbf{W}_j}}}]} \in \mathtt{An}(X_{j[\mathbf{w}_j]}) \subseteq \mathtt{An}(\mathbf{X}_*)$. In order for $\mathbf{X}_*$ to be realizable, the subscript assignments must be consistent, implying $\mathbf{w}_i = \mathbf{w}_j \cap \mathtt{An}(X_i)_{\mathcal{G}_{\overline{\mathbf{W}_j}}} \subseteq \mathbf{w}_j$. This completes the proof. $\square$

Consider the causal diagrams in Fig. 12a. The counterfactual regime $\mathbf{X}_* = \{X_{1[w_1,w_2]}, X_{2[w_2,w_3,w_4]}\}$ in $\mathcal{G}_1$ is *not* realizable. To see this, observe that $X_{1[w_1,w_2]} \in \mathtt{An}(X_{1[w_1,w_2]}) \subseteq \mathtt{An}(\mathbf{X}_*)$. Hence, for $\mathbf{X}_*$ to be realizable (i.e., $\mathtt{An}(Y_{\mathbf{x}}, \mathbf{X}_*)$ is realizable), there must not exist any variable of the form $X_{1[\mathbf{t}]}$ with $\mathbf{t} \neq \{w_1, w_2\}$ in $\mathtt{An}(\mathbf{X}_*)$. Since $X_1 \in \mathtt{An}(X_2)_{\mathcal{G}_{1\overline{\mathbf{w}_2}}}$, $\mathtt{An}(X_{2[\mathbf{w}_2]})$ contains $X_{1[\mathbf{w}_2 \cap \mathtt{An}(X_1)_{\overline{\mathbf{w}_2}}]}$ by the definition of counterfactual ancestors. However, we have $\mathbf{w}_2 \cap \mathtt{An}(X_1)_{\overline{\mathbf{w}_2}} = \{w_2\} \neq \{w_1, w_2\}$, which results in a conflict between $X_{1[w_1,w_2]} \in \mathtt{An}(X_{1[\mathbf{w}_1]})$ and $X_{1[w_2]} \in \mathtt{An}(X_{2[\mathbf{w}_2]})$. Therefore, $\mathbf{X}_*$ is non-realizable, and $\mathbf{X}_*$ is not a valid action in CTF-SCB with respect to $\langle \mathcal{G}_1, Y \rangle$.

Next, we consider the causal diagrams in Fig. 12b. Similarly, $\mathbf{X}_* = \{X_{1[w_1,w_2]}, X_{2[w_2,w_3,w_4,w_5]}\}$ is also *not* realizable, as $X_{1[w_2,w_3,w_5]} \in \mathtt{An}(X_{1[\mathbf{w}_1]}) \subseteq$ conflicts with $X_{1[w_1,w_2]} \in \mathtt{An}(X_{2[\mathbf{w}_2]})$.

In contrast, $\mathbf{X}_* = \{X_{1[w_2]}, X_{2[w_2,w_3]}\}$ is realizable, and constitutes a CTF-MIS with respect to $\langle \mathcal{G}_3, Y \rangle$ (Fig. 12c), since $\mathbf{w}_2 \cap \mathtt{An}(X_1)_{\mathcal{G}_{\overline{\mathbf{W}_2}}} = \{w_2\}$ implies that $X_{1[w_2]} \in \mathtt{An}(X_{2[\mathbf{w}_2]})$ does not conflict with $X_{1[w_2]} \in \mathtt{An}(X_{1[\mathbf{w}_1]})$ (it can be also checked by $W_4$).

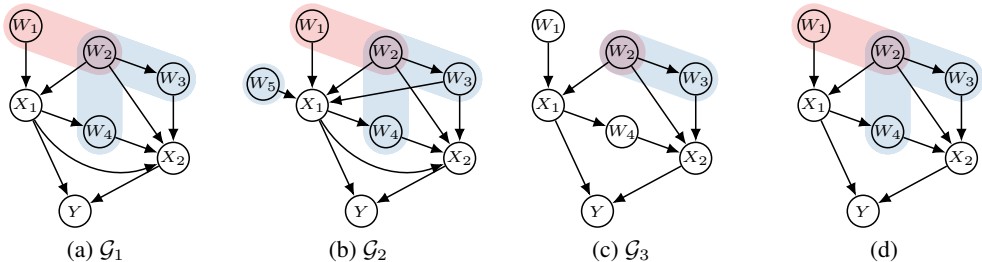

(a) $\mathcal{G}_1$      (b) $\mathcal{G}_2$      (c) $\mathcal{G}_3$      (d)

Figure 12: (a–c) Causal diagrams $\mathcal{G}_1$, $\mathcal{G}_2$ and $\mathcal{G}_3$; red region represents $\mathbf{w}_1$ and the blue one represents $\mathbf{w}_2$. Notably, ancestral relations are independent of confounders; (c) $\mathcal{G}_3$ with a realizable regime (valid action) when the value of $W_2$ is consistent; (d) Realizable regardless the value of $W_2$.

Moreover, $\mathbf{X}_* = \{X_{1[w_1,w_2]}, X_{2[w_2,w_3,w_4]}\}$ in Fig. 12d is also realizable because $\mathrm{An}(X_{2[\mathbf{w}_2]})$ does not include any counterfactual variable of the form $X_{1[\mathbf{t}]}$. These examples illustrated in Fig. 12 indicate that if $X_1 \in \mathrm{An}(X_2)_{\mathcal{G}_{\overline{\mathbf{w}_2}}}$, then $\mathbf{w}_2 \cap \mathrm{An}(X_1)_{\mathcal{G}_{\overline{\mathbf{w}_2}}}$ must be identical with $\mathbf{w}_1$.

### E.3 ALGORITHM DETAILS

We provide a more detailed version of Algo. 1, as demonstrated in Algo. 2.

---

**Algorithm 2:** Algorithm enumerating all representative CTF-POMISs (**full version**)

**Input:** Causal diagram $\mathcal{G}$; and a reward variable $Y$
**Output:** All representative CTF-POMISs with respect to $\langle \mathcal{G}, Y \rangle$

1   Set $\mathcal{H} = \mathcal{G}[\mathrm{An}(Y)_{\mathcal{G}}]$.
2   **for** each children choice $(\mathbf{X}_W)_{W \in \mathbf{V} \setminus \{Y\}} \in \bigtimes_{W \in \mathbf{V} \setminus \{Y\}} 2^{\mathrm{Ch}(W)_{\mathcal{H}}}$ **do**
3      Initialize a map $\mathrm{pa}[X] = \emptyset$ for all $X \in \mathbf{V}(\mathcal{H})$.
4      **for** each $W \in \mathbf{W}$ and each child $X \in \mathbf{X}_W$ **do**
5         Update $\mathrm{pa}[X] = \mathrm{pa}[X] \cup \{W\}$ **if** $W \neq Y$ **else** $\mathrm{pa}[X] = \mathrm{pa}[X] \cup \{X\}$.
6         **if** $X \subsetneq \mathrm{pa}[X]$ **then break**
7      **else**
8         **if** $\mathrm{IB}(\mathcal{H}', Y) = \emptyset$ where $\mathcal{H}' = \mathcal{H} \setminus \{W \to X \mid \forall X \in \mathbf{V}(\mathcal{H}), \forall W \in \mathrm{pa}[X]\}$ **then**
9            Let $\pi = \{X_1 \prec X_2 \prec \cdots \prec X_n\}$ be a topological order of $\mathbf{V}(\mathcal{H}) \setminus \{Y\}$.
10           Initialize $\mathbf{X}_* = \{\}$.
11           **for** each $X \in \pi$ with $\mathrm{pa}[X] \neq \emptyset$ **do**
12              Add $\{X_{\mathbf{z}} \mid X \in \bigcup_{W \in \mathbf{W}} \mathbf{X}_W\}$ into $\mathbf{X}_*$ where $\mathbf{z} = \bigcup_{T \in \mathrm{An}(X)_{\mathcal{H}_{\overline{\mathrm{pa}[X]}}}} \mathrm{pa}[T]$.
13           Compute $\mathbf{X}'_* = \text{CTF-MISIFY}(\mathcal{G}, \mathbf{X}_*, Y)$.
14           **yield** $\mathbf{X}'_*$ **if** $\mathrm{An}(Y_{\mathbf{x}}, \mathbf{X}'_*)$ satisfies Prop. 1.

---

**Lines 4–5.** When constructing the map pa, if $X = Y$, we replace $X$ with $W$ since $Y_{Y_w} = Y_{W_w}$ (Sec. 7.3 in Pearl (2000)) for all $X \in \mathbf{V}(\mathcal{H})$ and $W \in \mathrm{pa}[X]$. This substitution ensures that the algorithm does not violate the conditions $X_i \in \mathbf{V} \setminus \{Y\}$ and $\mathbf{W}_i \subseteq \mathbf{V} \setminus \{Y\}$ for any potential result $X_{i[\mathbf{w}_i]} \in \mathbf{X}_*$.

**Line 6.** If $X \subsetneq \mathrm{pa}[X]$, this will induce counterfactuals of the form $X_{x,\cdots}$, which are equivalent to either $X_x$ or $\mathbf{C}_x$, where $\mathbf{C} = \mathrm{Ch}(X)_{\mathcal{G}}$. Therefore, this case will be addressed, or has already been addressed, in another loop corresponding to line 2.

**Line 8.** It is evident by Thm. 2 and Cor. 3.

**Lines 9–12.** Note that $\{X_{\mathrm{pa}[X]}\}_{X \in \cup_{w \in \mathbf{W}} \mathbf{X}_W}$ may *not* be a CTF-MIS or CTF-POMIS since its realizability has not been verified. To maintain the same structure of the subgraph $\mathcal{H}'$ while modifying $\{X_{\mathrm{pa}[X]}\}$ into a CTF-POMIS, we first append the subscripts induced by the ancestors of each counterfactual variable of $X_{\mathrm{pa}[X]}$ (Prop. 5). Through this procedure, $\mathbf{X}_*$ satisfies Lem. 4 in symbolic form; yet a conflict may still arise between $\mathrm{An}(Y_{\mathbf{x}})$ and $\mathrm{An}(\mathbf{X}_*)$.

**Lines 13–14.** Finally, CTF-MISIFY removes unnecessary counterfactuals from $\mathbf{X}_*$ according to Thm. 1. In this process, the counterfactual regime graph $\mathcal{H}_{\mathbf{X}_*}$ does not change, since its construction accounts for ancestrality from the perspective of CTF-Rule 3 and realizability. Therefore, by checking for conflicts through Prop. 1, we can confirm that this counterfactual symbol constitutes a valid representative CTF-POMIS.

### E.4 NESTED COUNTERFACTUAL REGIMES

One may raise a concern regarding actions based on *nested* counterfactuals (Shpitser, 2013; Correa et al., 2021). For example, consider a nested realizable regime $\{T_{W_x}, Z_{x'}\}$ (i.e., $\mathrm{An}(Y_{tz}, T_w, W_x, Z_{x'})$ does not contain any conflict) in the causal diagram shown in Fig. 7a. Then, we can derive as follows:

$$\mu_{T_{W_x}, Z_{x'}} = \sum_{y,z,t,w} y P(y_{tz}, t_w, w_x, z_{x'}) \qquad \text{CUT} \qquad (14)$$

$$= \sum_{y,z,t,w} y P(y_{wtz}, t_{wz}, w_x, z_{x'}) \qquad \text{CTF-Rule 3} \qquad (15)$$

$$= \sum_{y,z,t,w} y P(y_{wz}, t_{wz}, w_x, z_{x'}) \qquad \text{CTF-Rule 1} \qquad (16)$$

$$= \sum_{y,z,w} y P(y_{wz}, w_x, z_{x'}) \qquad \text{summation} \qquad (17)$$

$$= \mu_{W_x, Z_{x'}}. \qquad \text{CUT} \qquad (18)$$

Hence, we observe that $\mathbf{X}_* = \{W_x, Z_{x'}\}$ can account for the nested counterfactual regime $\{T_{W_x}, Z_{x'}\}$. The following proposition shows that (possibly infinite) iterated nested counterfactual regimes can be disregarded in general.

**Proposition 6.** *For any realizable nested counterfactual regime, there exists an action that is equivalent to it; that is, $\mathcal{A}^\star$ covers all nested realizable counterfactual regimes.*

*Proof.* Let $\mathbf{X}_{**} = \{X_{i[\mathbf{Z}_*^i]}\}_{i=1}$ where $\mathbf{Z}_*^i = \{Z_{j[\mathbf{t}_j^i]}^i\}_{j=1}$ be a realizable nested regime. We suppose $\mathbf{Z}^i \subseteq \mathrm{An}(X_i)_{\mathcal{G}_{\overline{\mathbf{Z}_i}}}$ and $\mathbf{T}_j^i \subseteq \mathrm{An}(Z_j^i)_{\mathcal{G}_{\overline{\mathbf{T}_j^i}}}$; otherwise, one can find an equivalent regime by interventional minimization (Lem. 1). We now derive as follows:

$$\mu_{\mathbf{X}_{**}} = \sum_{y,\mathbf{x},\mathbf{z}} y P(y_\mathbf{x}, \bigwedge_i x_{i[\mathbf{z}^i]}, \bigwedge_{i,j} z_{j[\mathbf{t}_j^i]}^i) \qquad \text{CUT} \qquad (19)$$

$$= \sum_{y,\mathbf{x},\mathbf{z}} y P(y_{\mathbf{xz}}, \bigwedge_i x_{i[\mathbf{z}^i]}, \bigwedge_{i,j} z_{j[\mathbf{t}_j^i]}^i) \qquad \textbf{Claim 1} \qquad (20)$$

$$= \sum_{y,\mathbf{x},\mathbf{z}} y P(y_\mathbf{z}, \bigwedge_i x_{i[\mathbf{z}^i]}, \bigwedge_{i,j} z_{j[\mathbf{t}_j^i]}^i) \qquad \textbf{Claim 2} \qquad (21)$$

$$= \sum_{y,\mathbf{z}} y P(y_\mathbf{z}, \bigwedge_{i,j} z_{j[\mathbf{t}_j^i]}^i) \qquad \text{marginalization} \qquad (22)$$

$$= \mu_{\bigcup_i \mathbf{Z}_*^i}. \qquad (23)$$

Note that for any pair of counterfactual variables $Z_{j[\mathbf{t}_j^i]}^i \in \mathbf{Z}_*^i$ and $Z_{l[\mathbf{t}_l^k]}^k \in \mathbf{Z}_*^k$, if $Z^i = Z^k$, then $\mathbf{t}_j^i = \mathbf{t}_l^k$ holds by the realizability condition (Prop. 1). Hence, the realizable nested regime $\mathbf{X}_{**}$ is equivalent to $\bigcup_i \mathbf{Z}_*^i \in \mathcal{A}$ under **Claim 1** and **Claim 2**. Consequently, establishing the validity of these two claims implies that CTF-POMIS actions are sufficient to cover *all* realizable nested counterfactual regimes. We now provide a line-by-line explanation of the main part of the proof.

**Claim 1.** Let $\mathbf{Z}' \triangleq \mathbf{Z} \cap \mathrm{An}(Y)_{\mathcal{G}_{\overline{\mathbf{X}}}}$ and $\mathbf{Z}'' \triangleq \mathbf{Z} \setminus \mathbf{Z}'$, implying $\mathbf{Z}'' \cap \mathrm{An}(Y)_{\mathcal{G}_{\overline{\mathbf{X}}}} = \emptyset$. Then, according to the realizability condition, for any $Z_j^i \in \mathbf{Z}'$ we have $Z_{j[\mathbf{x}]}^i = Z_{j[\mathbf{x} \cap \mathrm{An}(Z_j^i)_{\mathcal{G}_{\overline{\mathbf{X}}}}]}^i \in \mathrm{An}(Y_\mathbf{x})$ which coincides with $Z_{j[\mathbf{t}_j^i]}^i \in \mathbf{Z}_*^i$. Therefore, $\bigwedge_{i,j} z_{j[\mathbf{t}_j^i]}^i = \bigwedge_{i,j} z_{j[\mathbf{x}]}^i = \mathbf{z}'_\mathbf{x}$ holds. This property allows us

to write as follows:

$$
\text{Eq. (19)} = \sum_{y,\mathbf{x},\mathbf{z}} y P(y_{\mathbf{x}}, \bigwedge_i x_{i[\mathbf{z}^i]}, \overbrace{\bigwedge_{l,j} z^l_{j[\mathbf{t}^l_j]}}^{\mathbf{Z}'}, \overbrace{\bigwedge_{k,j} z^k_{j[\mathbf{t}^k_j]}}^{\mathbf{Z}''}) \qquad \text{def} \qquad (24)
$$

$$
= \sum_{y,\mathbf{x},\mathbf{z}} y P(y_{\mathbf{x}}, \bigwedge_i x_{i[\mathbf{z}^i]}, \mathbf{z}'_{\mathbf{x}}, \bigwedge_{k,j} z^k_{j[\mathbf{t}^k_j]}) \qquad \text{realizability} \qquad (25)
$$

$$
= \sum_{y,\mathbf{x},\mathbf{z}} y P(y_{\mathbf{x}\mathbf{z}'}, \bigwedge_i x_{i[\mathbf{z}^i]}, \mathbf{z}'_{\mathbf{x}}, \bigwedge_{k,j} z^k_{j[\mathbf{t}^k_j]}) \qquad \text{CTF-Rule 1} \qquad (26)
$$

$$
= \sum_{y,\mathbf{x},\mathbf{z}} y P(y_{\mathbf{x}\mathbf{z}'\mathbf{z}''}, \bigwedge_i x_{i[\mathbf{z}^i]}, \mathbf{z}'_{\mathbf{x}}, \bigwedge_{k,j} z^k_{j[\mathbf{t}^k_j]}) \qquad \text{CTF-Rule 3} \qquad (27)
$$

$$
= \sum_{y,\mathbf{x},\mathbf{z}} y P(y_{\mathbf{x}\mathbf{z}}, \bigwedge_i x_{i[\mathbf{z}^i]}, \mathbf{z}'_{\mathbf{x}}, \bigwedge_{k,j} z^k_{j[\mathbf{t}^k_j]}) \qquad \text{def} \qquad (28)
$$

$$
= \sum_{y,\mathbf{x},\mathbf{z}} y P(y_{\mathbf{x}\mathbf{z}}, \bigwedge_i x_{i[\mathbf{z}^i]}, \bigwedge_{l,j} z^l_{j[\mathbf{t}^l_j]}, \bigwedge_{k,j} z^k_{j[\mathbf{t}^k_j]}) \qquad (29)
$$

which concludes the proof of the first claim. We now proceed to demonstrate the next claim.

**Claim 2.** Without loss of generality, we suppose $X_1 \prec X_2 \prec \cdots \prec X_n$ where $n = |\mathbf{V}(\mathbf{X}_{**})|$. Consider an arbitrary pair $X_i, X_j$ with $X_i \prec X_j$.

First, **if** $X_i \in \text{An}(X_j)_{\mathcal{G}_{\overline{\mathbf{Z}^j}}}$ holds, then $\mathbf{z}^i = \mathbf{z}^j \cap \text{An}(X_i)_{\mathcal{G}_{\overline{\mathbf{Z}^j}}} \subseteq \mathbf{z}^j$ by Prop. 5. Therefore, we obtain that $\{x_{i[\mathbf{z}^i]}, x_{j[\mathbf{z}^j]}\}$ can be written as $\{x_{i[\mathbf{z}^i\mathbf{z}^j]}, x_{j[\mathbf{z}^i\mathbf{z}^j]}\}$.

Otherwise, **if** $X_i \notin \text{An}(X_j)_{\mathcal{G}_{\overline{\mathbf{Z}^j}}}$, then $\mathbf{Z}^i \cap \text{An}(X_j)_{\mathcal{G}_{\overline{\mathbf{Z}^j}}} = \emptyset$ holds, implying $\{x_{i[\mathbf{z}^i]}, x_{j[\mathbf{z}^i\mathbf{z}^j]}\}$ by CTF-Rule 3. Furthermore, suppose $\mathbf{Z}^j \cap \text{An}(X_i)_{\mathcal{G}_{\overline{\mathbf{Z}^i}}} \neq \emptyset$. Then, this means $X_{i[\mathbf{z}^i]} \in \text{An}(X_{j[\mathbf{z}^j]})$, implying $\mathbf{z}^i = \mathbf{z}^j \cap \text{An}(X_i)_{\mathcal{G}_{\overline{\mathbf{Z}^j}}} \subseteq \mathbf{z}^j$ by Prop. 5. Therefore, $\{x_{i[\mathbf{z}^i]}, x_{j[\mathbf{z}^i\mathbf{z}^j]}\} = \{x_{i[\mathbf{z}^i\mathbf{z}^j]}, x_{j[\mathbf{z}^i\mathbf{z}^j]}\}$ holds. Otherwise, if $\mathbf{Z}^j \cap \text{An}(X_i)_{\mathcal{G}_{\overline{\mathbf{Z}^i}}} = \emptyset$, applying CTF-Rule 3 results in $\{x_{i[\mathbf{z}^i]}, x_{j[\mathbf{z}^i\mathbf{z}^j]}\} = \{x_{i[\mathbf{z}^i\mathbf{z}^j]}, x_{j[\mathbf{z}^i\mathbf{z}^j]})\}$.

Hence, we can say that $\{x_{i[\mathbf{z}^i]}, x_{j[\mathbf{z}^j]}\} = \{x_{i[\mathbf{z}^i\mathbf{z}^j]}, x_{j[\mathbf{z}^i\mathbf{z}^j]}\}$ for any arbitrary pair $X_i, X_j$ with $X_i \prec X_j$. Therefore, we can express as follows:

$$
\text{Eq. (20)} = \sum_{y,\mathbf{x},\mathbf{z}} y P(y_{\mathbf{x}\mathbf{z}}, \bigwedge_i x_{i[\mathbf{z}]}, \bigwedge_{i,j} z^i_{j[\mathbf{t}^i_j]}) \qquad \text{realizability} \qquad (30)
$$

$$
= \sum_{y,\mathbf{x},\mathbf{z}} y P(y_{\mathbf{x}\mathbf{z}}, \mathbf{x}_{\mathbf{z}}, \bigwedge_{i,j} z^i_{j[\mathbf{t}^i_j]}) \qquad \text{def} \qquad (31)
$$

$$
= \sum_{y,\mathbf{x},\mathbf{z}} y P(y_{\mathbf{z}}, \mathbf{x}_{\mathbf{z}}, \bigwedge_{i,j} z^i_{j[\mathbf{t}^i_j]}) \qquad \text{CTF-Rule 1} \qquad (32)
$$

$$
= \sum_{y,\mathbf{x},\mathbf{z}} y P(y_{\mathbf{z}}, \bigwedge_i x_{i[\mathbf{z}^i]}, \bigwedge_{i,j} z^i_{j[\mathbf{t}^i_j]}) \qquad (33)
$$

which concludes the proof of the second claim, and thus completes the main proof. $\qquad\square$

# F  OMITTED PROOFS

In this section, we provide detailed proofs of the statements presented in the main body of the paper. For readability, we restate all of them.

## F.1  PROOF OF THEOREM 1

**Theorem 1** (Graphical characterization of CTF-MIS). *A counterfactual* $\mathbf{X}_* = \{X_{i[\mathbf{w}_i]}\}_{i=1} \in \mathcal{A}$ *is a* CTF-*MIS if and only if (i)* $\mathbf{X} \subseteq \text{An}(Y)_{\mathcal{G}_{\overline{\mathbf{X}}}}$ *and (ii) for any* $X_{i[\mathbf{w}_i]} \in \mathbf{X}_*$, $\mathbf{W}_i \cap \text{An}(X_i)_{\mathcal{G}_{\overline{\mathbf{X}\setminus\{X_i\}}}} \neq \emptyset$.

*Proof.* We first note that actions in the $\mathcal{L}_{\leq 2}$ regime such as $do(\mathbf{w})$ correspond to intervention on a set of counterfactual variables of the form $do(\mathbf{W} = \mathbf{W_w})$ within $\mathbf{X}_*$. For instance, in the counterfactual expression $Y_{z,X_w}$, the corresponding regime is $\mathbf{X}_* = \{Z_z, X_w\}$ and $\mathbf{V}(\mathbf{X}_*) = \{Z, X\}$[8].

**(Only if)** If some $\mathbf{X}_*$ does *not* satisfy the first condition, it means there exists $X_{i[\mathbf{w}_i]} \in \mathbf{X}_*$ such that $X_i$ does not have any *proper causal path* to $Y$ with respect to $\mathbf{V}(\mathbf{X}_*)$[9], implying $\mu_{\mathbf{X}_*} = \mu_{\mathbf{X}_* \setminus \{X_{i[\mathbf{w}_i]}\}}$. To see this, we derive as follows:

$$\mu_{\mathbf{X}_*} = \sum_{y,\mathbf{x}} y P(y_\mathbf{x}, \mathbf{X}_* = \mathbf{x}) \qquad \text{CUT} \qquad (34)$$

$$= \sum_{y,x_i,\mathbf{x}\setminus\{X_i\}} y P(y_{x_i,\mathbf{x}\setminus\{X_i\}}, x_{i[\mathbf{w}_i]}, \bigwedge_{j\neq i} x_{j[\mathbf{w}_j]}) \qquad \text{def} \qquad (35)$$

$$= \sum_{y,x_i,\mathbf{x}\setminus\{X_i\}} y P(y_{\mathbf{x}\setminus\{X_i\}}, x_{i[\mathbf{w}_i]}, \bigwedge_{j\neq i} x_{j[\mathbf{w}_j]}) \qquad \text{CTF-Rule 3} \qquad (36)$$

$$= \sum_{y,\mathbf{x}\setminus\{X_i\}} y P(y_{\mathbf{x}\setminus\{X_i\}}, \bigwedge_{j\neq i} x_{j[\mathbf{w}_j]}) \qquad \text{summation} \qquad (37)$$

$$= \mu_{\mathbf{X}_* \setminus \{X_{i[\mathbf{w}_i]}\}}. \qquad (38)$$

Therefore, we have shown that if $\mathbf{X}_*$ violates the first condition, then it is *not* a CTF-MIS. We now proceed to next step.

For the sake of contradiction, let $\mathbf{X}_* \in \mathcal{A}$ be a CTF-SCB action satisfying the first condition but *not* the second; we assume $\mathbf{W}_i \cap \text{An}(X_i)_{\mathcal{G}_{\overline{\mathbf{X}\setminus\{X_i\}}}} = \emptyset$, implying that there exists a minimal $\mathbf{X}' = \{X_j\}_{j=1} \subseteq \mathbf{V}(\mathbf{X}_*)\setminus\{X_i\}$ such that $\mathbf{W}_i \cap \text{An}(X_i)_{\mathcal{G}_{\overline{\mathbf{X}'}}} = \emptyset$. This implies $\mathbf{X}'$ blocks *all* directed paths from $\mathbf{W}_i$ to $X_i$. The notable point is that $X_{j[\mathbf{w}_j]} = X_{j[\mathbf{w}_i \cap \text{An}(X_j)_{\mathcal{G}_{\overline{\mathbf{W}_i}}}]} = X_{j[\mathbf{w}_i]}$ and $\mathbf{W}_j \subseteq \mathbf{W}_i$ for any $X_j \in \mathbf{X}'$ (see Prop. 5); otherwise, $X_{j[\mathbf{w}_j]}$ would conflict with $X_{j[\mathbf{w}_i \cap \text{An}(X_j)_{\mathcal{G}_{\overline{\mathbf{W}_i}}}]}$ violating the realizablity condition (Prop. 1). Hence, we can derive $\{X_{j[\mathbf{w}_j]} \mid X_j \in \mathbf{X}'\} = \{X_{j[\mathbf{w}_i]}\}_{j=1} = \mathbf{X}'_{\mathbf{w}_i}$.

For convenience, we denote by $\mathbf{x}^\mathsf{c}$, the value of $X_i \in \mathbf{X}$ that constitutes neither $\mathbf{X}'$ nor $X_i$. Then, we can derive as follows:

$$\mu_{\mathbf{X}_*} = \sum_{y,\mathbf{x}} y P(y_\mathbf{x}, \mathbf{X}_* = \mathbf{x}) \qquad \text{CUT} \qquad (39)$$

$$= \sum_{y,x_i,\mathbf{x}',\mathbf{x}^\mathsf{c}} y P(y_{x_i \mathbf{x}'}, x_{i[\mathbf{w}_i]}, \bigwedge_{X_j \in \mathbf{X}'} x_{j[\mathbf{w}_j]}, \bigwedge_{X_k \notin \mathbf{X}' \cup \{X_i\}} x_{k[\mathbf{w}_k]}) \qquad \text{def} \qquad (40)$$

$$= \sum_{y,x_i,\mathbf{x}',\mathbf{x}^\mathsf{c}} y P(y_{x_i \mathbf{x}'}, x_{i[\mathbf{w}_i]}, \mathbf{x}'_{\mathbf{w}_i}, \bigwedge_{X_k \notin \mathbf{X}' \cup \{X_i\}} x_{k[\mathbf{w}_k]}) \qquad \text{realizability} \qquad (41)$$

$$= \sum_{y,x_i,\mathbf{x}',\mathbf{x}^\mathsf{c}} y P(y_{x_i \mathbf{x}'}, x_{i[\mathbf{w}_i \mathbf{x}']}, \mathbf{x}'_{\mathbf{w}_i}, \bigwedge_{X_k \notin \mathbf{X}' \cup \{X_i\}} x_{k[\mathbf{w}_k]}) \qquad \text{CTF-Rule 1} \qquad (42)$$

$$= \sum_{y,x_i,\mathbf{x}',\mathbf{x}^\mathsf{c}} y P(y_{x_i \mathbf{x}'}, x_{i[\mathbf{x}']}, \mathbf{x}'_{\mathbf{w}_i}, \bigwedge_{X_k \notin \mathbf{X}' \cup \{X_i\}} x_{k[\mathbf{w}_k]}) \qquad \textbf{CTF-Rule 3} \qquad (43)$$

$$= \sum_{y,x_i,\mathbf{x}',\mathbf{x}^\mathsf{c}} y P(y_{\mathbf{x}'}, x_{i[\mathbf{x}']}, \mathbf{x}'_{\mathbf{w}_i}, \bigwedge_{X_k \notin \mathbf{X}' \cup \{X_i\}} x_{k[\mathbf{w}_k]}) \qquad \text{CTF-Rule 1} \qquad (44)$$

$$= \sum_{y,\mathbf{x}',\mathbf{x}^\mathsf{c}} y P(y_{\mathbf{x}'}, \mathbf{x}'_{\mathbf{w}_i}, \bigwedge_{X_k \notin \mathbf{X}' \cup \{X_i\}} x_{k[\mathbf{w}_k]}) \qquad \text{summation} \qquad (45)$$

$$= \sum_{y,\mathbf{x}',\mathbf{x}^\mathsf{c}} y P(y_{\mathbf{x}'}, \bigwedge_{X_j \in \mathbf{X}'} x_{j[\mathbf{w}_i \cap \text{An}(X_j)_{\mathcal{G}_{\overline{\mathbf{W}_i}}}]}, \bigwedge_{X_k \notin \mathbf{X}' \cup \{X_i\}} x_{k[\mathbf{w}_k]}) \qquad \text{minimization} \qquad (46)$$

$$= \mu_{\mathbf{X}_* \setminus \{X_{i[\mathbf{w}_i]}\}} \qquad (47)$$

---

[8]It means that corresponding reward is $Y_\mathbf{w} = Y_{\mathbf{W}_\mathbf{w}}$; if we force a variable $\mathbf{W}$ to have the value $\mathbf{w}$, then $\mathbf{W}$ will indeed take on the value $\mathbf{w}$. Further details can be found in Sec. 7.3 in Pearl (2000).

[9]It refers to a directed path from $X_i$ to $Y$ which does not pass any nodes in $\mathbf{V}(\mathbf{X}_*) \setminus \{X_i\}$.

where CTF-Rule 3 in Eq. (43) holds from $\mathbf{W}_i \cap \text{An}(X_i)_{\mathcal{G}_{\overline{\mathbf{X}'}}} = \emptyset$. We find that $\mathbf{W}_i \cap \text{An}(X_i)_{\mathcal{G}_{\overline{\mathbf{X}'}}} = \emptyset$ induces the equivalence $\mu_{\mathbf{X}_*} = \mu_{\mathbf{X}_* \setminus \{X_{i[\mathbf{w}_i]}\}}$, which contradicts the assumption that $\mathbf{X}_*$ is a CTF-MIS. This concludes the proof.

**(If)** Assume that $\mathbf{X}_* = \{X_{i[\mathbf{w}_i]}\}_{i=1}$ is *not* a CTF-MIS; that is, there exists $\mathbf{X}'_* = \{X_{j[\mathbf{w}_j]}\}_{j=1} \subsetneq \mathbf{X}_*$ such that $\mu_{\mathbf{X}_*} = \mu_{\mathbf{X}'_*}$ for all SCMs. Consider an SCM $\mathcal{M}$ with all variables real-valued where each variable $V_i \in \mathbf{V}$ associates with its own binary exogenous variable $U_i$ following a fair coin, $\text{Bern}(0.5)$. Let the function of an endogenous variable be the sum of values of its parents, i.e., $f_V = \sum \mathbf{pa}_V + \mathbf{u}_V$.

For the sake of contradiction, assume that the two conditions hold: (i) $\mathbf{X} \subseteq \text{An}(Y)_{\mathcal{G}_{\overline{\mathbf{X}}}}$, and (ii) for any $X_{i[\mathbf{w}_i]} \in \mathbf{X}_*$, we have $\mathbf{W}_i \cap \text{An}(X_i)_{\mathcal{G}_{\overline{\mathbf{X} \setminus \{X_i\}}}} \neq \emptyset$. Thus, there exists $X_{k[\mathbf{w}_k]} \in \mathbf{X}_* \setminus \mathbf{X}'_*$. Let $\mathbf{X}''_* \triangleq \mathbf{X}_* \setminus \mathbf{X}'_*$. Then, there exist directed paths from $X_k \in \mathbf{X}'$ to $Y$ in $\mathcal{G}$, and from $\mathbf{W}'_k = \mathbf{W}_k \setminus \text{An}(X_k)_{\mathcal{G}_{\overline{\mathbf{X} \setminus \{X_k\}}}}$ to $X_k$, which are not constrained by the realizability subscripts (Prop. 5). Hence, setting the values of each $\mathbf{W}'_k$ as $\mathbb{E}[\mathbf{W}'_{k[\mathbf{x}'_*]}] + 1$ yields a larger outcome, breaking the equality, which contradicts the assumption that $\mathbf{X}_*$ is *not* a CTF-MIS with respect to $\langle \mathcal{G}, Y \rangle$. This concludes the proof of this direction. □

### F.2 PROOF OF PROPOSITION 2 AND COROLLARY 2

**Proposition 2.** *If $Y$ is not confounded with $\text{An}(Y)_{\mathcal{G}} \setminus \{Y\}$, then $\text{Pa}(Y)_{\mathcal{G}}$ is the only CTF-POMIS.*

*Proof.* Let $\mathbf{X}_*$ be an arbitrary CTF-MIS relative to $\langle \mathcal{G}, Y \rangle$. Let $\mathbf{Z} = \text{Pa}(Y)_{\mathcal{G}} \setminus \mathbf{V}(\mathbf{X}_*)$ and $\mathbf{X}' = \mathbf{V}(\{X_{\mathbf{w}} \in \mathbf{X}_* \mid X \notin \text{Pa}(Y)_{\mathcal{G}}\})$. Then, we derive as follows:

$$\mu_{\mathbf{X}_*} = \sum_{y,\mathbf{x}} y P(y_{\mathbf{x}}, \mathbf{X}_* = \mathbf{x}) \qquad \text{CUT} \qquad (48)$$

$$= \sum_{\mathbf{z},y,\mathbf{x}} y P(y_{\mathbf{x}}, \mathbf{z}_{\mathbf{x}}, \mathbf{X}_* = \mathbf{x}) \qquad \text{marginalization} \qquad (49)$$

$$= \sum_{\mathbf{z},y,\mathbf{x}} y P(y_{\mathbf{xz}}, \mathbf{z}_{\mathbf{x}}, \mathbf{X}_* = \mathbf{x}) \qquad \text{CTF-Rule 1} \qquad (50)$$

$$= \sum_{\mathbf{z},y,\mathbf{x}} y P(y_{\mathbf{x} \cap \mathbf{Pa}_Y, \mathbf{x}'\mathbf{z}}, \mathbf{z}_{\mathbf{x}}, \mathbf{X}_* = \mathbf{x}) \qquad \text{def} \qquad (51)$$

$$= \sum_{\mathbf{z},y,\mathbf{x}} y P(y_{\mathbf{x} \cap \mathbf{Pa}_Y, \mathbf{z}}, \mathbf{z}_{\mathbf{x}}, \mathbf{X}_* = \mathbf{x}) \qquad \text{CTF-Rule 3} \qquad (52)$$

$$= \sum_{\mathbf{z},y,\mathbf{x}} y P(y_{\mathbf{x} \cap \mathbf{Pa}_Y, \mathbf{z}}) P(\mathbf{z}_{\mathbf{x}}, \mathbf{X}_* = \mathbf{x}) \qquad \textbf{Unconfounded} \qquad (53)$$

$$= \sum_{\mathbf{z},\mathbf{x}} \mathbb{E}[Y_{\mathbf{x} \cap \mathbf{Pa}_Y, \mathbf{z}}] P(\mathbf{z}_{\mathbf{x}}, \mathbf{X}_* = \mathbf{x}) \qquad \text{def} \qquad (54)$$

$$\leq \sum_{\mathbf{z},\mathbf{x}} \mu_{\mathbf{pa}_Y^*} P(\mathbf{z}_{\mathbf{x}}, \mathbf{X}_* = \mathbf{x}) \qquad \text{algebra} \qquad (55)$$

$$= \sum_{\mathbf{z}} \mu_{\mathbf{pa}_Y^*} P(\mathbf{z}_{\mathbf{X}_*}) \qquad \text{CUT} \qquad (56)$$

$$= \mu_{\mathbf{pa}_Y^*}. \qquad (57)$$

The derivation begins by applying CUT over $\mathbf{X}_*$, and marginalization over $\mathbf{Z}_{\mathbf{x}}$ in Eqs. (48) and (49). Note that Eqs. (51) and (52) represent the rewriting of counterfactual terms, and removing $\mathbf{x}'$ according to CTF-Rule 3 by the fact that $Y_{\mathbf{x} \cap \mathbf{Pa}_Y, \mathbf{z}}(\mathbf{u}) = f_Y(\mathbf{x} \cap \mathbf{Pa}_Y, \mathbf{z}, \mathbf{u} \cap \mathbf{U}_Y)$; once all of its parents are fixed by intervention $\mathbf{x} \cap \mathbf{Pa}_Y, \mathbf{z}$, the only source of variation for the variable of $Y_{\mathbf{x} \cap \mathbf{Pa}_Y, \mathbf{z}}(\mathbf{u})$ depends only on $\mathbf{u} \cap \mathbf{U}_Y$, which justifies Eq. (53) (see independence restriction of CTFBN in Def. 9). The remaining steps Eqs. (55) to (57) are straightforward. □

**Corollary 2** (Markovian CTF-POMIS). *if $\mathcal{G}$ is Markovian, then $\text{Pa}(Y)_{\mathcal{G}}$ is the only CTF-POMIS.*

*Proof.* In Markovian settings, there are no unobserved confounders in the causal diagram. Therefore, we conclude the proof by Prop. 2. □

### F.3 PROOF OF PROPOSITION 3

**Proposition 3** (Existence of equivalent action). *For any* CTF*-(PO)MIS* $\mathbf{Z}_*$ *for* $\langle \mathcal{G}, Y \rangle$*, there exists an equivalent action* $\mathbf{X}_* = \{X_{i[\mathbf{w}_i]}\}_{i=1} \subseteq \mathtt{An}(\mathbf{Z}_*)$ *satisfying* $\mathbf{X} \subseteq \mathbf{W} \cup \mathtt{Ch}(\mathbf{W})_{\mathcal{G}}$ *where* $\mathbf{W} \triangleq \bigcup_i \mathbf{W}_i$*.*

*Proof.* Let $\mathbf{Z}_* = \{Z_{j[\mathbf{t}_j]}\}_{j=1}$ be a CTF-MIS with respect to $\langle \mathcal{G}, Y \rangle$. Without loss of generality, we only consider $\mathbf{T}_j \neq \{Z_j\}$; if so, we choose $\{X_k\} = \mathbf{T}_k = \{Z_k\}$. We denote by $\mathbf{X}_*^j \subseteq \mathbf{X}_* = \{X_{i[\mathbf{w}_i]}\}_{i=1}$, a set of counterfactual variables such that (i) $\mathbf{T}_j \cap \mathtt{An}(Z_j)_{\mathcal{G}_{\overline{\mathbf{X}^j}}} = \emptyset$, and (ii) for all $X_{i[\mathbf{w}_i]} \in \mathbf{X}_*^j$, $\mathbf{w}_i$ is consistent with $\mathbf{t}_j$, i.e., $\mathbf{w}_i = \mathbf{t}_j \cap \mathtt{An}(X_i)_{\mathcal{G}_{\overline{\mathbf{T}_j}}}$. Note that $\mathbf{X}_*^j$ can always be found as $\mathbf{X}_*^j \triangleq \{X_{i[\mathbf{t}_j \cap \mathtt{An}(X_i)_{\mathcal{G}_{\overline{\mathbf{T}_j}}}]} \mid X_i \in \mathtt{An}(Z_j)_{\mathcal{G}} \cap \mathtt{Ch}(\mathbf{T}_j)_{\mathcal{G}}\}$ for each $\mathbf{t}_j$. Let us derive as follows:

$$\mu_{\mathbf{Z}_*} = \sum_{y,\mathbf{z}} y P(y_{\mathbf{z}}, \mathbf{z}_*) \qquad\qquad \text{CUT} \qquad (58)$$

$$= \sum_{y,\mathbf{z},\mathbf{x}} y P(y_{\mathbf{z}}, \bigwedge_j Z_{j[\mathbf{t}_j]} = z_j, \bigwedge_i X_{i[\mathbf{w}_i]} = x_i) \qquad\qquad \text{marginalization} \qquad (59)$$

$$= \sum_{y,\mathbf{z},\mathbf{x}} y P(y_{\mathbf{z}}, \bigwedge_j Z_{j[\mathbf{t}_j]} = z_j, \bigwedge_j \mathbf{X}_*^j = \mathbf{x}^j) \qquad\qquad \text{def} \qquad (60)$$

where any pair of $(\mathbf{x}^{j_1}, \mathbf{x}^{j_2})$ for $j_1 \neq j_2$ is consistent; that is, $X_{\mathbf{w}} \in \mathbf{X}^{j_1} \cap \mathbf{X}^{j_2}$ implies $\mathbf{w} = \mathbf{w}_{j_1} \cap \mathtt{An}(X)_{\mathcal{G}_{\overline{\mathbf{W}_{j_1}}}} = \mathbf{w}_{j_2} \cap \mathtt{An}(X)_{\mathcal{G}_{\overline{\mathbf{W}_{j_2}}}}$. We now provide a line-by-line explanation of the main part of the proof.

$$\text{Eq. (60)} = \sum_{y,\mathbf{z},\mathbf{x}} y P(y_{\mathbf{z}}, \bigwedge_j Z_{j[\mathbf{x}^j]} = z_j, \bigwedge_j \mathbf{X}_*^j = \mathbf{x}^j) \qquad\qquad \textbf{Claim 1} \qquad (61)$$

$$= \sum_{y,\mathbf{z},\mathbf{x}} y P(y_{\mathbf{z}\mathbf{x}}, \mathbf{z}_{\mathbf{x}}, \bigwedge_j \mathbf{X}_*^j = \mathbf{x}^j) \qquad\qquad \textbf{Claim 2} \qquad (62)$$

$$= \sum_{y,\mathbf{z},\mathbf{x}} y P(y_{\mathbf{x}}, \mathbf{z}_{\mathbf{x}}, \bigwedge_j \mathbf{X}_*^j = \mathbf{x}^j) \qquad\qquad \text{CTF-Rule 1} \qquad (63)$$

$$= \sum_{y,\mathbf{x}} y P(y_{\mathbf{x}}, \bigwedge_j \mathbf{X}_*^j = \mathbf{x}^j) \qquad\qquad \text{summation} \qquad (64)$$

$$= \sum_{y,\mathbf{x}} y P(y_{\mathbf{x}}, \bigwedge_i X_{i[\mathbf{w}_i]} = x_i) \qquad\qquad \text{def} \qquad (65)$$

$$= \mu_{\mathbf{X}_*}. \qquad\qquad \text{CUT} \qquad (66)$$

**Claim 1.** We can derive Eq. (61) from the fact that $\mathbf{X}_*^j = \mathbf{X}_{\mathbf{t}_j}^j$, since all of $X_{i[\mathbf{w}_i]} \in \mathbf{X}_*^j$ are already consistent with $\mathbf{t}_j$. Therefore, the following holds:

$$\text{Eq. (60)} = \sum_{y,\mathbf{z},\mathbf{x}} y P(y_{\mathbf{z}}, \bigwedge_j Z_{j[\mathbf{w}_j]} = z_j, \bigwedge_j \mathbf{X}_{\mathbf{t}_j}^j = \mathbf{x}^j) \qquad\qquad \text{consistency} \qquad (67)$$

$$= \sum_{y,\mathbf{z},\mathbf{x}} y P(y_{\mathbf{z}}, \bigwedge_j Z_{j[\mathbf{w}_j \mathbf{x}^j]} = z_j, \bigwedge_j \mathbf{X}_{\mathbf{t}_j}^j = \mathbf{x}^j) \qquad\qquad \text{CTF-Rule 1} \qquad (68)$$

$$= \sum_{y,\mathbf{z},\mathbf{x}} y P(y_{\mathbf{z}}, \bigwedge_j Z_{j[\mathbf{w}_j \mathbf{x}^j]} = z_j, \bigwedge_j \mathbf{X}_*^j = \mathbf{x}^j) \qquad\qquad \text{consistency} \qquad (69)$$

$$= \sum_{y,\mathbf{z},\mathbf{x}} y P(y_{\mathbf{z}}, \bigwedge_j Z_{j[\mathbf{x}^j]} = z_j, \bigwedge_j \mathbf{X}_*^j = \mathbf{x}^j) \qquad\qquad \text{CTF-Rule 3} \qquad (70)$$

where the last equation holds due to $\mathbf{T}_j \cap \mathtt{An}(Z_j)_{\mathcal{G}_{\overline{\mathbf{V}(\mathbf{X}_*^j)}}} = \emptyset$ (see the construction of $\mathbf{X}_*^j$).

**Claim 2.** By the construction of $\mathbf{X}_*^j$ and consistency, we have $Z_{j[\mathbf{x}_j]} = Z_{j[\mathbf{x}]}$, which leads to Eq. (71). Furthermore, according to $\mathbf{V}(\mathbf{X}_*) \cap \mathtt{An}(Y)_{\mathcal{G}_{\overline{\mathbf{V}(\mathbf{Z}_*)}}} = \emptyset$ (by the construction of $\mathbf{X}_*^j$), we can apply CTF-Rule 3, as shown in Eq. (72). Therefore, we can write:

$$\text{Eq. (61)} = \sum_{y,\mathbf{z},\mathbf{x}} y P(y_{\mathbf{z}}, \bigwedge_j Z_{j[\mathbf{x}]} = z_j, \bigwedge_j \mathbf{X}_*^j = \mathbf{x}^j) \qquad\qquad \text{consistent } (\mathbf{x}^{j_1}, \mathbf{x}^{j_2}) \qquad (71)$$

$$\begin{aligned} &= \sum_{y, \mathbf{z}, \mathbf{x}} y P(y_{\mathbf{z}}, \mathbf{z}_{\mathbf{x}}, \bigwedge_j \mathbf{X}_*^j = \mathbf{x}^j) && \text{def} && (72) \\ &= \sum_{y, \mathbf{z}, \mathbf{x}} y P(y_{\mathbf{z}\mathbf{x}}, \mathbf{z}_{\mathbf{x}}, \bigwedge_j \mathbf{X}_*^j = \mathbf{x}^j). && \text{CTF-Rule 3} && (73) \end{aligned}$$

This concludes the proof. $\qquad\square$

---

**Algorithm 3:** Construct counterfactual world SCM $\mathcal{M}_{\mathbf{X}_*}$

---

**Input:** SCM $\mathcal{M}$; a CTF-MIS $\mathbf{X}_*$; and a reward variable $Y$;
**Output:** counterfactual world SCM $\mathcal{M}_{\mathbf{X}_*}$.

1 Initialize $\mathcal{M}_{\mathbf{X}_*} = \langle \mathbf{V}, \mathbf{U}, \mathcal{F}, P(\mathbf{U}) \rangle$.
2 **for** each counterfactual variable $X_{i[\mathbf{w}_i]} \in \mathbf{X}_*$ **do**
3     **for** each ancestral counterfactual $T_{\mathbf{z}} \in \text{An}(X_{i[\mathbf{w}_i]})$ **do**
4         **if** $f_T \in \mathcal{F}$ has arguments as $\mathbf{Z}' \subseteq \mathbf{Z}$ **then**
5             Set the arguments $\mathbf{Z}'$ in $f_T \in \mathcal{F}_{\mathbf{X}_*}$ as fixed values $\mathbf{z}'$.
6 **return** $\mathcal{M}_{\mathbf{X}_*}$.

---

### F.4 PROOF OF COROLLARY 3

**Corollary 3.** *Given a* CTF*-MIS* $\mathbf{X}_*$, *the counterfactual regime graph* $\mathcal{H}_{\mathbf{X}_*}$ *is a subgraph of the causal diagram compatible with* $\mathcal{M}_{\mathbf{X}_*}$ *over* $\mathbf{V}^\dagger$.

*Proof.* Let $\mathbf{X}_* = \{X_{i[\mathbf{w}_i]}\}_{i=1}$ be a CTF-MIS with respect to $\langle \mathcal{G}, Y \rangle$. An SCM can be constructed as follows the procedure shown in Algo. 3, which mirrors the definition of $do(\mathbf{X} = \mathbf{X}_*)$. According to realizability (Prop. 1), when $T_{\mathbf{z}} \in \text{An}(X_{i[\mathbf{w}_i]})$ and $T_{\mathbf{s}} \in \text{An}(X_{j[\mathbf{w}_j]})$ for $i \neq j$, we have $\mathbf{z} = \mathbf{s}$. Therefore, Algo. 3 mirrors the way in which the values of $\mathbf{V}$ is determined under $\mathbf{X}_*$. We denote by $\mathcal{G}_{\mathbf{X}_*}$, the causal diagram (Def. 8) of $\mathcal{M}_{\mathbf{X}_*}$. Then, all edges $Z \to T$ are removed from $\mathcal{G}$ for any $Z \in \mathbf{Z} \ (= \mathbf{W}_i \cap \text{An}(T)_{\mathcal{G}_{\overline{\mathbf{W}_i}}} \subseteq \mathbf{W}_i)$ in $\mathcal{G}_{\mathbf{X}_*}$. Hence, we obtain the counterfactual regime graph $\mathcal{H}_{\mathbf{X}_*} = \mathcal{G}_{\mathbf{X}_*}[\text{An}(Y)_{\mathcal{G}_{\mathbf{X}_*}}] = \mathcal{G}_{\mathbf{X}_*}[\mathbf{V}^\dagger]$ where $\mathbf{V}^\dagger = \mathbf{V}(\text{An}(Y_{\mathbf{x}}, \mathbf{X}_*))$. Therefore, $\mathcal{H}_{\mathbf{X}_*}$ is a subgraph of the causal diagram $\mathcal{G}_{\mathbf{X}_*}$ over $\mathbf{V}^\dagger$. This concludes the proof. $\qquad\square$

### F.5 PROOF OF THEOREM 2 AND COROLLARY 1

**Theorem 2** (Graphical characterization of CTF-POMIS). *A* CTF*-MIS* $\mathbf{X}_*$ *with respect to* $\langle \mathcal{G}, Y \rangle$ *is a* CTF*-POMIS if and only if* $\text{IB}(\mathcal{H}_{\mathbf{X}_*}, Y) = \emptyset$ *holds.*

*Proof.* **(If)** Suppose $\text{IB}(\mathcal{H}_{\mathbf{X}_*}, Y) = \emptyset$ holds. By the definition of interventional border, this means $\mathbf{V}^\dagger = \text{MUCT}(\mathcal{H}_{\mathbf{X}_*}, Y)$ holds. The intuition of the proof of this direction is to construct an SCM, conforming to $\mathcal{G}$, for which the single best strategy involves intervening on $\mathbf{X}_*$. In this proof, every exogenous variable is a binary variable with its domain being $\{0, 1\}$. Let $\oplus$ denote the exclusive-or function and $\bigvee$ the logical OR operator. The proof follows a similar argument to that of the proof of Proposition 4 in Lee and Bareinboim (2018).

An easy case is when $\text{MUCT}(\mathcal{H}_{\mathbf{X}_*}, Y) = \{Y\}$ holds; thus, we obtain $\mathbf{W} \triangleq \bigcup_i \mathbf{W}_i = \text{Pa}(Y)_{\mathcal{G}}$ and $\mathbf{X}_* = \text{Pa}(Y)_{\mathcal{G}} \setminus \{Y\} = \mathbf{W}_{\mathbf{w}}$. In this case, we can express $\mathbb{E}Y_{\mathbf{X}_*} = \mathbb{E}Y_{\mathbf{w}} = \mathbb{E}[Y \mid do(\mathbf{w})]$. We construct an SCM such that (i) Each endogenous variable $V \in \mathbf{V}$ associates with an unobserved variable $U_V$; (ii) $f_Y = 1 - (\bigvee \mathbf{u}_Y \oplus (\bigvee \mathbf{pa}_Y)) \approx 1$ with $P(\mathbf{U}_Y = 0) \approx 1$; and (iii) $f_V = (\bigoplus \mathbf{u}_V) \oplus (\bigoplus \mathbf{pa}_V)$ for $V \in \mathbf{V} \setminus \{Y\}$ and $U \in \mathbf{U} \setminus \mathbf{U}_Y$ following a fair coin $\texttt{Bern}(0.5)$. By taking conditional expectations, it holds:

$$\begin{aligned} \mathbb{E}[Y \mid do(\mathbf{W} = 0)] &= \mathbb{E}[Y \mid do(\text{Pa}(Y)_{\mathcal{G}} = 0)] && (74) \\ &= \mathbb{E}[Y \mid do(\text{Pa}(Y)_{\mathcal{G}} = 0), \mathbf{U}_Y \neq 0] P(\mathbf{U}_Y \neq 0) && (75) \\ &+ \mathbb{E}[Y \mid do(\text{Pa}(Y)_{\mathcal{G}} = 0), \mathbf{U}_Y = 0] P(\mathbf{U}_Y = 0) && (76) \\ &= P(\mathbf{U}_Y = 0) \approx 1. && (77) \end{aligned}$$

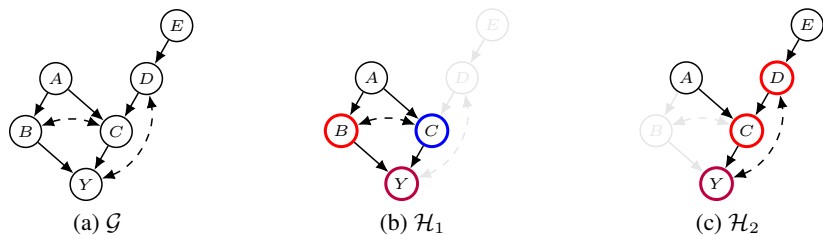

(a) $\mathcal{G}$      (b) $\mathcal{H}_1$      (c) $\mathcal{H}_2$

Figure 13: (a) Causal diagram; (b, c) colored subgraphs for each unobserved confounders $U_1$ and $U_2$.

|  |  | $\mathcal{M}_1$ | | | $\mathcal{M}_2$ | | | $\mathcal{M}$ | | | | |
| --- | --- | --- | --- | --- | --- | --- | --- | --- | --- | --- | --- | --- |
| $U_1$ | $U_2$ | $B^{(1)}$ | $C^{(1)}$ | $Y^{(1)}$ | $D^{(2)}$ | $C^{(2)}$ | $Y^{(1)}$ | $B$ | $D$ | $C$ | $Y'$ | $Y$ |
| 0 | 0 | 0 | 1 | 2 | 1 | 1 | 2 | 00 00 | 00 01 | 00 01 | 10 10 | **1** |
|  | 1 |  |  |  | 0 | 0 | 1 |  | 00 00 | 00 00 | 10 01 |  |
| 1 | 0 | 1 | 0 | 1 | 1 | 1 | 2 | 01 00 | 00 01 | 00 01 | 01 10 |  |
|  | 1 |  |  |  | 0 | 0 | 1 |  | 00 00 | 00 00 | 01 01 |  |

Table 3: Assignments with $B_{a=0}, C_{a'=1,e=1}$ where values for $\mathcal{M}$. The target variables are shown as bit sequences, e.g., $Y'$ represents $(4y^{(1)} + y^{(2)})_2$.

Meanwhile, all other interventions yield expectations less than or equal to $0.5$. Therefore, $\mathbf{X}_* = \mathbf{W_w}$ is a CTF-POMIS with respect to $\langle \mathcal{G}, Y \rangle$.

Now, we consider a general case where $\{Y\} \subsetneq \mathbf{V}^\dagger$. That is, there exists at least one unobserved confounder between $Y$ and its ancestors. As a first step, it will be shown that there exists an SCM $\mathcal{M}$ conforming to $\mathcal{G}$ where $do(\bigwedge_i X_{i[\mathbf{W}_i = \texttt{ord}(X_i) \pmod 2]})$ is the single best strategy. Let $\mathbf{U}' = \{U_j\}_{j=1}^k$ be the set of unobserved confounders in $\mathcal{H} = \mathcal{G}[\mathbf{V}^\dagger \cup \mathbf{W}]$.

Given $U_j \in \mathbf{U}'$, let $B^{(j)}$ and $R^{(j)}$ denote its two children. We define an SCM $\mathcal{M}_j$ where the graph structure is given by:

$$\mathcal{H}_j = \mathcal{H}[\texttt{De}(\{B^{(j)}, R^{(j)}\})_\mathcal{H} \cup (\mathbf{W} \cap \texttt{Pa}(\texttt{De}(\{B^{(j)}, R^{(j)}\})_\mathcal{H})_\mathcal{H})] \tag{78}$$

with all bidirected edges removed except $U_j$. In order to set the mechanisms for variables in $\mathcal{H}_j$, the vertices will be labeled as described below.

We label (i) vertices in $\texttt{De}(B^{(j)}) \setminus \texttt{De}(R^{(j)})$ as blue; (ii) $\texttt{De}(R^{(j)}) \setminus \texttt{De}(B^{(j)})$ as red; and (iii) $\texttt{De}(B^{(j)}) \cap \texttt{De}(R^{(j)})$ as purple. Each of $B^{(j)}$ and $R^{(j)}$ perceives $U_j$ as a parent colored blue with value $u_j$ and red with value $1 - u_j$, respectively.

Each variable $X$—blue, red, and purple colored–are assigned to 3 if any value of their parents in $\mathbf{W}$ is *not* $\texttt{ord}(X) \pmod 2$[10]. Otherwise, their values are determined as follows. For every blue and red vertex, the associated structural equation returns the common value of its parents of the same color and returns **3** if the values of the colored parents are *not* homogeneous. For every purple vertex, its corresponding equation returns **2** if every blue, red and purple parent is 0,1,2, respectively, and returns **1** if 1,0,1, respectively. For other cases, the function returns **3**.

We now merge the $k$ SCMs $\{\mathcal{M}_j\}_{j=1}^k$ into one single SCM that is compatible with $\mathcal{H}$. In $\mathcal{M}_j$, two bits are sufficient to represent every variable. We build a unified SCM where each variable in $\mathbf{V}^\dagger$ is represented with $2k$ bits where an SCM for $U_j$ will take $2j-1_\text{th}$ and $2j_\text{th}$ bits. We then binarize $Y$ by setting 1 if $2j-1_\text{th}$ and $2j_\text{th}$ bits are 01 or 10 for every $j$ and 0 otherwise. Let $P(u_j = 1) = 0.5$ for $U_j \in \mathbf{U}'$. This unified SCM $\mathcal{M}$ provides a core mechanism to output $Y = 1$ if $do(\bigwedge_i X_{i[\mathbf{W}_i = \texttt{ord}(X_i) \pmod 2]})$ and $Y = 0$ otherwise. If any of variable in $\mathbf{V}^\dagger$ is intervened, then at least one sub-SCM will be disrupted yielding an expectation smaller than or equal to $0.5$.

---

[10]That is, the parity of the ASCII code of $X$. Note that all possible CTF-POMIS configurations can be obtained by applying permutations to the variable labels.

The previously defined SCM for $\mathcal{H} = \mathcal{G}[\mathbf{V}^\dagger \cup \mathbf{W}]$, will be extended to an SCM for $\mathcal{G}$. However, we can ignore joint probability distributions for any exogenous variables only affecting endogenous variables outside of $\mathcal{H}$. Setting structural equations for endogenous variables outside of $\mathcal{H}$ is redundant as well. For $V \in \text{An}(Y)_{\mathcal{G}} \setminus \mathbf{V}^\dagger$, we define the structural equations as $f_V = (\bigoplus \mathbf{u}_V) \oplus (\bigoplus \mathbf{pa}_V)$. For $U \in \mathbf{U} \setminus \mathbf{U}'$, we set $P(U = 0) = 0.5$ if $U$'s child(ren) is disjoint to $\mathbf{V}^\dagger$, and $P(U_V \equiv \text{ord}(V) \pmod 2) \approx 1$ if it intersects with $V \in \mathbf{V}^\dagger$. Note that $do(\bigwedge_i X_{i[\mathbf{W}_i = \text{ord}(X_i) \pmod 2]})$ is still the single optimal counterfactual intervention. Therefore, $\mathbf{X}_*$ is a CTF-POMIS with respect to $\langle \mathcal{G}, Y \rangle$.

We provide an example in Fig. 13 illustrating how $k$ SCMs are constructed. Further, values of variables for $\mathcal{M}_1$, $\mathcal{M}_2$ and a unified $\mathcal{M}$ are shown in Table 3.

**(Only if)** We will prove the contrapositive statement; that is, if $\text{IB}(\mathcal{H}_{\mathbf{X}_*}, Y) \neq \emptyset$, then a CTF-MIS $\mathbf{X}_* = \{X_{i[\mathbf{w}_i]}\}_{i=1}$ is *not* a CTF-POMIS with respect to $\langle \mathcal{G}, Y \rangle$. Let $\mathbf{T} = \{T_j\}_{j=1} \triangleq \text{IB}(\mathcal{H}_{\mathbf{X}_*}, Y) \neq \emptyset$. We denote by $\mathbf{X}_*^* = \{X_{i[\mathbf{w}_i^\dagger]}\}_{i=1}$, the assigned value of $\mathbf{W}_i$ consisting of the best action of $\mathbf{X}_*$.

Furthermore, we consider $\mathbf{Z}_* = \{Z_{k[\mathbf{t}_k]}\}_{k=1}$ such that $Z_i \in \text{MUCT}(\mathcal{H}_{\mathbf{X}_*}, Y)$ with $\bigcup_k \mathbf{T}_k = \mathbf{T}$. To maintain consistency (i.e., to avoid any conflicts that lead to non-realizability), for any $Z_{i[\mathbf{t}_i]} \in \mathbf{Z}_*$, if $Z_i \in \mathbf{V}(\mathbf{X}_*)$, then $Z_{i[\mathbf{t}_i]} = Z_{i[\mathbf{w}_i \cap \mathbf{T}_i, \mathbf{t}_i \setminus \mathbf{W}_i]}$. We denote $\mathbf{X}' \triangleq (\mathbf{X} \setminus \mathbf{Z}) \cap \text{An}(\mathbf{T})_{\mathcal{G}}$ and $\mathbf{X}'' \triangleq (\mathbf{X} \setminus \mathbf{Z}) \setminus \mathbf{X}'$. Consistently, we denote $\mathbf{X}_*' = \{\mathbf{X}_{i[\mathbf{w}_i] \in \mathbf{X}_*} \mid X_i \in \mathbf{X}'\}$. We are now ready to derive as follows:

$$\mu_{\mathbf{X}_*^*} = \sum_{y, \mathbf{x}} y P(y_{\mathbf{x}}, \bigwedge_i x_{i[\mathbf{w}_i^\dagger]}) \qquad \text{CUT} \quad (79)$$

$$= \sum_{y, \mathbf{x} \cup \mathbf{z}} y P(y_{\mathbf{x}}, \bigwedge_k z_{k[\mathbf{x} \setminus \mathbf{Z}]}, \bigwedge_i x_{i[\mathbf{w}_i^\dagger]}) \qquad \text{marginalization} \quad (80)$$

$$= \sum_{y, \mathbf{x} \cup \mathbf{z}} y P(y_{\mathbf{x} \setminus \mathbf{Z}, \mathbf{z}}, \bigwedge_k z_{k[\mathbf{x} \setminus \mathbf{Z}]}, \bigwedge_i x_{i[\mathbf{w}_i^\dagger]}) \qquad \text{CTF-Rule 1} \quad (81)$$

$$= \sum_{y, \mathbf{x} \cup \mathbf{z}, \mathbf{t}} y P(y_{\mathbf{x} \setminus \mathbf{Z}, \mathbf{z}}, \bigwedge_k z_{k[\mathbf{x} \setminus \mathbf{Z}]}, \bigwedge_i x_{i[\mathbf{w}_i^\dagger]}, \mathbf{t}_{\mathbf{x} \setminus \mathbf{Z}}) \qquad \text{marginalization} \quad (82)$$

$$= \sum_{y, \mathbf{x} \cup \mathbf{z}, \mathbf{t}} y P(y_{\mathbf{x} \setminus \mathbf{Z}, \mathbf{z}}, \bigwedge_k z_{k[\mathbf{x} \setminus \mathbf{Z}, \mathbf{t}]}, \bigwedge_i x_{i[\mathbf{w}_i^\dagger]}, \mathbf{t}_{\mathbf{x} \setminus \mathbf{Z}}) \qquad \text{CTF-Rule 1} \quad (83)$$

$$= \sum_{y, \mathbf{x} \cup \mathbf{z}, \mathbf{t}} y P(y_{\mathbf{x} \setminus \mathbf{Z}, \mathbf{z}}, \bigwedge_k z_{k[\mathbf{t}_k]}, \bigwedge_i x_{i[\mathbf{w}_i^\dagger]}, \mathbf{t}_{\mathbf{x} \setminus \mathbf{Z}}) \qquad \text{CTF-Rule 3} \quad (84)$$

$$= \sum_{y, \mathbf{x} \cup \mathbf{z}, \mathbf{t}} y P(y_{\mathbf{x} \setminus \mathbf{Z}, \mathbf{z}}, \bigwedge_{\mathbf{Z} \cap \mathbf{X}} z_{k[\mathbf{w}_k^\dagger \cap \mathbf{T}_k, \mathbf{t}_k \setminus \mathbf{W}_k]}, \bigwedge_{\mathbf{Z} \setminus \mathbf{X}} z_{k[\mathbf{t}_k]}, \bigwedge_{\mathbf{X} \setminus \mathbf{Z}} x_{i[\mathbf{w}_i^\dagger]}, \mathbf{t}_{\mathbf{x} \setminus \mathbf{Z}}) \qquad \text{algebra} \quad (85)$$

$$= \sum_{y, \mathbf{x} \cup \mathbf{z}, \mathbf{t}} y P(y_{\mathbf{x}' \mathbf{x}'' \mathbf{z}}, \bigwedge_{\mathbf{Z} \cap \mathbf{X}} z_{k[\mathbf{w}_k^\dagger \cap \mathbf{T}_k, \mathbf{t}_k \setminus \mathbf{W}_k]}, \bigwedge_{\mathbf{Z} \setminus \mathbf{X}} z_{k[\mathbf{t}_k]}, \bigwedge_{\mathbf{X} \setminus \mathbf{Z}} x_{i[\mathbf{w}_i^\dagger]}, \mathbf{t}_{\mathbf{x}' \mathbf{x}''}) \qquad \text{def} \quad (86)$$

$$= \sum_{y, \mathbf{x} \cup \mathbf{z}, \mathbf{t}} y P(y_{\mathbf{x}'' \mathbf{z}}, \bigwedge_{\mathbf{Z} \cap \mathbf{X}} z_{k[\mathbf{w}_k^\dagger \cap \mathbf{T}_k, \mathbf{t}_k \setminus \mathbf{W}_k]}, \bigwedge_{\mathbf{Z} \setminus \mathbf{X}} z_{k[\mathbf{t}_k]}, \bigwedge_{\mathbf{X} \setminus \mathbf{Z}} x_{i[\mathbf{w}_i^\dagger]}, \mathbf{t}_{\mathbf{x}'}) \qquad \text{CTF-Rule 3} \quad (87)$$

$$= \sum_{y, \mathbf{x}'', \mathbf{z}, \mathbf{t}} y P(y_{\mathbf{x}'' \mathbf{z}}, \bigwedge_{\mathbf{Z} \cap \mathbf{X}} z_{k[\mathbf{w}_k^\dagger \cap \mathbf{T}_k, \mathbf{t}_k \setminus \mathbf{W}_k]}, \bigwedge_{\mathbf{Z} \setminus \mathbf{X}} z_{k[\mathbf{t}_k]}, \bigwedge_{\mathbf{X}''} x_{i[\mathbf{w}_i^\dagger]}, \mathbf{t}_{\mathbf{X}_*'}) \qquad \text{CUT} \quad (88)$$

$$= \sum_{y, \mathbf{x}'', \mathbf{z}, \mathbf{t}} y P(y_{\mathbf{x}'' \mathbf{z}}, \bigwedge_{\mathbf{Z} \cap \mathbf{X}} z_{k[\mathbf{w}_k^\dagger \cap \mathbf{T}_k, \mathbf{t}_k \setminus \mathbf{W}_k]}, \bigwedge_{\mathbf{Z} \setminus \mathbf{X}} z_{k[\mathbf{t}_k]}, \bigwedge_{\mathbf{X}''} x_{i[\mathbf{w}_i^\dagger]}, ) P(\mathbf{t}_{\mathbf{X}_*'}) \qquad \textbf{MUCT\&IB} \quad (89)$$

$$\leq \mu_{\{\mathbf{X}_*'' \cup \mathbf{Z}_*\}^*} \qquad (90)$$

where Eq. (89) holds, since $\mathbf{T} = \text{IB}(\mathcal{H}_{\mathbf{X}_*}, Y)$ is not confounded with any variables in $\mathbf{X}'' \cup \mathbf{Z}$ and its descendants. Therefore, $\mathbf{X}_*$ is *not* a CTF-POMIS with respect to $\langle \mathcal{G}, Y \rangle$, leading to a contradiction. This concludes the proof of this direction. $\qquad \square$

**Corollary 1.** *Let $\mathbf{x}^\star = \arg\max_{\mathbf{x} \in \mathcal{D}_{\mathbf{X}}, \mathbf{X} \subseteq \mathbf{V} \setminus \{Y\}} \mu_{\mathbf{x}}$ be an optimal arm in $\mathcal{L}_{\leq 2}$. Then, $\mu_{\mathbf{x}^\star} \leq \mu_{\mathbf{X}_*^\star}$.*

*Proof.* First, since $\mathbf{x}^\star = \mathbf{X}_{\mathbf{x}^\star}$, $\mathcal{L}_{\leq 2}$ optimal actions are necessarily contained in the CTF-SCB action space $\mathcal{A}$. Therefore, $\mu_{\mathbf{x}^\star} \leq \mu_{\mathbf{X}_*^\star}$ holds. Furthermore, unless the POMIS and CTF-POMIS spaces are exactly the same (e.g., Markovian settings as shown in Cor. 2), we can construct an SCM such that $\mu_{\mathbf{x}^\star} < \mu_{\mathbf{X}_*^\star}$, following the construction in the proof of Thm. 2 (see Fig. 13 and Table 3). $\qquad \square$

### F.6    PROOF OF PROPOSITION 3

**Theorem 3.** *The Algorithm 1 returns all and only representative* CTF-*POMISs given* $\langle \mathcal{G}, Y \rangle$.

*Proof.* When constructing the map pa, if $X = Y$, we replace $X$ with $W$ for all $X \in \mathbf{V}(\mathcal{H})$ and $W \in \mathrm{pa}[X]$ since $Y_{Y_w} = Y_{W_w}$ (Sec. 7.3 in Pearl (2000)). This substitution allows the algorithm to reformulate any regime to be defined over $X_i \in \mathbf{V} \setminus \{Y\}$ and $\mathbf{W}_i \subseteq \mathbf{V} \setminus \{Y\}$ for any $X_{i[\mathbf{w}_i]} \in \mathbf{X}_*$. Excluding the case where $X = Y$, any $\mathcal{L}_2$-level intervention can be written in the form of a subscript $\mathbf{C}_W$, where $\mathbf{C}_W$ denotes $\mathrm{Ch}(W)_\mathcal{G}$.

Therefore, the brute-force manner over all edge selections (based on Prop. 3) considers *all* possible equivalent classes of regimes. Since verification of CTF-MISIFY is straightforward from the completeness of Thm. 1, and its validity as an action is ensured by Prop. 1 and Prop. 5, we conclude that this counterfactual symbol constitutes a valid representative CTF-POMIS, and that the algorithm yields all such representations. $\qquad\square$

## IMPACT STATEMENT

This work addresses a counterfactual structural causal bandit framework that leverages counterfactual-level causal reasoning from a causal diagram. The approach has potential applications in practical settings such as personalized healthcare, adaptive education, and resource-constrained recommendation systems, where a decision-maker seeks to make optimal decisions by considering counterfactual actions (future desirable regimes) with counterfactual mediators. However, improper specification of causal structures or inappropriate counterfactual mediator selection may lead to misleading conclusions and biased decisions. Therefore, careful validation and domain-specific causal modeling are essential prior to deployment in high-stakes environments.

