# OpenReview forum: "Counterfactual Structural Causal Bandits"
_ICLR.cc/2026/Conference — ICLR 2026 Poster_

### Official Review · Reviewer_nMUN · 2025-10-29

**Soundness:** 3
**Presentation:** 3
**Contribution:** 2
**Rating:** 4
**Confidence:** 3

**Summary:**

This paper extends the structural causal bandit framework to a counterfactual scenario, where interventions are mixed counterfactuals. In this setup, the action space is defined to satisfy ancestral consistency. Leveraging the existing result on the possibly-optimal minimal intervention set (POMIS), the paper developed a method to search for the POMISs in this counterfactual setup.

**Strengths:**

- The problem formulation is novel and relevant to the field.
- I skimmed through the theoretical results and found them sound.
- The experimental section effectively demonstrates the merits of the algorithm. In particular, it is helpful to see comparisons when the optimal action lies in $\mathcal{L}_{\leq 2}$.

**Weaknesses:**

- The challenge of the problem is unclear to me, as the method for finding POMISs is already available.

**Questions:**

- Could you explain the challenge in algorithm design? By checking Figure 7, Algorithm 1 seems to be an application of Lee and Bareinboim (2018) (algorithm to find POMISs).

- The counterfactual framework in the paper is somewhat confusing. Standard counterfactual inference typically requires observed data to constrain the exogenous variables, and then uses these constraints to reason about what would happen under a hypothetical intervention. Simply replacing the hard interventions in Lee and Bareinboim (2018) with counterfactual distributions does not appear to constitute a substantial contribution. Moreover, the mixed counterfactuals considered here could, in principle, be handled by embedding
$W$ into a multi-world SCM.

- In Page 2 "(e.g. $X_{1,[w_1]}$)", the  X should not be boldface.

---

> ### Author Response · Authors · 2025-11-17
>
> Thank you very much for your detailed and constructive review. We appreciate the concerns you raised and acknowledge that some parts of our paper may not have clearly communicated our main contributions; we hope the clarifications below will help refine your assessment.
>
> > [Summary] Leveraging the existing result on the possibly-optimal minimal intervention set (POMIS), the paper developed a method to search for the POMISs in this counterfactual setup.
> >
>
> A. We would like to clarify our main contributions: First, we define the CTF-SCB (counterfactual structural causal bandits) framework, in which an agent’s action corresponds to realizable $\mathcal{L}_3$ regimes $do(\mathbf{X}\_\ast)$, and we introduce CTF-MIS and CTF-POMIS with complete characterizations. In addition, recognizing that multiple CTF-POMISs may be equivalent, we introduce the notion of representative CTF-POMIS with an enumerating algorithm.
>
> We would like to respectfully clarify that our work **does not leverage** the $\mathcal{L}\_{\leq 2}$ POMISs of Lee and Bareinboim (2018), but instead develops CTF-POMIS. As shown in Corollary 1, the optimal action in $\mathcal{L}\_{\leq 2}$ is generally dominated by CTF-POMIS actions. Moreover, our experiments explicitly demonstrate that using $\mathcal{L}\_{\leq 2}$ POMIS suffers from linear regret with a  strict positive gap $\Delta\_{\mathbf{X}\_\ast} >0$.
>
> > [Q2-1] The counterfactual framework in the paper is somewhat confusing. Standard counterfactual inference typically requires observed data to constrain the exogenous variables, and then uses these constraints to reason about what would happen under a hypothetical intervention.
> >
>
> A. When performing counterfactual inference from observational data, we follow the standard procedure: (1) abduction, (2) action, and (3) prediction [Pearl, 2000]. The objective of counterfactual inference is to infer counterfactual quantities from observational data (and possibly experimental data). A typical goal is to estimate the *effect of the treatment on the treated* (ETT) $\mathbb{E}[Y_x \mid x']$.
>
> Indeed, prior work [Bareinboim et al, 2015; Forney et al., 2017] studies bandit settings in which agents aim to maximize $\mathbb{E}[Y_x \mid x']$ rather than $\mathbb{E}[Y_x]$, following the three-step procedure, i.e., the agent selects an action according to its natural intention, and then randomizes the actual decision. Building on this idea, Raghavan and Bareinboim (2025) further proposed a bandit strategy that leverages additional evidence of the form $\mathbb{E}[Y_x \mid x', z_{x''}]$. These established studies focus on extracting evidence about an agent’s intentions and using such information as **context**, placing them close to the personalized decision-making framework or contextual bandits. This forms a key distinction from our work. In contrast, our framework assumes a setting in which the agent can directly obtain counterfactual data (i.e., rewards) through online interaction enabled by counterfactual randomization. This comparison is addressed in the third paragraph in Appendix A (”Related Work” section).
>
> In summary, the goal of counterfactual inference is to infer (**identify or estimate**) $\mathcal{L}_3$ queries from $\mathcal{L}\_{\leq 2}$ data, whereas counterfactual action refers to directly **sampling** from the environment to obtain realizable $\mathcal{L}_3$ quantities.
>
> Additionally, we refer the reviewer to Appendix B.2—B.3 (“Counterfactual Calculus” and “Realizability” sections), where the distinction between *counterfactual identification* [Correa et al., 2021] and *counterfactual realizability* is discussed in detail with examples (see Figure 11).
>
> > [Q2-3] Moreover, the mixed counterfactuals considered here could, in principle, be handled by embedding W into a multi-world SCM.
> >
>
> A. We are glad to have the opportunity to compare our approach with one that explicitly introduces multi-world network such as counterfactual graphs [Shpitser and Pear, 2007] or AMWN [Correa and Bareinboim, 2025].
>
> Constructing such graphical representations requires applying the unnesting procedure, which **breaks the original consistency (CTF-Rule 1)** relations, as discussed in Section 3.2.1. This makes the existing multi-world structures unsuitable for our setting.
>
> Moreover, we show that if $\mathbf{X}\_\ast$ is realizable, then all relevant causal relations can be represented in a **single-world** form—without introducing multiple versions of the same variable (e.g., $X\_w, X\_{w'}$). This result is guaranteed by Corollary 3.
>
> In summary, existing multi-world constructions are not well suited for capturing realizable counterfactual actions, whereas our representation—-the *counterfactual regime graph*—maintains consistency and preserves a simple single world-form representation.

---

> ### Author Response · Authors · 2025-11-17
>
> > [W1] The challenge of the problem is unclear to me, as the method for finding POMISs is already available.
> >
>
> > [Q1] Could you explain the challenge in algorithm design? By checking Figure 7, Algorithm 1 seems to be an application of Lee and Bareinboim (2018) (algorithm to find POMISs).
> >
>
> > [Q2-2] Simply replacing the hard interventions in Lee and Bareinboim (2018) with counterfactual distributions does not appear to constitute a substantial contribution.
> >
>
> A. We are grateful for the opportunity to highlight the challenges and clarify our contributions. At first glance, one might view our work as a trivial extension of existing methods—-for example, by replacing the standard do-calculus [Pearl, 1995] with the CTF-calculus [Correa and Bareinboim, 2025]. However, as the saying goes, “*the devil is in the details.”* Extending SCB from $\mathcal{L}\_{\leq 2}$ to **realizable** $\mathcal{L}_3$ presents several nontrivial challenges, including the following:
>
> 1. **Iterated nested counterfactual regimes.** An agent may, in principle, need to consider arbitrary *nested* realizable counterfactual regimes as actions (e.g. $do({T\_{W\_x}})$). While we do not discuss this point in detail in the main body for readability, we defer to Appendix E.4 with a brief mention in Footnote 2. In Proposition 6 in Appendix E.4, we prove that any nested realizable counterfactual regime has an equivalent regime of the form $\mathbf{X}\_\ast \in \mathcal{A}$. This guarantees that it is not necessary to consider iterated nested counterfactual regimes. For instance, in the causal diagram shown in Figure 7a, the nested realizable regime $\lbrace T\_{W_{x}}, Z\_{x'}\rbrace$ is equivalent to $\lbrace W_x, Z\_{x'}\rbrace$, implying that the agent does not need to consider the nested version $\lbrace T\_{W_{x}}, Z\_{x'}\rbrace$. Thus, this theoretical foundation frees the agent from dealing with potentially *infinite* iterative nesting of counterfactual regimes.
> 2. **Realizability induces structural constraints.** We do not define an agent’s action as an arbitrary $\mathcal{L}_3$ regime, but rather as **realizable** regimes, since it only makes sense to regard $\mathbf{X}\_\ast$ as an action when the agent can actually perform it and sample rewards from it. This introduces non-trivial graphical constraints on $\mathbf{X}\_\ast$, as discussed in Appendix E.2. For example, consider the causal diagram in Figure 3b. If$X_w \in \mathbf{X}\_\ast$ and $Z \in \mathbf{V}(\mathbf{X}\_\ast)$, then for $\mathbf{X}\_\ast$to be a valid action, the subscripts of $Z$ must contain $w$. Moreover, this subscript entanglement directly affects the characterization of CTF-MIS through its second condition. Since this entanglement is induced by realizability, such structural constraints do **not** arise in the framework of Lee and Bareinboim (2018).
> 3. **Minimality does not fully encode equivalence.** As shown in Remark 1, while the notion of *minimality* is sufficient to characterize equivalence among $\mathcal{L}\_{\leq 2}$ actions, this is no longer the case in the counterfactual setting. This implies that CTF-POMIS is insufficient on its own, and there remains room to further eliminate redundant actions with respect to expected reward. To address this issue, Proposition 3 shows that for any CTF-POMIS, there is an equivalent action of the form $\lbrace X\_{i[\mathbf{w}_i]} \rbrace\_{i=1}$ where $X_i$ and $\mathbf{W}_i$ follow the child-parent relations. For example, $\{R_x\}$ in Figure 7b is a CTF-POMIS and $\lbrace Z_x\rbrace$ is an equivalent CTF-POMIS, while $\lbrace Z_x\rbrace$ satisfies the child-parent relations. We refer to such CTF-POMISs as representative CTF-POMIS. This proposition allows us to reduce the search problem to selecting edges (e.g., $X \to Z$) in order to identify representative CTF-POMIS, as leveraged in Algorithm 1. In summary, relying solely on CTF-POMIS is generally not enough to fully prune redundant and suboptimal actions.
>
> Furthermore, although Figure 7 may give the impression that Algorithm 1 simply applies the POMIS characterization of Lee and Bareinboim (2018), this is only because we intentionally designed the *counterfactual regime graph* so that CTF-POMIS can be characterized via the IB (Interventional Border) in a conceptually intuitive manner. However, the underlying theoretical challenges are fundamentally different, as discussed above. We hope the reviewer appreciates that these challenges partly account for the length and complexity of the proofs provided in Appendices E and F.
>
> > In Page 2 "(e.g. $\mathbf{X}_{1[\mathbf{w}_1]}$})", the X should not be boldface.
> >
>
> A. Thanks to the reviewer’s careful observation! We have updated the manuscript accordingly.
>
> We once again appreciate your constructive questions and hope that this discussion clarifies the concerns that arose from a misunderstanding. If you have any further questions or suggestions, please do not hesitate to let us know!

---

> ### Author Response · Authors · 2025-11-28
>
> Dear Reviewer `nMUN`
>
> We would like to kindly ask whether our response has addressed your concerns. In particular,
>
> - We clarified the misunderstanding regarding leveraging the existing $\mathcal{L}\_{\leq 2}$ POMIS algorithm [Lee and Bareinboim, 2018] by emphasizing that, as shown in Corollary 1 and our experiments, the $\mathcal{L}\_{\leq 2}$ POMIS approach suffers from linear regret with a strictly positive gap.
> - Furthermore, we elaborated on the core challenges of our work. While one might assume that actions require considering “iteratively nested counterfactual regimes,” we theoretically show that this is unnecessary. Moreover, due to the realizability property, the space of valid actions is constrained (see Proposition 1 and Proposition 5 in Appendix E.2), which leads to the non-trivial second condition of CTF-MIS—-something that does not arise in the $\mathcal{L}_{\leq 2}$ MIS setting.
>
> We believe that our rebuttal has been helpful in resolving your concerns. If you have any remaining questions or suggestions, please let us know. We will do our best to address them.
>
> Best, Authors 23429

---

### Official Review · Reviewer_Yncn · 2025-11-01

**Soundness:** 3
**Presentation:** 2
**Contribution:** 2
**Rating:** 4
**Confidence:** 2

**Summary:**

This paper introduces a variant of the causal bandit setting in which the agent has more power: they can also perform certain "counterfactual" interventions in which a variable $X$ is set to value $x$ as seen by one child, but to $x'$ as seen by another.

**Strengths:**

The topic is theoretically interesting. I have the impression that the theory is sound.

**Weaknesses:**

- I am not convinced of the significance of this contribution. My impression is that counterfactual actions as used here are only possible in practice under special circumstances. See also my first question below.
- The paper is very dense in technical material. Additionally, it builds closely on very recent work; familiarity with that work is necessary to build intuition about the present work. This makes the paper hard to review in a reasonable amount of time.

**Questions:**

- To what extent can counterfactual actions be modelled by defining a new graph which explicitly adds counterfactual mediators, and performing ordinary interventions on it?
- line 78-79 (3rd contribution): what does it mean that suboptimal interventions are "clearly" removed?
- line 100/101: I initially didn't understand what you meant by "when the variables are indexed". Now I think you mean: when the main variable already has a subscript, the counterfactual subscript is put between brackets for visual distinction. Could you confirm?
- In Proposition 1, what does it mean for a counterfactual to "consist of" an action space?

##### Comments
- The limitations section is in the supplement and is not referenced from the main paper.
- line 105: "correlated" should be "dependent" (only the same for Gaussians)

##### Textual
- line 31: "were" -> "was" ("were" is subjunctive mood, but this is factual)
- line 125: "behave**s**"
- Definition 3: "no ~~an~~other"
- line 282: "are cannot be"
- several places: "interventional bo~~a~~rder"

---

> ### Author Response · Authors · 2025-11-17
>
> We sincerely thank the reviewer for the constructive and thoughtful feedback. We respond to your comments as follows:
>
> > [W1] I am not convinced of the significance of this contribution. My impression is that counterfactual actions as used here are only possible in practice under special circumstances. See also my first question below.
> >
>
> > [C1] The limitations section is in the supplement and is not referenced from the main paper.
> >
>
> A. We first would like to emphasize that our work addresses a fundamental and theoretically in-depth problem within the SCM framework and provides a solid foundation for promising future research on counterfactual-level decision-making. While developing immediately applicable practical tools is certainly important, we believe that theoretical contributions of this nature are equally essential for advancing realizable counterfactual decision-making.
>
> However, we fully agree that the limitation related to practicability should be made more explicit to readers. Therefore, we have **moved the discussion from Appendix D to the main body (page 10)** to ensure that this point is clearly visible.
>
> > [Q1] To what extent can counterfactual actions be modelled by defining a new graph which explicitly adds counterfactual mediators, and performing ordinary interventions on it?
> >
>
> A. Thank you for your valuable question. We are glad to have the opportunity to compare our approach with one that explicitly adds counterfactual mediators to a new graph.
>
> Indeed, Raghavan and Bareinboim (2025) introduced the *expanded SCM* $\mathcal{M}^+$ and the corresponding *expanded causal graph* $\mathcal{G}^+$, which are defined over an enlarged set of endogenous variables $\mathbf{V}^+ \supset \mathbf{V}$ that explicitly includes counterfactual mediators (Definition E.1 in Raghavan and Bareinboim, 2025).
>
> However, one must handle ordinary $\mathcal{L}_2$-inference with care in such expanded structures for the following reasons: (1) Counterfactual mediators induce *deterministic* relations. By definition, the mapping $\tilde{X} \mapsto X$ is deterministic, as counterfactual mediators serve as substitutes for the original variable $X$. This determinism violates the positivity assumption (i.e., it is possible that $P(\mathbf{v}^+) = 0$ for some $\mathbf{v}^+$) that prevents the direct application of standard causal inference tools (e.g., d-separation) over the enlarged domain $\mathbf{V}^+$[Hwang et al, 2024]. (2) Furthermore, the expanded causal graph may append nodes according to the number of children, which results in an extremely large graph. (3) Moreover, performing interventions on the additional counterfactual mediators in such an expanded graph yields the resulting graph to take the form of our counterfactual regime graph. Consequently, the expanded structure introduces unnecessary graphical operations. (4) Finally, this procedure is insufficient to capture the equivalence among CTF-POMISs (i.e., *representative* CTF-POMIS), since this type of equivalence is unique to the counterfactual setting and does not appear under ordinary interventions.
>
> In summary, the expanded structure does not offer any advantages for characterizing the CTF-SCB framework; instead, it makes inference substantially more complicated, and fails to identify representative CTF-POMIS.
>
> - Hwang et al, *On Positivity Condition for Causal Inference*, ICML 2024
>
> > [W2] The paper is very dense in technical material. Additionally, it builds closely on very recent work; familiarity with that work is necessary to build intuition about the present work. This makes the paper hard to review in a reasonable amount of time.
> >
>
> A. We have made efforts to consistently follow the notation used in prior work [Bareinboim et al., 2022; Correa et al., 2021; Correa and Bareinboim, 2025, Raghavan and Bareinboim, 2025; Yang and Bareinboim, 2025]. Furthermore, to aid readers’ understanding, we provide more detailed background in Appendix B, including summaries of recent works such as counterfactual calculus [Correa and Bareinboim, 2025], and counterfactual realizability [Raghavan and Bareinboim, 2025].
>
> That said, we fully understand that our work is closely tied to several very recent developments [Correa and Bareinboim, 2025; Raghavan and Bareinboim, 2025; Yang and Bareinboim, 2025], which may place an additional burden on reviewers given the limited time available for evaluation. We sincerely appreciate the effort required to engage with these materials under such constraints, and we are fully committed to incorporating any feedback to further improve the readability of the manuscript.

---

> ### Author Response · Authors · 2025-11-17
>
> > [Q2] line 78-79 (3rd contribution): what does it mean that suboptimal interventions are "clearly" removed?
> >
>
> A. This means that our algorithm removes only the suboptimal interventions from the action space. We have revised the expression from “suboptimal interventions are **clearly** removed” to “**redundant or verifiably suboptimal** interventions are removed.” Thank you for helping us clarify this contribution.
>
> > [Q3] line 100/101: I initially didn't understand what you meant by "when the variables are indexed". Now I think you mean: when the main variable already has a subscript, the counterfactual subscript is put between brackets for visual distinction. Could you confirm?
> >
>
> A. Your understanding is correct. We have clarified this expression in the revised version.
>
> > [Q4] In Proposition 1, what does it mean for a counterfactual to "consist of" an action space?
> >
>
> A. In CTF-SCB, intervening on $\mathbf{X}\_\ast$ corresponds to taking an action. For example, intervention on $\lbrace W\_x,Z\_{x'} \rbrace$ in the introductory example constitutes a valid action of CTF-SCB. Accordingly,  we express this by saying that “$\mathbf{X}\_\ast$consists of the CTF-SCB action space $\mathcal{A}$” in Proposition 1.
>
> > [C2; Typo] line 105: "correlated" should be "dependent" (only the same for Gaussians); line 31: "were" -> "was" ("were" is subjunctive mood, but this is factual); line 125: "behave**s**"; Definition 3: "no ~~an~~other"; line 282: "are cannot be"; several places: "interventional bo~~a~~rder"
> >
>
> A. We have updated the manuscript accordingly.
>
> If you have any further questions or recommendations, please do not hesitate to let us know. We are confident that this discussion will further strengthen our manuscript.

---

> ### Author Response · Authors · 2025-11-28
>
> Dear Reviewer `Yncn`
>
> We would like to kindly ask whether our response has addressed your concerns. In particular,
>
> - we addressed your comment by explaining that the naive approach—-first extending the graph by explicitly introducing additional counterfactual mediators and then applying the existing $\mathcal{L}_{\leq2}$ level POMIS algorithm [Lee and Bareinboim, 2018]—-is not only inefficient but also fails to capture all redundant relations among actions.
> - In addition, we have incorporated your valuable suggestions into the revised manuscript, including correcting all typos and expressions and moving the limitations section into the main body.
>
> We sincerely hope that our rebuttal has been helpful in resolving your concerns. If you have any remaining questions or suggestions, please let us know. We will do our best to address them.
>
> Best, Authors 23429

---

### Official Review · Reviewer_cvdK · 2025-11-02

**Soundness:** 3
**Presentation:** 2
**Contribution:** 3
**Rating:** 6
**Confidence:** 2

**Summary:**

The paper extends the structural causal bandit framework by introducing **Counterfactual Structural Causal Bandits (CTF-SCB)**, wherein actions corresponds to realizable counterfactual regimes. It defines minimal counterfactual action sets (CTF-MIS), and further refines them by identifying those that are *possibly-optimal* (CTF-POMIS). Building on this, the authors present an enumeration algorithm with complexity $\mathcal{O}(n^2\cdot 2^{|E|})$ that systematically constructs a representative CTF-POMIS set suitable for standard bandit solvers . They prove that restricting exploration to this set preserves optimality and can reduce regret. Across synthetic tasks using Thompson Sampling and KL-UCB, the method consistently achieves lower cumulative regret than baselines that explore either larger counterfactual spaces (CTF-MIS) or purely lower-level action spaces (POMIS). In Markovian graphs, the procedure collapses to intervening on the parents of the outcome node, yielding no additional benefit from counterfactuals.

**Strengths:**

- Addresses an interesting problem and introduces a novel framework for realizable counterfactual interventions within causal bandits.
- Provides a clear and coherent motivation, positioning the extension of counterfactual reasoning to bandit settings as a natural and meaningful conceptual advance.
- Offers a potentially valuable theoretical foundation for subsequent research that may benefit from richer intervention classes in sequential decision-making.

**Weaknesses:**

- The exposition presupposes substantial familiarity with the CTF-calculus (Correa & Bareinboim, 2025) and related work (e.g., Correa et al., 2021). Consequently, several statements would benefit from further elaboration.
- The paper is quite dense, and the notation is not intuitive, which makes it hard to read.
- Although the substantial improvement over the super-exponential naive verification, the proposed algorithm remains exponential in the number of edges, raising questions about applicability.
- At present, the manuscript does not address finite-time regret guarantees; a short discussion would be beneficial.

**Questions:**

1. Could you please clarify the difference between $\boldsymbol{Pa}_V$ and $\boldsymbol{pa}_V$? If you are using the convention: capital letter -> variable and lowercase -> realization, where does the randomness come from in the $Pa$ operator, given a variable $V$?

2. What does $X_{\boldsymbol{w}}$ (line 102-103) refer to? You haven't defined before a rv $X$ with subscript $\boldsymbol{w}$

3. Could you please clarify how $\mathbb{E} N_T(\boldsymbol{X}_*)$ arises in eq. 1? Is an expectation missing?

4. Does $An(Y_x, X_*)$ mean $An(\\{Y_x, X_*\\})=An(Y_x) \cup An(X_*)$? Could you please clarify it in the paper?

5. It would be interesting if the authors can formally discuss (or better derive) finite-time regret guarantees that quantify the benefit of pruning to the representative CTF-POMIS set.


Suggestions:
- move the sentence "We use kinship notation for variable relationships..." (line 105) above mentioning Pa (line 94-95). Also the font used is different.
- end of line 100-101: $\boldsymbol{X}_{1[\boldsymbol{w}_1]}$ should not be bold.
- typo line 282: "... are cannot..".

---

> ### Author Response · Authors · 2025-11-17
>
> We appreciate the reviewer’s time and feedback provided to improve the manuscript. We respond to your comments as follows:
>
> > [W1] The exposition presupposes substantial familiarity with the CTF-calculus (Correa & Bareinboim, 2025) and related work (e.g., Correa et al., 2021). Consequently, several statements would benefit from further elaboration
> >
>
> > [W2] The paper is quite dense, and the notation is not intuitive, which makes it hard to read.
> >
>
> A. We have made efforts to consistently follow the notation used in prior work [Bareinboim et al., 2022; Correa et al., 2021; Correa and Bareinboim, 2025; Raghavan and Bareinboim, 2025; Yang and Bareinboim, 2025]. Furthermore, to aid readers’ understanding, we provide more detailed background in Appendix B, including summaries of SCB [Lee and Bareinboim, 2018], counterfactual calculus [Correa and Bareinboim, 2025], and counterfactual realizability [Raghavan and Bareinboim, 2025]. While some degree of notational complexity is unavoidable due to the nature of counterfactual inference, we have taken care to clarify key concepts and improve readability throughout the paper by providing various illustrative examples.
>
> If there are specific points that you believe would benefit from additional clarification or reorganization, we would be grateful to receive your suggestions and will carefully integrate them into the revised version.
>
> > [W3] Although the substantial improvement over the super-exponential naive verification, the proposed algorithm remains exponential in the number of edges, raising questions about applicability.
> >
>
> A. We appreciate the reviewer’s concern regarding the exponential complexity. We would like to note that the exponential worst-case complexity is an intrinsic property of the problem itself, stemming from the inherent combinatorial nature, rather than a weakness specific to our approach. For example, consider a causal diagram where all variables $V \in \mathbf{V} \setminus \lbrace Y\rbrace$ are parents of $Y$, and all pairs of variables $V_i, V_j \in \mathbf{V}$ are connected by bidirected edges. In this case, every subset of $\mathbf{V} \setminus \lbrace Y \rbrace$ becomes a representative CTF-POMIS, implying an exponential number of candidates regardless of the algorithm employed.
>
> > [W4] At present, the manuscript does not address finite-time regret guarantees; a short discussion would be beneficial.
> >
>
> > [Q5] It would be interesting if the authors can formally discuss (or better derive) finite-time regret guarantees that quantify the benefit of pruning to the representative CTF-POMIS set.
> >
>
> A. We appreciate your thoughtful suggestion. Our algorithm first shrinks the action space from $\mathcal{A}$ to the representative CTF-POMIS action space $\mathcal{A}^\star$ using graphical knowledge, and then applies standard bandit solvers (e.g., TS, KL-UCB) to $\mathcal{A}^\star$. These solvers enjoy well-established finite-time regret guarantees of order $\mathcal{O}(\sum_\{\mathbf{X}_\ast \in \mathcal{A}^\star : \Delta\_{\mathbf{X}\_{\ast}} > 0} \frac{\log T}{\Delta\_{\mathbf{X}\_{\ast}}})$.  We have added this clarification in Footnote 7 in the revised version.
>
> > [Q1] Could you please clarify the difference between $\mathbf{Pa}_V$ and $\mathbf{pa}_V$? If you are using the convention: capital letter -> variable and lowercase -> realization, where does the randomness come from in the operator, given a variable ?
> >
>
> A. Yes. $\mathbf{Pa}_V$ denotes the endogenous variables that serve as arguments of $f_V \in \mathcal{F}$, whereas  $\mathbf{pa}_V$ refers to their realizations. The randomness comes from exogenous variables $\mathbf{U}_V \subseteq \mathbf{U}$ and their distribution $P(\mathbf{U}_V)$ in the SCM framework $\mathcal{M} = \langle \mathbf{V},\mathbf{U}, \mathcal{F}, P(\mathbf{U}))$.
>
> > [Q2] What does (line 102-103) refer to? You haven't defined before a rv X with subscript w
> >
>
> A. Thank you for pointing this out. We have revised the corresponding part of the manuscript to make the notation explicit and to clarify the single counterfactual variable $X_\mathbf{w}$.

---

> ### Author Response · Authors · 2025-11-17
>
> > [Q3] Could you please clarify how arises $\mathbb{E}N_T(\mathbf{X}_\ast)$ in eq. 1? Is an expectation missing?
> >
>
> A. This derivation follows the *Regret decomposition lemma* [Lemma 4.5 in Lattimore and Szepesvári, 2020].
>
> Let $S_T = \sum\_{t=1}^T Y_{\mathbf{X}\_\ast^{(t)}}$ denote the total reward over $T$ rounds. Note that for any fixed $t \in [T]$ we have $\sum_{\mathbf{X}\_\ast \in \mathcal{A}} \mathbb{I}\lbrace \mathbf{X}\_\ast^{(t)} = \mathbf{X}\_\ast \rbrace = 1$. Hence, $S_T = \sum\_{t=1}^T\sum\_{\mathbf{X}\_\ast}Y_{\mathbf{X}\_\ast^{(t)}} \mathbb{I}\lbrace \mathbf{X}\_\ast^{(t)} = \mathbf{X}\_\ast\rbrace$, thus, the following holds:
>
> $$
> \begin{align} R_{T}^{\mathcal{A}} = T\mu\_{\mathbf{X}\_\ast^\star} - \mathbb{E}S_t
> = \sum\_{\mathbf{X}_\ast \in \mathcal{A}} \sum\_{t=1}^T \mathbb{E}[(\mu\_{\mathbf{X}\_\ast^\star}-Y\_{\mathbf{X}\_\ast^{(t)}}) \mathbb{I}\lbrace \mathbf{X}\_\ast^{(t)} - \mathbf{X}\_\ast \rbrace]\end{align}
> $$
>
> The expected reward in round $t$ intervened on $\mathbf{X}\_\ast^{(t)}$ is $\mu\_{\mathbf{X}_\ast^{(t)}}$, which means that
>
> $$
> \begin{align}
> \mathbb{E}[(\mu\_{\mathbf{X}\_\ast^\star}-Y\_{\mathbf{X}\_\ast^{(t)}}) \mathbb{I}\lbrace \mathbf{X}\_\ast^{(t)} = \mathbf{X}\_\ast\rbrace]
> & = \mathbb{I}\lbrace\mathbf{X}\_\ast^{(t)} = \mathbf{X}\_\ast\rbrace (\mu\_{\mathbf{X}\_\ast^\star}-\mu\_{\mathbf{X}\_\ast^{(t)}}) \\\\
> &= \mathbb{I}\lbrace\mathbf{X}\_\ast^{(t)}  = \mathbf{X}\_\ast\rbrace (\mu\_{\mathbf{X}\_\ast^\star}-\mu\_{\mathbf{X}\_\ast})
> \\\\
>  &= \mathbb{I}\lbrace\mathbf{X}\_\ast^{(t)} = \mathbf{X}\_\ast\rbrace \Delta\_{\mathbf{X}\_\ast}.
> \end{align}
> $$
>
> The result is completed by plugging this into Eq. (1) and using the definition of $N_T(\mathbf{X}\_\ast) = \sum\_{t=1}^T \mathbb{I}\lbrace \mathbf{X}\_\ast^{(t)} = \mathbf{X}\_\ast\rbrace$ (the number of times an action $\mathbf{X}\_\ast$ was chosen up to $T$).
>
> > [Q4] Does $An(Y_x,X_{\ast})$ mean $An(\{Y_x,X_{\ast}\}) = An(Y_x ) \cup An(X_{\ast})$? Could you please clarify it in the paper?
> >
>
> A. Yes. This expression is conventional in recent counterfactual inference literature [Correa and Bareinboim, 2025, Raghavan and Bareinboim, 2025; Yang and Bareinboim, 2025]. To clarify this notation, we have added Footnote 4 in the revised manuscript. Your suggestion greatly contributes to improving future readers’ understanding.
>
> > [S1] move the sentence "We use kinship notation for variable relationships..." (line 105) above mentioning Pa (line 94-95). Also the font used is different.
> >
>
> A. Thank you for pointing this out. We would like to clarify this slight distinction between $\mathbf{Pa}\_V$  and $\mathtt{Pa}(V)\_{\mathcal{G}}$. The former denotes the argument of the structural mechanism $f_V \in \mathcal{F}$ in the SCM $\mathcal{M} = \langle \mathbf{V},\mathbf{U}, \mathcal{F}, P(\mathbf{U}))$, whereas the latter refers to the graphical parents of $V$ in the causal diagram $\mathcal{G}$. Accordingly, in the manuscript, we use $\mathtt{Pa}(V)\_{\mathcal{G}}$ when emphasizing parent sets computed from a *given graph* $\mathcal{G}$, whereas $\mathbf{Pa}_V$ or its realization $\mathbf{pa}_V$ when referring to the actual parent variables or their values in the *underlying SCM* $\mathcal{M}$. This notational distinction is also used in Lee and Bareinboim (2018).
>
> That said, we agree that this distinction required further clarification. We have added “Note that $\mathtt{Pa}(V)_{\mathcal{G}}$ corresponds to $\mathbf{Pa}_V$”, and have therefore revised the wording from “kinship notation for **variable** relationships” to “kinship notation for **graphical** relationships”.
>
> > [S2] end of line 100-101: should not be bold.
> >
>
> > [S3] typo line 282: "... are cannot..”
> >
>
> A. Thank you for catching the typo. We have corrected it.
>
> We thank you again for your insightful and constructive suggestions. If you have any further questions or suggestions, please do not hesitate to let us know. We are confident that this discussion will help strengthen our manuscript!

---

> ### Author Response · Authors · 2025-11-28
>
> Dear Reviewer `cvdK`
>
> We would like to kindly ask whether our response has adequately addressed your concerns. We have incorporated all of your suggestions into the revised manuscript, which we believe has further strengthened the final version of our paper. We sincerely hope that our rebuttal has been helpful. If you have any remaining questions or suggestions, please feel free to let us know—-we will do our best to address them.
>
> Best, Authors 23429

---

### Official Review · Reviewer_qZvW · 2025-11-04

**Soundness:** 4
**Presentation:** 4
**Contribution:** 4
**Rating:** 8
**Confidence:** 4

**Summary:**

Summary
- SCB but also taking into account counterfactuals from L3
- Logical and crucial extension of theoretical base
- well executed

My review is short since I am familiar with the previous SCB work and can see this is an obvious and clear next step of extensions. It is written and present very clear, thorough, yet accessible, with the right amount of empirical experimental evidence.

**Strengths:**

- Very thorough, clear and precise presentation of
    - CTF MIS, CTF POMIS
    - Their algorithms
    - Their theorems

**Weaknesses:**

The paper is very well written and makes a clear and strong contribution, hence I only have one point on writing style:

- This reads a bit clunky with a rather long subclause” When X⋆∗ lies in L≤2 (i.e., ∆L≤2 = 0)—a special case and does not undermine our theoretical results, since the deployed agent can never be certain prior to interaction whether 471 the optimal arm lies in in L≤2—the smaller action space allows POMIS to converge faster than the 472 others.”
    - Maybe rephrase as: “a special scale that does not undermine”

**Questions:**

- In Fig 8, Task 3, left: is POMIS about to cross over CTFMIS TS at 100k trials?

---

> ### Author Response · Authors · 2025-11-17
>
> We sincerely thank the reviewer for the time and effort devoted to evaluating our work. We respond to your comments as follows:
>
> > [W1] The paper is very well written and makes a clear and strong contribution, hence I only have one point on writing style
> >
>
> A. Thank you for your valuable suggestion. We have revised the sentence accordingly by changing ”and” to “that”, as you recommended.
>
> > [Q1] In Fig 8, Task 3, left: is POMIS about to cross over CTFMIS TS at 100k trials?
> >
>
> A. Yes. As shown in Table 1 in Appendix B, POMIS with KL-UCB  crosses over CTF-MIS with TS at the final trial:
>
> - the cumulative regret (CR) of CTF-MIS with TS (purple dashed) is 5160.63, and
> - the CR of POMIS with KL-UCB (green solid) is 5179.9.
>
> Thank you for your positive assessment and valuable suggestions once again. If you have any further questions or suggestions, please do not hesitate to let us know!

---

### Author Response · Authors · 2025-11-17

We sincerely appreciate all reviewers for taking the time to review our paper. Your insightful suggestions and discussions will improve our work.

- The revised parts of the manuscript have been **highlighted in blue** for clarity.
- We follow ICLR’s the third code sharing policy:

> After we open the discussion forums for all submitted papers, we make a comment directed to the reviewers and area chairs and put a link to an anonymous repository.
>

Please refer to the anonymous Github link: https://anonymous.4open.science/r/CTF-SCB-45FE

---

### Meta-Review · Area_Chair_sBxh · 2026-01-10

**Summary:**

Overall all the reviewers are positive about the paper being the main concerns about the presentation of the paper which feels very dense and at times not easy to follow; and the lack of an explicit discussion on the limitations of the work.  Regarding the first point, the authors claim that they followed previous work notation and style, which I can see, and thus performed only minor changes to the paper. While the paper in that sense remains above the acceptance bar, I still believe the authors could significantly improve the presentation to make it more accessible (same for the papers they cite). The second line of concerns has been addressed by the authors with a completely new section that clarifies important points. Thus, overall the final version is a solid paper and contribution for the ICLR community.

**Reviewer Concerns:**

As stated above I do not think that major concerns remain, although definitely I would encourage the authors to improve the accessibility of their paper with a simpler notation and explanations of the convoluted math in layman terms.

**Reviewer Scores:**

I believe that overall the reviewers were positive and their final assessment would have remained or slightly improved due to the new limitation section and further clarifications.

---

### Decision · Program_Chairs · 2026-01-26

Accept (Poster)